# Stochastic Modified Equations and Dynamics of Dropout Algorithm

**Zhongwang Zhang**[1], **Yuqing Li**[1,2] *, **Tao Luo**[1,2,3,4,5] †, **Zhi-Qin John Xu**[1,3,4,6]‡

[1] School of Mathematical Sciences, Shanghai Jiao Tong University
[2] CMA-Shanghai, Shanghai Jiao Tong University
[3] Institute of Natural Sciences, MOE-LSC, Shanghai Jiao Tong University
[4] Qing Yuan Research Institute, Shanghai Jiao Tong University
[5] Shanghai Artificial Intelligence Laboratory
[6] Shanghai Seres Information Technology Company, Ltd

## Abstract

Dropout is a widely utilized regularization technique in the training of neural networks, nevertheless, its underlying mechanism and impact on achieving good generalization abilities remain to be further understood. In this work, we start by undertaking a rigorous theoretical derivation of the stochastic modified equations, with the primary aim of providing an effective approximation for the discrete iterative process of dropout. Meanwhile, we experimentally verify SDE's ability to approximate dropout under a wider range of settings. Subsequently, we empirically delve into the intricate mechanisms by which dropout facilitates the identification of flatter minima. This exploration is conducted through intuitive approximations, exploiting the structural analogies inherent in the Hessian of loss landscape and the covariance of dropout. Our empirical findings substantiate the ubiquitous presence of the Hessian-variance alignment relation throughout the training process of dropout.

## 1 Introduction

Dropout is a technique integrated into gradient-based algorithms for training neural networks (NNs) (Hinton et al., 2012; Srivastava et al., 2014). It constitutes a pivotal component contributing to the attainment of state-of-the-art test performance in deep learning (Tan and Le, 2019; Helmbold and Long, 2015). The key idea behind dropout is to randomly deactivate a subset of neurons during the training process. Specifically, the output of each neuron is multiplied by a random variable that takes the value $1/p$ with probability $p$ and zero otherwise. This random variable is independently sampled at each feedforward operation. Despite its widespread adoption and empirical success, the mechanism by which dropout enhances generalization in deep learning remains an ongoing area of research.

The noise structure introduced by stochastic algorithms plays a crucial role in understanding their training behaviors. A series of recent works reveal that the noise structure inherent in stochastic gradient descent (SGD) is vital for exploring flatter solutions (Keskar et al., 2016; Feng and Tu, 2021; Zhu et al., 2018). Analogously, the dropout algorithm introduces a specific form of noise, acting as an implicit regularizer that facilitates improved generalization abilities (Hinton et al., 2012; Srivastava et al., 2014; Wei et al., 2020; Zhang and Xu, 2022; Zhu et al., 2018).

In this paper, we first employ the stochastic modified equations (SMEs) (Li et al., 2017) framework to analyze the dynamics of the dropout algorithm applied to two-layer NNs. By application of SMEs, we embark on an exhaustive quantification of the leading order dynamics governing dropout, and we fortify this analytical approach through some empirical validations. In addition, we calculate the covariance matrix associated with the noise introduced by dropout. Hence our analytical exploration

---

*Corresponding author: liyuqing_551@sjtu.edu.cn
†Corresponding author: luotao41@sjtu.edu.cn
‡Corresponding author: xuzhiqin@sjtu.edu.cn

is further enriched by an investigation of the alignment relation between this covariance matrix and the Hessian matrix, a relationship conceptually framed as the Hessian-variance alignment relation (Zhu et al., 2018; Wu et al., 2022). We emphasize that this alignment property occupies a central role in sculpting the flatness attributes inherent in the solutions favored by NN models, and it has been firmly established that flatter solutions tend to exhibit enhanced generalization capabilities (Keskar et al., 2016; Neyshabur et al., 2017).

## 2 RELATED WORKS

A flurry of recent works aims to shed light on the regularization effect conferred by dropout. Wager et al. (2013) show that dropout performs a form of adaptive regularization in the context of linear regression and logistic problems. McAllester (2013) propose a PAC-Bayesian bound, whereas Wan et al. (2013); Mou et al. (2018) derive some Rademacher-complexity-type error bounds specifically tailored for dropout. Cavazza et al. (2018); Mianjy and Arora (2020); Wei et al. (2020); Arora et al. (2021) demonstrate that dropout regularizes the inductive bias under different settings. Jin et al. (2022) try to explain the generalization ability of dropout from the new perspective of weight expansion. Finally, Zhang and Xu (2022) establish that dropout facilitates condensation (Luo et al., 2021; Zhou et al., 2021; 2022) through an additional regularization term endowed by dropout.

Continuous formulations have been extensively utilized to study the dynamical behavior of stochastic algorithms. Li et al. (2017; 2019) present an entirely rigorous and self-contained mathematical formulation of the SME framework that applies to a wide class of stochastic algorithms. Furthermore, Feng et al. (2017) adopt a semigroup approach to investigate the dynamics of SGD and online PCA. Malladi et al. (2022) derive the SME approximations for the adaptive stochastic algorithms including RMSprop and Adam, additionally, they provide efficient experimental verification of the validity of square root scaling rules arising from the SMEs.

One noteworthy observation is the association between the flatness of minima and improved generalization ability (Li et al., 2017; Jastrzebski et al., 2017; 2018). Specifically, SGD is shown to preferentially select flat minima, especially under conditions of large learning rates and small batch sizes (Jastrzebski et al., 2017; 2018; Wu et al., 2018). Papyan (2018; 2019) attribute such enhancement of flatness by SGD to the similarity between covariance of the noise and Hessian of the loss function. Furthermore, Zhu et al. (2018); Wu et al. (2022) unveil the Hessian-variance alignment property of SGD noise, shedding light on the role of SGD in escaping from sharper minima and locating flatter minima.

## 3 PRELIMINARY

In this section, we present the notations and definitions utilized in our theoretical analysis. *We remark that our experimental settings are more general than the counterparts in the theoretical analysis.*

### 3.1 NOTATIONS

We set a special vector $(1, 1, 1, \ldots, 1)^\intercal$ by $\mathbf{1} := (1, 1, 1, \ldots, 1)^\intercal$ whose dimension varies. We set $n$ for the number of input samples and $m$ for the width of the NN. We let $[n] = \{1, 2, \ldots, n\}$. We denote $\otimes$ as the Kronecker tensor product, and $\langle \cdot, \cdot \rangle$ for standard inner product between two vectors. We denote vector $L^2$ norm as $\|\cdot\|_2$, vector or function $L_\infty$ norm as $\|\cdot\|_\infty$. We also denote $\mathrm{Tr}(\cdot)$ as the trace of a square matrix, $\boldsymbol{I}_d$ as the identity matrix of size $d \times d$, and $\|\cdot\|_\mathrm{F}$ signifies the Frobenius norm of a matrix. Finally, we denote the set of continuous functions $f(\cdot) : \mathbb{R}^D \to \mathbb{R}$ possessing continuous derivatives of order up to and including $r$ by $\mathcal{C}^r(\mathbb{R}^D)$, the space of bounded measurable functions by $\mathcal{B}_b(\mathbb{R}^D)$, and the space of bounded continuous functions by $\mathcal{C}_b(\mathbb{R}^D)$.

### 3.2 TWO-LAYER NEURAL NETWORKS AND LOSS FUNCTION

We consider the empirical risk minimization problem given by the quadratic loss:

$$\min_{\boldsymbol{\theta}} R_{\mathcal{S}}(\boldsymbol{\theta}) = \frac{1}{2n} \sum_{i=1}^{n} \left( f_{\boldsymbol{\theta}}(\boldsymbol{x}_i) - y_i \right)^2, \tag{1}$$

where $\mathcal{S} := \{(\boldsymbol{x}_i, y_i)\}_{i=1}^n$ is the training sample, $f_{\boldsymbol{\theta}}(\boldsymbol{x})$ is the prediction function, $\boldsymbol{\theta}$ are the parameters, and their dependence is modeled by a two-layer NN with $m$ hidden neurons

$$f_{\boldsymbol{\theta}}(\boldsymbol{x}) := \sum_{r=1}^m a_r \sigma(\boldsymbol{w}_r^\intercal \boldsymbol{x}), \tag{2}$$

where $\boldsymbol{x} \in \mathbb{R}^d$, $\boldsymbol{\theta} = \text{vec}(\boldsymbol{\theta}_a, \boldsymbol{\theta}_{\boldsymbol{w}}) \in \mathbb{R}^D$, where $D := m(d+1)$ throughout this paper. We remark that $\boldsymbol{\theta}$ is the set of parameters with $\boldsymbol{\theta}_a = \text{vec}(\{a_r\}_{r=1}^m)$, $\boldsymbol{\theta}_{\boldsymbol{w}} = \text{vec}(\{\boldsymbol{w}_r\}_{r=1}^m)$, and we impose hereafter that the activation function $\sigma(\cdot)$ to be continuously differentiable up to order 6, i.e., $\sigma \in \mathcal{C}^6(\mathbb{R})$. More precisely, $\boldsymbol{\theta} = \text{vec}(\{\boldsymbol{q}_r\}_{r=1}^m)$, where for each $r \in [m]$, $\boldsymbol{q}_r := (a_r, \boldsymbol{w}_r^\intercal)^\intercal$, and the bias term $b_r$ can be incorporated by expanding $\boldsymbol{x}$ and $\boldsymbol{w}_r$ to $(\boldsymbol{x}^\intercal, 1)^\intercal$ and $(\boldsymbol{w}_r^\intercal, b_r)^\intercal$.

### 3.3 DROPOUT

For a fixed learning rate $\eta > 0$, then at the $N$-th iteration where $t_N := N\eta$, a scaling vector $\boldsymbol{\delta}_N \in \mathbb{R}^m$ is sampled with independent random coordinates: For each $k \in [m]$,

$$(\boldsymbol{\delta}_N)_k = \begin{cases} \frac{1}{p} & \text{with probability } p, \\ 0 & \text{with probability } 1-p, \end{cases} \tag{3}$$

and we observe that $\{\boldsymbol{\delta}_N\}_{N \geq 1}$ is an i.i.d. Bernoulli sequence with $\mathbb{E}\boldsymbol{\delta}_N = \mathbf{1}$. With slight abuse of notations, the $\sigma$-fields $\mathcal{F}_N := \{\sigma(\boldsymbol{\delta}_1, \boldsymbol{\delta}_2, \cdots \boldsymbol{\delta}_N)\}$ forms a natural filtration. We then apply dropout to the two-layer NNs by computing

$$f_{\boldsymbol{\theta}}(\boldsymbol{x}; \boldsymbol{\delta}) := \sum_{r=1}^m (\boldsymbol{\delta})_r a_r \sigma(\boldsymbol{w}_r^\intercal \boldsymbol{x}), \tag{4}$$

and we denote the empirical risk associated with dropout by

$$R_{\mathcal{S}}^{\text{drop}}(\boldsymbol{\theta}; \boldsymbol{\delta}) := \frac{1}{2n} \sum_{i=1}^n (f_{\boldsymbol{\theta}}(\boldsymbol{x}_i; \boldsymbol{\delta}) - y_i)^2 = \frac{1}{2n} \sum_{i=1}^n \left( \sum_{r=1}^m (\boldsymbol{\delta})_r a_r \sigma(\boldsymbol{w}_r^\intercal \boldsymbol{x}_i) - y_i \right)^2. \tag{5}$$

We remark that the parameters at the $N$-th step are updated as follows:

$$\boldsymbol{\theta}_N = \boldsymbol{\theta}_{N-1} - \eta \nabla_{\boldsymbol{\theta}} R_{\mathcal{S}}^{\text{drop}}(\boldsymbol{\theta}_{N-1}; \boldsymbol{\delta}_N), \tag{6}$$

where $\boldsymbol{\theta}_0 := \boldsymbol{\theta}(0)$. Finally, we denote hereafter that for all $i \in [n]$,

$$e_i^N := e_i(\boldsymbol{\theta}_{N-1}; \boldsymbol{\delta}_N) := f_{\boldsymbol{\theta}_{N-1}}(\boldsymbol{x}_i; \boldsymbol{\delta}_N) - y_i.$$

## 4 STOCHASTIC MODIFIED EQUATIONS FOR DROPOUT

In this section, we approximate the iterative process of dropout (6) in the weak sense (Definition 1).

### 4.1 MODIFIED LOSS

As the dropout iteration (6) reads

$$\boldsymbol{\theta}_N - \boldsymbol{\theta}_{N-1} = -\eta \nabla_{\boldsymbol{\theta}} R_{\mathcal{S}}^{\text{drop}}(\boldsymbol{\theta}_{N-1}; \boldsymbol{\delta}_N) = -\frac{\eta}{n} \sum_{i=1}^n e_i^N \nabla_{\boldsymbol{\theta}} e_i^N.$$

Since $\boldsymbol{\theta} = \text{vec}(\{\boldsymbol{q}_r\}_{r=1}^m) = \text{vec}(\{(a_r, \boldsymbol{w}_r)\}_{r=1}^m)$, then given $\boldsymbol{\theta}_{N-1}$, for each $k \in [m]$, the expectation of the increment restricted to $\boldsymbol{q}_k$ reads

$$\mathbb{E}_{\boldsymbol{\theta}_{N-1}} \left[ \sum_{i=1}^n e_i^N \nabla_{\boldsymbol{q}_k} e_i^N \right] = \mathbb{E}_{\boldsymbol{\theta}_{N-1}} \left[ \sum_{i=1}^n e_i^N (\boldsymbol{\delta}_N)_k \nabla_{\boldsymbol{q}_k} (a_k \sigma(\boldsymbol{w}_k^\intercal \boldsymbol{x}_i)) \right]$$

$$= \sum_{i=1}^n e_i \nabla_{\boldsymbol{q}_k} (a_k \sigma(\boldsymbol{w}_k^\intercal \boldsymbol{x}_i)) + \frac{1-p}{p} \sum_{i=1}^n a_k \sigma(\boldsymbol{w}_k^\intercal \boldsymbol{x}_i) \nabla_{\boldsymbol{q}_k} (a_k \sigma(\boldsymbol{w}_k^\intercal \boldsymbol{x}_i)),$$

where we denote for simplicity that $e_i := e_i(\boldsymbol{\theta}) := \sum_{r=1}^m a_r \sigma(\boldsymbol{w}_r^\intercal \boldsymbol{x}_i) - y_i$. Compared with $e_i^N$, $e_i$ does not depend on the random variable $\boldsymbol{\delta}_N$. Hence, as we define the *modified loss* $L_{\mathcal{S}}(\cdot) : \mathbb{R}^D \to \mathbb{R}$ for dropout:

$$L_{\mathcal{S}}(\boldsymbol{\theta}) := \frac{1}{2n} \sum_{i=1}^n e_i^2 + \frac{1-p}{2np} \sum_{i=1}^n \sum_{r=1}^m a_r^2 \sigma(\boldsymbol{w}_r^\intercal \boldsymbol{x}_i)^2. \tag{7}$$

We observe that for each $k \in [m]$, the gradient of $L_{\mathcal{S}}$ restricted to $\boldsymbol{q}_k$ reads

$$\nabla_{\boldsymbol{q}_k} L_{\mathcal{S}}(\boldsymbol{\theta}) = \frac{1}{n} \sum_{i=1}^n e_i \nabla_{\boldsymbol{q}_k} \left( a_k \sigma(\boldsymbol{w}_k^\intercal \boldsymbol{x}_i) \right) + \frac{1-p}{np} \sum_{i=1}^n a_k \sigma(\boldsymbol{w}_k^\intercal \boldsymbol{x}_i) \nabla_{\boldsymbol{q}_k} \left( a_k \sigma(\boldsymbol{w}_k^\intercal \boldsymbol{x}_i) \right),$$

which indicates that given $\boldsymbol{\theta}_{N-1}$, the conditional expectation of the increment of the parameter at the $N$-th step reads

$$\boldsymbol{\theta}_N - \boldsymbol{\theta}_{N-1} = -\eta \mathbb{E}_{\boldsymbol{\theta}_{N-1}} \left[ \nabla_{\boldsymbol{\theta}} R_{\mathcal{S}}^{\mathrm{drop}} \left( \boldsymbol{\theta}_{N-1}; \boldsymbol{\delta}_N \right) \right] = -\eta \nabla_{\boldsymbol{\theta}} L_{\mathcal{S}}(\boldsymbol{\theta}) \big|_{\boldsymbol{\theta} = \boldsymbol{\theta}_{N-1}}.$$

Then in the sense of expectations, $\{\boldsymbol{\theta}_N\}_{N \geq 0}$ follows close to the gradient descent (GD) trajectory of $L_{\mathcal{S}}(\boldsymbol{\theta})$ with fixed learning rate $\eta$. In the above procedure, we focus on the drift term of dropout and disregard its fluctuation term as we merely consider the first conditional moment of the parameter increment. Please refer to Appendix G.1 for the detailed derivation of $L_{\mathcal{S}}$.

## 4.2 STOCHASTIC MODIFIED EQUATIONS

In pursuit of a more comprehensive understanding of the dynamics of dropout, we integrate the fluctuation term of dropout into our analysis. Firstly, as shown above, we observe that given $\boldsymbol{\theta}_{N-1}$,

$$\boldsymbol{\theta}_N - \boldsymbol{\theta}_{N-1} = -\eta \nabla_{\boldsymbol{\theta}} L_{\mathcal{S}}(\boldsymbol{\theta}) \big|_{\boldsymbol{\theta} = \boldsymbol{\theta}_{N-1}} + \sqrt{\eta} \boldsymbol{V}(\boldsymbol{\theta}_{N-1}), \tag{8}$$

where $L_{\mathcal{S}}(\cdot) : \mathbb{R}^D \to \mathbb{R}$ is the modified loss defined in (7), and $\boldsymbol{V}(\cdot) : \mathbb{R}^D \to \mathbb{R}^D$ represents the fluctuation term of dropout. When given $\boldsymbol{\theta}_{N-1}$, $\boldsymbol{V}(\boldsymbol{\theta}_{N-1})$ has mean $\mathbf{0}$ and covariance $\eta \boldsymbol{\Sigma}(\boldsymbol{\theta}_{N-1})$, where $\boldsymbol{\Sigma}(\cdot) : \mathbb{R}^D \to \mathbb{R}^{D \times D}$, whose expression is deferred to Section 5.1.

Consider the stochastic differential equation (SDE),

$$\mathrm{d}\boldsymbol{\Theta}_t = \boldsymbol{b}(\boldsymbol{\Theta}_t) \, \mathrm{d}t + \boldsymbol{\sigma}(\boldsymbol{\Theta}_t) \, \mathrm{d}\boldsymbol{W}_t, \quad \boldsymbol{\Theta}_0 = \boldsymbol{\Theta}(0), \tag{9}$$

where $\boldsymbol{W}_t$ is a standard $D$-dimensional Brownian motion. Its Euler–Maruyama discretization with step size $\eta > 0$ at the $N$-th step reads

$$\boldsymbol{\Theta}_{\eta N} = \boldsymbol{\Theta}_{\eta(N-1)} + \eta \boldsymbol{b}\left( \boldsymbol{\Theta}_{\eta(N-1)} \right) + \sqrt{\eta} \boldsymbol{\sigma}\left( \boldsymbol{\Theta}_{\eta(N-1)} \right) \boldsymbol{Z}_N,$$

where $\boldsymbol{Z}_N \sim \mathcal{N}(\mathbf{0}, \boldsymbol{I}_D)$ and $\boldsymbol{\Theta}_0 = \boldsymbol{\Theta}(0)$. Thus, if we set

$$\begin{aligned} \boldsymbol{b}(\boldsymbol{\Theta}) &:= -\nabla_{\boldsymbol{\Theta}} L_{\mathcal{S}}(\boldsymbol{\Theta}), \\ \boldsymbol{\sigma}(\boldsymbol{\Theta}) &:= \sqrt{\eta} \left( \boldsymbol{\Sigma}(\boldsymbol{\Theta}) \right)^{\frac{1}{2}}, \\ \boldsymbol{\Theta}_0 &:= \boldsymbol{\theta}_0, \end{aligned} \tag{10}$$

then we would expect (9) to be a "good" approximation of (8) with time identification $t = \eta N$. Based on the previous work (Li et al., 2017), we use approximations in the *weak* sense (Kloeden and Platen, 2011, Section 9.7) since the path of dropout and the corresponding SDE are driven by noises sampled in different spaces.

To compare different discrete time approximations, we need to take the rate of weak convergence into consideration, and we also need to choose an appropriate class of functions as the space of test functions. We introduce the following set of smooth functions:

$$\mathcal{C}_b^M\left(\mathbb{R}^D\right) = \left\{ f \in \mathcal{C}^M\left(\mathbb{R}^D\right) \,\middle|\, \|f\|_{\mathcal{C}^M} := \sum_{|\beta| \leq M} \left\| \mathrm{D}^\beta f \right\|_\infty < \infty \right\}, \tag{11}$$

where $\mathrm{D}$ is the usual differential operator. We remark that $\mathcal{C}_b^M(\mathbb{R}^D)$ is a subset of $\mathcal{G}(\mathbb{R}^D)$, the class of functions with polynomial growth, which is chosen to be the space of test functions in previous works (Li et al., 2017; Kloeden and Platen, 2011). To ensure validity of our analysis, we assume that

**Assumption 1.** *There exists $T^* > 0$, such that for any $t \in [0, T^*]$, there exists a unique $t$-continuous solution $\mathbf{\Theta}_t$ to SDE (9). Furthermore, for each $l \in [3]$, there exists $C(T^*, \mathbf{\Theta}_0) > 0$, such that*

$$\sup_{0 \leq s \leq T^*} \mathbb{E}\left(\|\mathbf{\Theta}_s(\cdot)\|_2^{2l}\right) \leq C(T^*, \mathbf{\Theta}_0). \tag{12}$$

*Moreover, for the dropout iterations (6), let $0 < \eta < 1$, $T > 0$ and set $N_{T,\eta} := \lfloor \frac{T}{\eta} \rfloor$. There exists $\eta_0 > 0$, such that given any learning rate $\eta \leq \eta_0$, then for all $N \in [0 : N_{T^*,\eta}]$ and for each $l \in [3]$, there exists $C(T^*, \boldsymbol{\theta}_0, \eta_0) > 0$, such that*

$$\sup_{0 \leq N \leq [N_{T^*,\eta}]} \mathbb{E}\left(\|\boldsymbol{\theta}_N\|_2^{2l}\right) \leq C(T^*, \boldsymbol{\theta}_0, \eta_0). \tag{13}$$

We remark that local existence of the solution to SDE and estimates of all $2l$-moments of the solution to SDE can be guaranteed for smooth coefficients and sufficiently small time $T^* > 0$. Moreover, as the constants $C(T^*, \mathbf{\Theta}_0)$ and $C(T^*, \boldsymbol{\theta}_0, \eta_0)$ are exponential in time, the $2l$-moments of the solution might blow up for large enough $T^*$, which is unavoidable since we are unable to impose the uniform Lipschitz condition on $\nabla \mathcal{L}_S$ and $\mathbf{\Sigma}$. However, our empirical findings suggest that the SME still possess the desired approximation ability to dropout even for a large learning rate, as shown in Fig. 1 (a). We also remark that if $\mathcal{G}(\mathbb{R}^D)$ is chosen to be the test functions in Li et al. (2019), then similar relations to (12) and (13) shall be imposed, except that in our cases, we only require the second, fourth and sixth moments to be uniformly bounded.

**Definition 1.** *The SDE (9) is an order $\alpha$ weak approximation to the dropout (6), if for every $g \in \mathcal{C}_b^M\left(\mathbb{R}^D\right)$, there exists $C > 0$ and $\eta_0 > 0$, such that given any $\eta \leq \eta_0$ and $T \leq T^*$, then for all $N \in [N_{T,\eta}]$,*

$$|\mathbb{E}g(\mathbf{\Theta}_{\eta N}) - \mathbb{E}g(\boldsymbol{\theta}_N)| \leq C(T^*, g, \eta_0)\eta^\alpha. \tag{14}$$

We now state formally our approximation results.

**Theorem 1.** *Fix time $T \leq T^*$ and learning rate $\eta > 0$. If $\sigma \in \mathcal{C}^6(\mathbb{R})$, then for all $t \in [0, T]$, the stochastic processes $\mathbf{\Theta}_t$ satisfying*

$$\mathrm{d}\mathbf{\Theta}_t = \boldsymbol{b}_1\left(\mathbf{\Theta}_t\right)\mathrm{d}t + \boldsymbol{\sigma}_1\left(\mathbf{\Theta}_t\right)\mathrm{d}\boldsymbol{W}_t, \tag{15}$$

*is an order-1 approximation of dropout (6), where*

$$\boldsymbol{b}_1(\mathbf{\Theta}) = -\nabla_{\mathbf{\Theta}} L_S(\mathbf{\Theta}),$$

$$\boldsymbol{\sigma}_1(\mathbf{\Theta}) = \sqrt{\eta}\left(\mathbf{\Sigma}\left(\mathbf{\Theta}\right)\right)^{\frac{1}{2}},$$

*and the expression of $L_S(\cdot)$ is located in (7), and the expression of $\mathbf{\Sigma}(\cdot)$ can be found in Appendix J. Moreover, if $\sigma \in \mathcal{C}^6(\mathbb{R})$, then for all $t \in [0, T]$, the stochastic processes $\mathbf{\Theta}_t$ satisfying*

$$\mathrm{d}\mathbf{\Theta}_t = \boldsymbol{b}_2\left(\mathbf{\Theta}_t\right)\mathrm{d}t + \boldsymbol{\sigma}_2\left(\mathbf{\Theta}_t\right)\mathrm{d}\boldsymbol{W}_t, \tag{16}$$

*is an order-2 approximation of dropout (6), where*

$$\boldsymbol{b}_2(\mathbf{\Theta}) = -\nabla_{\mathbf{\Theta}}\left(L_S(\mathbf{\Theta}) + \frac{\eta}{4}\|\nabla_{\mathbf{\Theta}} L_S(\mathbf{\Theta})\|_2^2\right),$$

$$\boldsymbol{\sigma}_2(\mathbf{\Theta}) = \sqrt{\eta}\left(\mathbf{\Sigma}\left(\mathbf{\Theta}\right)\right)^{\frac{1}{2}}.$$

It is noteworthy that our findings reproduce the explicit regularization effect attributed to dropout (Wei et al., 2020; Zhang and Xu, 2022). This regularization effect modifies the expected training objective from the empirical risk $R_S(\boldsymbol{\theta})$ to $L_S(\boldsymbol{\theta})$, and it stems from the inherent stochastic nature of dropout. Unlike SGD, where the noise arises from the stochasticity involved in the selection of training samples, dropout introduces noise by means of the stochastic removal of parameters.

### 4.3 NUMERICAL SIMULATION OF STOCHASTIC MODIFIED EQUATIONS

In this subsection, we conduct an empirical validation of the effectiveness of SMEs for dropout. This validation is conducted through an exploration of the resemblance between the numerical simulation of the SME and the real-time training process of dropout. For the numerical simulation of the SME, unless otherwise specified, we employ the Euler-Maruyama method to approximate its dynamic

evolution by the order-1 approximation. It is worth noting that the noise term $\boldsymbol{\sigma}(\boldsymbol{\theta})$ in Equ. (10) involves the computation of the square root of the covariance matrix $\boldsymbol{\Sigma}(\boldsymbol{\theta})$. Consequently, the size of the covariance matrix significantly affects the speed and accuracy of the numerical simulation process. To mitigate the computational demands associated with the covariance matrix, we resize the MNIST data to $7 \times 7$, thereby reducing the number of network parameters involved in the simulations. Additional details of the experimental setup can be found in Appendix A.

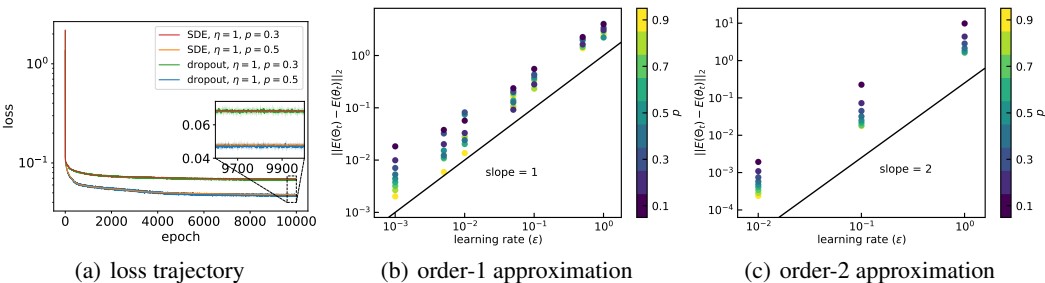

(a) loss trajectory   (b) order-1 approximation   (c) order-2 approximation

Figure 1: We train two-layer fully connected networks on MNIST. The curves and points are derived from the average results of individual trials, each of which utilized the same initialization distribution. (a) The training loss trajectory obtained by SME simulation or dropout training under four cases of different learning rates and dropout rates. The error bands, portrayed with greater transparency, are derived from the maximum and minimum loss values observed across these six random trials at each training step. (b, c) Convergence-order verification of first-order and second-order SME approximations. Each point represents the value of $\|\mathbb{E}(\boldsymbol{\Theta}_t) - \mathbb{E}(\boldsymbol{\theta}_t)\|_2$ under a given learning rate (abscissa) and dropout rate (color).

Fig. 1 illustrates the close correspondence between the dynamics of the SME and dropout throughout the training process. This similarity is examined from two perspectives: the trajectory of loss functions and the approximation order. In Fig. 2, we emphasize that although dropout introduces a noise component that modifies the loss function from $R_{\mathcal{S}}(\boldsymbol{\theta})$ to $L_{\mathcal{S}}(\boldsymbol{\theta})$, when contrasted with SMEs, the training behavior of gradient descent utilizing $L_{\mathcal{S}}(\boldsymbol{\theta})$ as the loss function is distinct from dropout. This distinction becomes apparent when large learning rates are employed in the optimization process.

To comprehensively assess the similarity between dropout and SME simulations, we first consider four distinct cases, each characterized by various dropout rates and learning rates shown in Fig. 1(a). Fig. 1(a) depicts the evolution of loss values under these distinct settings for both dropout and SME simulations. To ensure the robustness of our analysis, for each configuration, we conduct six independent trials of dropout and SME simulations, all initialized with identical distribution. The displayed curves represent the means of these six random trials. Moreover, the error band, indicated by lighter colors, covers the range between the maximum and minimum loss values obtained from the six trials. The observed alignment of loss trajectories between the SME simulation and dropout, as evident in Fig. 1(a), underscores a prominent resemblance in their respective loss trajectories.

To further evaluate the similarity of their parameters, we verify the approximation orders of different SME simulations. Figs. 1(b, c) numerically verify the approximation orders of the first-order and the second-order approximation equation in Theorem 1 respectively. Each point represents the value of $\|\mathbb{E}(\boldsymbol{\Theta}_t) - \mathbb{E}(\boldsymbol{\theta}_t)\|_2$ under a given learning rate (abscissa) and dropout rate (color). The expectation is obtained by calculating the mean of 10 independent experiments with the same initialization for both dropout and SME simulation. The logarithmic plots clearly illustrate the experimental validation of the theoretical approximation order of SME, for both order-1, shown in Fig. 1(b), and order-2, shown in Fig. 1(c). Additionally, under the same learning rate, larger values of $p$ exhibit enhanced approximation capabilities. This improvement is attributed to the reduction in noise with increasing $p$, consequently minimizing the impact of noise on the training process.

We also conduct experiments to validate the applicability of the SME approximation in complex networks and SGD settings. In the former, we simulate complex networks through numerical approximation of the drift term, while in the latter, we rely on the fact that SGD noise is unbiased. For a thorough discussion and detailed numerical results, please refer to Appendix C.

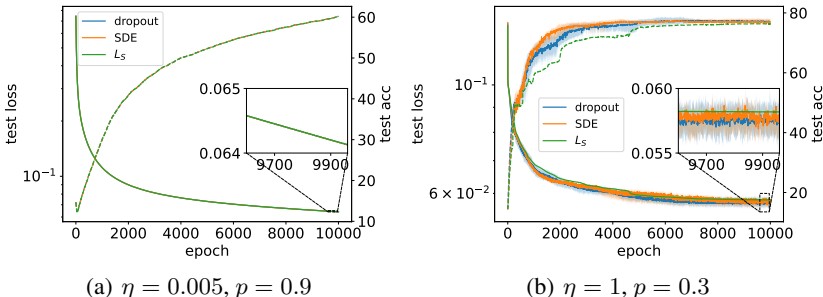

(a) $\eta = 0.005$, $p = 0.9$        (b) $\eta = 1$, $p = 0.3$

Figure 2: The test loss and test accuracy trajectory obtained by SME simulation, dropout training, and gradient descent training with loss function $L_{\mathcal{S}}(\boldsymbol{\theta})$ under two settings. The curves are derived from the average results of six individual trials, each of which utilized the same initialization distribution. The error bands, portrayed with greater transparency, are derived from the maximum and minimum loss values observed across these six random trials at each training step.

Fig. 2 depicts the test loss and test accuracy associated with $\eta = 0.005$, $p = 0.9$ and $\eta = 1$, $p = 0.3$. In Fig. 2(a), the trajectories of loss and accuracy, generated using three different training methods, exhibit a remarkable degree of concurrence. This phenomenon can be primarily attributed to the utilization of a small learning rate, where the diffusion component significantly diminishes, consequently endowing the drift term $L_{\mathcal{S}}(\boldsymbol{\theta})$ with a dominant influence. In contrast, as illustrated in Fig. 2(b), a notable divergence becomes evident in the loss and accuracy trajectories. This discrepancy arises from the impact of the diffusion term during training, particularly with the application of large learning rates. Notably, in contrast to the training behavior of gradient descent utilizing $L_{\mathcal{S}}(\boldsymbol{\theta})$ as the loss function, the trajectory generated by SME simulation exhibits a closer alignment with the trajectory of dropout.

It is noteworthy that as large learning rate and dropout rate contribute to an increased amplitude of diffusion, methods incorporating noise such as SME and dropout tend to exhibit enhanced generalization performance. As demonstrated in Figure 2(b), the test accuracy attained by $L_{\mathcal{S}}(\boldsymbol{\theta})$ consistently remains below the lower threshold of test accuracy attained by noise-inclusive methods for the major portion of the training duration. In the sequel, we delve into an in-depth exploration of the influence exerted by noise on our learning outcomes.

## 5 THE EFFECT OF DROPOUT NOISE STRUCTURE

We begin this section by examining the noise structure of dropout.

### 5.1 EXPLICIT FORM OF THE NOISE STRUCTURE OF DROPOUT

In this subsection, we present the expression for the covariance $\boldsymbol{\Sigma}(\boldsymbol{\theta})$. Once again, as $\boldsymbol{\theta} = \text{vec}(\{\boldsymbol{q}_r\}_{r=1}^{m}) = \text{vec}\left(\{(a_r, \boldsymbol{w}_r)\}_{r=1}^{m}\right)$, then as we denote covariance of $\nabla_{\boldsymbol{\theta}} R_{\mathcal{S}}^{\text{drop}}\left(\boldsymbol{\theta}_{N-1}; \boldsymbol{\delta}_N\right)$ by $\boldsymbol{\Sigma}(\boldsymbol{\theta}_{N-1})$, i.e.,

$$\boldsymbol{\Sigma}_{kr}(\boldsymbol{\theta}_{N-1}) := \text{Cov}\left(\nabla_{\boldsymbol{q}_k} R_{\mathcal{S}}^{\text{drop}}\left(\boldsymbol{\theta}_{N-1}; \boldsymbol{\delta}_N\right), \nabla_{\boldsymbol{q}_r} R_{\mathcal{S}}^{\text{drop}}\left(\boldsymbol{\theta}_{N-1}; \boldsymbol{\delta}_N\right)\right),$$

then

$$\boldsymbol{\Sigma} = \begin{bmatrix} \boldsymbol{\Sigma}_{11} & \boldsymbol{\Sigma}_{12} & \cdots & \boldsymbol{\Sigma}_{1m} \\ \boldsymbol{\Sigma}_{21} & \boldsymbol{\Sigma}_{22} & \cdots & \boldsymbol{\Sigma}_{2m} \\ \vdots & \vdots & \vdots & \vdots \\ \boldsymbol{\Sigma}_{m1} & \boldsymbol{\Sigma}_{m2} & \cdots & \boldsymbol{\Sigma}_{mm} \end{bmatrix}.$$

Such expression of $\boldsymbol{\Sigma}$ arises from the inherent decoupling properties among neurons within the two-layer neural network. Due to space limitation, we defer the detailed expression of $\boldsymbol{\Sigma}_{kr}$ to Appendix J.

## 5.2 INTUITIVE EXPLANATION FOR THE HESSIAN-VARIANCE ALIGNMENT RELATIONS

In this subsection, we endeavor to show the structural similarity between the covariance and the Hessian in terms of Hessian-variance alignment relations. Under the assumption that $\boldsymbol{\theta}$ is close to a global minimum, we intuitively derive the structural similarity between the Hessian and the covariance at the final stage of the training process as follows:

$$\boldsymbol{H}(\boldsymbol{\theta}) \approx \frac{1}{n}\sum_{i=1}^{n}\left[\nabla_{\boldsymbol{\theta}}f_{\boldsymbol{\theta}}\left(\boldsymbol{x}_i\right)\otimes\nabla_{\boldsymbol{\theta}}f_{\boldsymbol{\theta}}\left(\boldsymbol{x}_i\right) + \frac{1-p}{p}\sum_{r=1}^{m}\nabla_{\boldsymbol{q}_r}\left(a_r\sigma(\boldsymbol{w}_r^{\mathsf{T}}\boldsymbol{x}_i)\right)\otimes\nabla_{\boldsymbol{q}_r}\left(a_r\sigma(\boldsymbol{w}_r^{\mathsf{T}}\boldsymbol{x}_i)\right)\right],$$

$$\boldsymbol{\Sigma}(\boldsymbol{\theta}) \approx \frac{1}{n}\sum_{i=1}^{n}\left[l_{i,1}\nabla_{\boldsymbol{\theta}}f_{\boldsymbol{\theta}}(\boldsymbol{x}_i)\otimes\nabla_{\boldsymbol{\theta}}f_{\boldsymbol{\theta}}(\boldsymbol{x}_i) + l_{i,2}\frac{1-p}{p}\sum_{r=1}^{m}\nabla_{\boldsymbol{q}_r}\left(a_r\sigma(\boldsymbol{w}_r^{\mathsf{T}}\boldsymbol{x}_i)\right)\otimes\nabla_{\boldsymbol{q}_r}\left(a_r\sigma(\boldsymbol{w}_r^{\mathsf{T}}\boldsymbol{x}_i)\right)\right],$$
$$(17)$$

where $\boldsymbol{H}(\boldsymbol{\theta}) := \nabla_{\boldsymbol{\theta}}^2 L_{\mathcal{S}}(\boldsymbol{\theta})$, and $l_{i,1} := (e_i)^2 + \frac{1-p}{p}\sum_{r=1}^{m}a_r^2\sigma(\boldsymbol{w}_r^{\mathsf{T}}\boldsymbol{x}_i)^2$, $l_{i,2} := (e_i)^2$. A detailed derivation of (17) is provided in Appendix K. With the establishment of structural similarity through the aforementioned intuitive approximations outlined in (17), we proceed to empirically investigate the intricate relationship between the Hessian and the covariance, and details of the experimental settings can be found in Appendix A.

## 5.3 EXPERIMENTAL RESULTS ON THE HESSIAN-VARIANCE ALIGNMENT RELATIONS

Motivated by the relation (17), we empirically demonstrate the structural similarity between the Hessian and the covariance of dropout, and this demonstration serves to validate the Hessian-variance alignment relation. Based on this relation, the introduction of dropout noise has the potential to expedite the escape of the model from locating sharp minima, thereby effectively enhancing the flatness of the solution. Furthermore, in Appendix B, we also explore another relationship between the Hessian and the covariance known as the inverse variance-flatness relation (Feng and Tu, 2021), which also contributes to aiding the model in avoidance of the sharp minima during its optimization process.

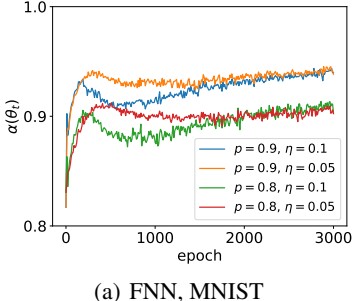

(a) FNN, MNIST

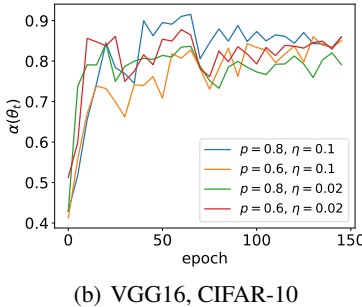

(b) VGG16, CIFAR-10

Figure 3: The cosine similarity $\alpha(\boldsymbol{\theta}_t)$ between the Hessian of the loss function and the covariance of the dropout noise at each training epoch $t$ for different choices of dropout rate and learning rate. (a) The FNN with size 784-50-50-10 is trained on the MNIST dataset using the first 10000 examples as the training dataset. The dropout layer is added after the first hidden layer. (b) The VGG16 is trained on the CFIAR-10 dataset using the full examples as the training dataset. The dropout layers are added after the first two convolutional layers of each block and the first fully-connected layer. The calculation of $\alpha(\boldsymbol{\theta}_t)$ is performed every five epochs.

To investigate the Hessian-Variance alignment relation, we study the cosine similarity quantity $\alpha(\boldsymbol{\theta}_t)$[1] between the covariance matrix $\boldsymbol{\Sigma}_t := \boldsymbol{\Sigma}(\boldsymbol{\theta}_t)$ and the Hessian matrix $\boldsymbol{H}_t := \boldsymbol{H}(\boldsymbol{\theta}_t)$ at each time step $t$. $\boldsymbol{\Sigma}_t$ is the covariance matrix of $\mathcal{D}_{\mathrm{grad}}$, a collection of gradients calculated with different dropout variables $\boldsymbol{\delta}$ sampled at the $t$th step, whose detailed definition can be found in Section B.1. On the other hand, $\boldsymbol{H}_t$ is the Hessian of the loss function evaluated at the $t$th iteration. Then the crucial

---

[1]This variable is also used in Wu et al. (2022) for studying SGD.

cosine similarity metric $\alpha(\boldsymbol{\theta}_t)$ is formally expressed as:

$$\alpha(\boldsymbol{\theta}_t) = \frac{\text{Tr}(\boldsymbol{H}_t \boldsymbol{\Sigma}_t)}{\|\boldsymbol{H}_t\|_{\text{F}} \|\boldsymbol{\Sigma}_t\|_{\text{F}}} \tag{18}$$

As depicted in Fig. 3, it is evident that throughout the training process, $\alpha(\boldsymbol{\theta}_t)$ consistently attains values surpassing 0.85 in Fig. 3(a) and 0.7 in Fig. 3(b), and these observations hold true across varying learning rates and dropout rates. It's worth noting that, based on 100 samples, the average cosine similarity between the two random matrices based on the selected parameters is only $7.8 \times 10^{-4}$. The eigenvalues of the two random matrices are derived from the eigenvalues of the Hessian matrix and covariance matrix of the selected parameters, and the corresponding eigenvectors are sampled from a normal distribution and normalized. The model parameters are derived from the final model represented by the blue line in Fig. 3(a). Consequently, the introduced noise is highly anisotropic in that it aligns well with the Hessian matrix across all directions. We acknowledged that due to computational constraints, this experiment limits the trace calculation to a subset of parameters.

## 6 CONCLUSIONS AND DISCUSSIONS

Our main contribution comprises two key aspects. First, we derive the SMEs that provide a weak approximation to the dynamics of the dropout algorithm applied to two-layer NNs. Second, we conduct an empirical inquiry that demonstrates the persistent validity of the Hessian-variance alignment relation throughout the training process of dropout. The Hessian-variance alignment relation has been established to be beneficial for the model to locate flatter minima, thus indicating that dropout acts as an implicit regularizer that enhances the generalization power possessed by the model.

**Extension of the SME framework to multi-layer networks and SGD.** While our theoretical analysis has predominantly centered around the dropout algorithm applied to two-layer neural networks and GD, it is important to note that the derivation of SMEs is not confined exclusively to two-layer neural networks, GD, or even to the dropout algorithm. For various types of neural networks, the feasibility of constructing such modified equations remains viable, provided that the stochastic algorithm iteratively updates the parameters in a recursive manner, i.e., iterations form a time-homogeneous Markov chain. Furthermore, this applicability holds as long as Taylor's theorem with the Lagrange form of the remainder remains valid for sufficiently small learning rates. It is worth acknowledging that the complexity introduced by multi-layer networks primarily arises from the presence of dropout layers within the activation functions. This introduces a high degree of non-linearity to the loss with respect to the dropout variable, rendering it challenging to explicitly calculate the drift and diffusion components of the SME. We numerically verify the SDE approximation capability of complex network structures and SGD in Appendix C.

**The effect of learning rate on dropout.** In the small learning rate regime, wherein the noise term exerts slight influence, the loss trajectories of $L_{\mathcal{S}}(\boldsymbol{\theta})$ and $R_{\mathcal{S}}^{\text{drop}}(\boldsymbol{\theta}; \boldsymbol{\delta})$ exhibit a notable degree of congruence. This observation has been affirmed through theoretical and empirical validations. However, it remains imperative to maintain the diffusion term is important if we aspire to gain deeper insights into the nature of dropout algorithms or other stochastic algorithms. As illustrated in Fig. 2(b), in the large learning rate regime, the trajectory derived from the SME simulation aligns more closely with its dropout counterpart, in stark contrast to the trajectory arising from GD training on $L_{\mathcal{S}}(\boldsymbol{\theta})$. Furthermore, SMEs consistently exhibit better generalization capability in comparison to GD. Therefore, a comprehensive analytical framework that duly accommodates both drift and diffusion terms stands as a more informative tool for the insightful analysis of dropout algorithms.

**More refined analysis of noise structures.** In addition to the Hessian-Variance alignment relation, the structural similarity between the Hessian and the covariance engenders yet another intriguing relationship known as the inverse variance-flatness relation (Feng and Tu, 2021). Different from the Hessian-Variance alignment relation, it focuses more on the similarity of the two feature directions. In Appendix B, an investigation has been conducted to examine the correlation between the noise structure introduced by dropout and the nature of the loss landscape. This relationship also plays a pivotal role in assisting the model to steer clear of sharp minima. The high similarity of the eigenvectors of two matrices is a natural extension of the inverse variance-flatness relation, please refer to Appendix B for detailed validation results. In Appendix D, we compare the effect of noise on the model in three training strategies, dropout, SGD, and parametric noise injection (Orvieto et al., 2023), which all appear to be helpful for flatness.

## ACKNOWLEDGMENTS

This work is sponsored by the National Key R&D Program of China Grant No. 2022YFA1008200 (Z. X., T. L.), the National Natural Science Foundation of China Grant No. 92270001 (Z. X.), 12371511 (Z. X.), 12101401 (T. L.), Shanghai Municipal of Science and Technology Major Project No. 2021SHZDZX0102 (Z. X., T. L.), Shanghai Municipal Science and Technology Key Project No. 22JC1401500 (T. L.), and the HPC of School of Mathematical Sciences and the Student Innovation Center, and the Siyuan-1 cluster supported by the Center for High Performance Computing at Shanghai Jiao Tong University.

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

## A    EXPERIMENTAL SETUPS

For Fig. 1, Fig. 2, Fig. 7(b), we use the FNN with size 49-40-10 for the MNIST classification task, where the input data is resized to $7 \times 7$ to reduce the amount of calculation of the root of the parameter covariance matrix. For Fig. 1, Fig. 2, we train the network using GD with the first 1000 images as the training set. We add a dropout layer behind the hidden layer. The dropout rate and learning rate are specified and unchanged in each experiment. For Fig. 7(b), we train the network using SGD with the whole training set, the batch size is 32. We add a dropout layer behind the hidden layer. The learning rate is 0.1 and $p = 0.8$. For the accuracy of SDE simulation, all parameters are used to calculate the Hessian matrix and covariance matrix.

For Fig. 3(a), Fig. 4, Fig. 6, Fig. 8(a), we use the FNN with size 784-50-50-10 for the MNIST classification task. We add a dropout layer behind the second layer. The dropout rate and learning rate are specified and unchanged in each experiment. We only consider the parameter matrix corresponding to the weight and the bias of the fully-connected layer between two hidden layers. Therefore, for experiments in Fig. 3(a), $D = 2500$. For Fig. 3(a), Fig. 4, Fig. 6, we train the network using GD with the first 10000 images as the training set. For Fig. 8(a), we train the network using SGD with the whole training set with batch size 32.

For Fig. 3(b), Fig. 8(b), we use the VGG16 for the CIFAR10 classification task. We train the network using SGD with batch size 128 for Fig. 3(b), and 32 for Fig. 8(b). We add the dropout layer after the first two convolutional layers of each block and the first fully-connected layer. The dropout rate and learning rate are specified and unchanged in each experiment. We only consider the parameter matrix corresponding to the weight and the bias of the first convolutional layer. Therefore, $D = 1728$ for the two experiments. Due to the computational cost of the Hessian matrix and the covariance matrix, we calculate cosine similarity every five epochs.

For Fig. 5(a, c, e, g), we add dropout layers after the convolutional layers, and for each dropout layer, $p = 0.8$. We only consider the parameter matrix corresponding to the weight of the first convolutional layer of the first block of the ResNet-20. Models are trained using full-batch GD on the CIFAR100 classification task for 1200 epochs. The learning rate is initialized at 0.01. Since the Hessian calculation of ResNet takes much time, we only perform it at a specific dropout rate and learning rate.

For Fig. 5(b, d, f, h), we use transformer Vaswani et al. (2017) with $d_{\mathrm{model}} = 50, d_k = d_v = 20, d_{\mathrm{ff}} = 256, h = 4, N = 3$, the meaning of the parameters is consistent with the original paper. We only consider the parameter matrix corresponding to the weight of the fully-connected layer whose output is queried in the Multi-Head Attention layer of the first block of the decoder. We apply dropout to the output of each sub-layer before it is added to the sub-layer input and normalized. In addition, we apply dropout to the sums of the embeddings and the positional encodings in both the encoder and decoder stacks. For each dropout layer, $p = 0.9$. For the English-German translation problem, we use the cross-entropy loss with label smoothing trained by full-batch Adam based on the Multi30k dataset. The learning rate strategy is the same as that in Vaswani et al. (2017). The warm-up step is 4000 epochs, the training step is 10000 epochs. We only use the first 2048 examples for training to compromise with the computational burden.

For Fig. 7(a), we use the FNN with size 49-40-40-40-10 for the MNIST classification task, where the input data is resized to $7 \times 7$ to reduce the amount of calculation of the root of the parameter covariance matrix. We train the network using GD with the first 1000 images as the training set. We add a dropout layer behind each hidden layer. The learning rate is 0.1 and $p = 0.5$. For the accuracy of SDE simulation, all parameters are used to calculate the Hessian matrix and covariance matrix. The drift term is simulated by sampling the random variable 2000 times.

For Fig. 9(a), we use the ReLU FNN with the width of 1000 to fit the target function as follows,

$$f(x) = \frac{1}{2}\sigma(-x - \frac{1}{3}) + \frac{1}{2}\sigma(x - \frac{1}{3}),$$

where $\sigma(x) = \mathrm{ReLU}(x)$. The learning rate is $1 \times 10^{-3}$. For SGD, the batch size is 1. For dropout, the dropout layer is added after the hidden layer with $p = 0.8$. For parameter noise injection, we use the layer noise with the noise standard deviation $\sigma = 0.001$. We initialize the parameters in the linear regime, $\boldsymbol{\theta} \sim N\left(0, \frac{1}{m^{0.2}}\right)$, where $m = 1000$ is the width of the hidden layer.

For Fig. 9(b), Fig. 9(c), we use the ReLU FNN with the size of 784-500-500-500-10 to classify the MNIST dataset, and the learning rate is 0.01. The batch size is 128. For dropout, the dropout layer is added after the hidden layers with $p = 0.8$. For parameter noise injection, we use the layer noise with the noise standard deviation $\sigma = 0.03$.

# B    EXTENDED EXPERIMENTS ON NOISE STRUCTURES.

## B.1    RANDOM DATA COLLECTION METHODS

We first introduce two types of dynamical datasets collected during dropout training to study the noise structure of dropout. These datasets are different from the training sample $\mathcal{S}$.

**Random trajectory data.** The training process of NNs usually consists of two phases: the fast convergence phase and the exploration phase (Shwartz-Ziv and Tishby, 2017). In the exploration phase, the network is often considered to be near a minimum, and the movement of parameters is largely affected by the noise structure. Based on the previous work (Feng and Tu, 2021), we collect parameter sets $\mathcal{D}_{\mathrm{para}} := \{\boldsymbol{\theta}_i\}_{i=1}^N$ from $N$ consecutive training steps in the exploration phase, where $\boldsymbol{\theta}_i$ is the network parameter set at $i$-th sample step. This sampling method requires a large number of training steps, so model parameters often have large fluctuations during the sampling process. To improve the sampling accuracy, we propose another type of random data to characterize the noise structure of dropout as follows.

**Random gradient data.** We train the network until the loss is near zero and then we freeze the training process, then we sample $N$ realizations of the dropout variable to get the random gradient dataset, i.e., $\mathcal{D}_{\mathrm{grad}} := \{\boldsymbol{g}_i\}_{i=1}^N$. The $i$-th sample point $\boldsymbol{g}_i$ is obtained as follows: i) Firstly, we generate a realization of the dropout variable $\boldsymbol{\delta}_i$ under a given dropout rate; ii) Then, we compute the gradient of the loss function with respect to the parameters, denoted by $\boldsymbol{g}_i(\cdot) := \nabla R_{\mathcal{S}}^{\mathrm{drop}}(\cdot; \boldsymbol{\delta}_i)$. Each element in $\mathcal{D}_{\mathrm{grad}}$ represents an evolution direction of network parameters, determined by the dropout variable. Therefore, studying the structure of $\mathcal{D}_{\mathrm{grad}}$ can help us understand how the dropout noise exerts an impact throughout the training process.

## B.2    INVERSE VARIANCE-FLATNESS RELATION

The alignment relation studied above also implies the inverse variance-flatness relation, i.e., the noise variance is large along the sharp direction of the loss landscape, and small along the flat direction. In this subsection, we verify this relation by two sets of experiments. Firstly, we present two different approaches to characterize the flatness of loss landscape and the covariance of noise from the random trajectory data $\mathcal{D}_{\mathrm{para}}$ and random gradient data $\mathcal{D}_{\mathrm{grad}}$, then we numerically demonstrate the inverse variance-flatness relation. For convenience, $\mathcal{D}$ refers to either the dataset $\mathcal{D}_{\mathrm{para}}$ or the dataset $\mathcal{D}_{\mathrm{para}}$ depending on its context, so is the case for their corresponding covariance $\boldsymbol{\Sigma}$ and Hessian $\boldsymbol{H}$. We then proceed to the definitions of **noise variance** and **interval flatness.**

**Definition 2** (**noise variance**). *For dataset $\mathcal{D}$ and its covariance $\boldsymbol{\Sigma}$, we denote $\lambda_i(\boldsymbol{\Sigma})$ as the $i$th eigenvalue of $\boldsymbol{\Sigma}$ and its corresponding eigen direction as $\boldsymbol{v}_i(\boldsymbol{\Sigma})$. Then we term $\lambda_i(\boldsymbol{\Sigma})$ the noise variance of $\mathcal{D}$ at the eigen direction $\boldsymbol{v}_i(\boldsymbol{\Sigma})$.*

The interval flatness below characterizes the flatness of the landscape around a local minimum.

**Definition 3** (**interval flatness**[2] ). *For a a local minimum $\boldsymbol{\theta}_0^*$, the loss function profile $R_{\boldsymbol{v}}$ along direction $\boldsymbol{v}$ reads:*

$$R_{\boldsymbol{v}}(\gamma) \equiv R_{\mathcal{S}}(\boldsymbol{\theta}_0^* + \gamma \boldsymbol{v}),$$

*where $\gamma$ represents the distance moved in the $\boldsymbol{v}$ direction. The interval flatness $F_{\boldsymbol{v}}$ is then defined as the width of the region within which $R_{\boldsymbol{v}}(\gamma) \leq 2R_{\boldsymbol{v}}(0)$. We determine $F_{\boldsymbol{v}}$ by finding two closest points $\theta_{\boldsymbol{v}}^l < 0$ and $\theta_{\boldsymbol{v}}^r > 0$ on each side of the minimum that satisfy $R_{\boldsymbol{v}}(\theta_{\boldsymbol{v}}^l) = R_{\boldsymbol{v}}(\theta_{\boldsymbol{v}}^r) = 2R_{\boldsymbol{v}}(0)$. The interval flatness is defined as:*

$$F_{\boldsymbol{v}} \equiv \theta_{\boldsymbol{v}}^r - \theta_{\boldsymbol{v}}^l. \tag{19}$$

**Remark.** *The experiments show that the result is not sensitive to the selection of the pre-factor 2. A larger value of $F_{\boldsymbol{v}}$ means a flatter landscape in the direction $\boldsymbol{v}$.*

We use PCA to study the weight variations when the training accuracy is nearly $100\%$. The networks are trained with full-batch GD for different learning rates and dropout rates under the same random seed. When the loss is small enough, we sample the parameters or gradients of parameters $N$ times ($N = 3000$ for this experiment) and study the relationship between $\{\lambda_i(\boldsymbol{\Sigma})\}_{i=1}^N$ and $\{F_{\boldsymbol{v}_i(\boldsymbol{\Sigma})}\}_{i=1}^N$ for both weight dataset $\mathcal{D}_{\mathrm{para}}$ and gradient dataset $\mathcal{D}_{\mathrm{grad}}$.

---

[2]This definition is also used in Feng and Tu (2021)

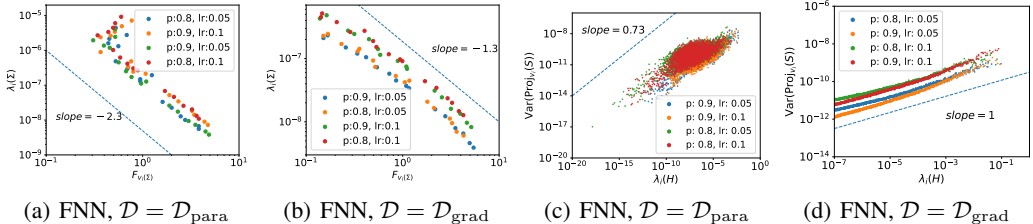

(a) FNN, $\mathcal{D} = \mathcal{D}_{\text{para}}$     (b) FNN, $\mathcal{D} = \mathcal{D}_{\text{grad}}$     (c) FNN, $\mathcal{D} = \mathcal{D}_{\text{para}}$     (d) FNN, $\mathcal{D} = \mathcal{D}_{\text{grad}}$

Figure 4: (a, b) The inverse relation between the variance $\{\lambda_i(\boldsymbol{\Sigma})\}_{i=1}^N$ and the interval flatness $\{F_{\boldsymbol{v}_i(\boldsymbol{\Sigma})}\}_{i=1}^N$ for different choices of $p$ and learning rate $lr$ with different network structures. The PCA is done for different datasets $\mathcal{D}$ sampled from parameters for the top line and sampled from gradients of parameters for the bottom line. The dashed lines give the approximate slope of the scatter. (c, d) The relation between the variance $\{\text{Var}(\text{Proj}_{\boldsymbol{v}_i(\boldsymbol{H})}(\mathcal{D}))\}_{i=1}^N$ and the eigenvalue $\{\lambda_i(\boldsymbol{H})\}_{i=1}^N$ for different choices of $p$ and learning rate $lr$ with different network structures. The projection is done for different datasets $\mathcal{D}$ sampled from parameters for the top line and sampled from gradients of parameters for the bottom line. The dashed lines give the approximate slope of the scatter.

For different learning rates and dropout rates, Fig. 4(a, b) reveal an inverse relationship between the interval flatness of the loss landscape denoted as $\{F_{\boldsymbol{v}_i(\boldsymbol{\Sigma})}\}_{i=1}^N$, and the noise variance represented by the PCA spectrum $\{\lambda_i(\boldsymbol{\Sigma})\}_{i=1}^N$. Notably, a power-law relationship can be established between $\{F_{\boldsymbol{v}_i(\boldsymbol{\Sigma})}\}_{i=1}^N$ and $\{\lambda_i(\boldsymbol{\Sigma})\}_{i=1}^N$. Specifically, in the low flatness region, the dropout-induced noise exhibits a large variance. As the loss landscape transitions into the high flatness regime, the linear relationship between variance and flatness becomes more evident. Overall, These findings consistently demonstrate the inverse relation between variance and flatness, as exemplified in Fig. 4(a, b). Subsequently, we delve into the definitions of **Projected variance** and **Hessian flatness.**

**Definition 4 (projected variance).** *For a given direction $\boldsymbol{v} \in \mathbb{R}^D$ and dataset $\mathcal{D} = \{\boldsymbol{\theta}_i\}_{i=1}^N$, where $\boldsymbol{\theta}_i \in \mathbb{R}^D$, the inner product of $\boldsymbol{v}$ and $\boldsymbol{\theta}_i$ is denoted by $\text{Proj}_{\boldsymbol{v}}(\boldsymbol{\theta}_i) := \langle \boldsymbol{\theta}_i, \boldsymbol{v} \rangle$, then we can define the projected variance for $\mathcal{D}$ at the direction $\boldsymbol{v}$ as follows,*

$$\text{Var}(\text{Proj}_{\boldsymbol{v}}(\mathcal{D})) = \frac{\sum_{i=1}^N (\text{Proj}_{\boldsymbol{v}}(\boldsymbol{\theta}_i) - \boldsymbol{\mu})^2}{N},$$

*where $\boldsymbol{\mu}$ is the mean value of $\{\text{Proj}_{\boldsymbol{v}}(\boldsymbol{\theta}_i)\}_{i=1}^N$.*

**Definition 5 (Hessian flatness).** *For Hessian $\boldsymbol{H}$, as we denote $\lambda_i(\boldsymbol{H})$ by the $i$-th eigenvalue of $\boldsymbol{H}$ corresponding to the eigenvector $\boldsymbol{v}_i(\boldsymbol{H})$, we term $\lambda_i(\boldsymbol{H})$ the Hessian flatness along direction $\boldsymbol{v}_i(\boldsymbol{H})$.*

The eigenvalues of the Hessian evaluated at a local minimum often serve as indicators of the flatness of the loss landscape, and larger eigenvalues correspond to sharper directions. In our investigation, we analyze the interplay between the eigenvalues of Hessian $\boldsymbol{H}$ at the final stage of the training process and the projected variance of dropout at each of the corresponding eigen directions, i.e., $\lambda_i(\boldsymbol{H})$ v.s. $\{\text{Var}(\text{Proj}_{\boldsymbol{v}_i(\boldsymbol{H})}(\mathcal{D}))\}_{i=1}^N$. Specifically, we sample the parameters or gradients of parameters $N$ times ($N = 1000$ for this experiment), and examine the relationship between $\{\lambda_i(\boldsymbol{H})\}_{i=1}^N$ and $\{\text{Var}(\text{Proj}_{\boldsymbol{v}_i(\boldsymbol{H})}(\mathcal{D}))\}_{i=1}^N$ for both the parameter dataset $\mathcal{D}_{\text{para}}$ and the gradient dataset $\mathcal{D}_{\text{grad}}$.

Under various dropout rates and learning rates, Fig. 4(c, d) presents establishes a consistent power-law relationship between $\{\lambda_i(\boldsymbol{H})\}_{i=1}^N$ and $\{\text{Var}(\text{Proj}_{\boldsymbol{v}_i(\boldsymbol{H})}(\mathcal{D}))\}_{i=1}^N$, and this relationship remains robust irrespective of the choice between parameter dataset $\mathcal{D}_{\text{para}}$ or the gradient dataset $\mathcal{D}_{\text{grad}}$. The positive correlation observed between the Hessian flatness and the projection variance provides insights into the structural characteristics of the dropout-induced noise. Specifically, these characteristics have the potential to facilitate the escape from sharp minima and enhance the generalization capabilities of NNs. Additionally, Fig. 4 highlights the distinct linear structure exhibited by gradient sampling in comparison to parameter sampling, which corroborates the discussions outlined in Section B.1.

Furthermore, we verify the inverse relation between the covariance matrix and the Hessian matrix of dropout through different data collection methods and projection methods on larger network

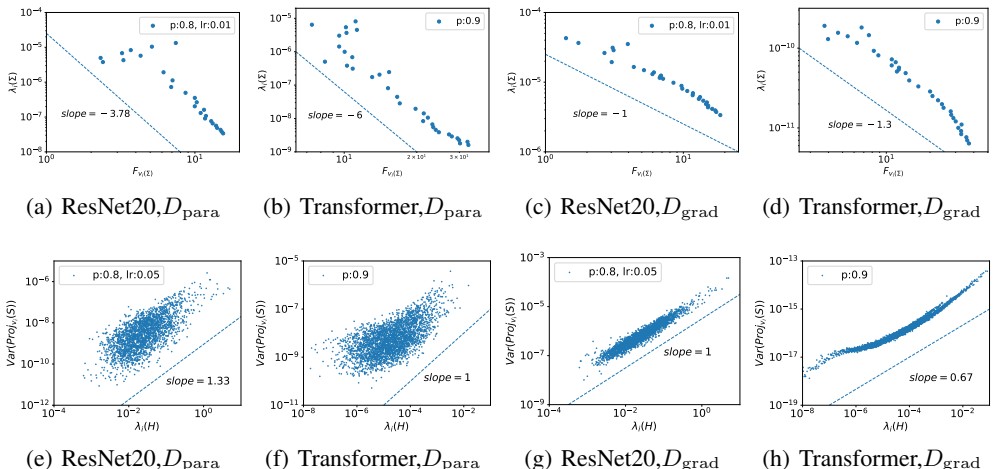

(a) ResNet20,$D_{\text{para}}$  (b) Transformer,$D_{\text{para}}$  (c) ResNet20,$D_{\text{grad}}$  (d) Transformer,$D_{\text{grad}}$

(e) ResNet20,$D_{\text{para}}$  (f) Transformer,$D_{\text{para}}$  (g) ResNet20,$D_{\text{grad}}$  (h) Transformer,$D_{\text{grad}}$

Figure 5: (a, b, c, d) The inverse relation between the variance $\{\lambda_i(\boldsymbol{\Sigma})\}_{i=1}^N$ and the interval flatness $\{F_{\boldsymbol{v}_i(\boldsymbol{\Sigma})}\}_{i=1}^N$ for different choices of $p$ and learning rate $lr$ with different network structures. The PCA is done for different datasets $D$ sampled from parameters for the top line and sampled from gradients of parameters for the bottom line. The dashed lines give the approximate slope of the scatter. (e, f, g, h) The relation between the variance $\{\text{Var}(\text{Proj}_{\boldsymbol{v}_i(H)}(D))\}_{i=1}^N$ and the eigenvalue $\{\lambda_i(H)\}_{i=1}^N$ for different choices of $p$ and learning rate $lr$ with different network structures. The projection is done for different datasets $D$ sampled from parameters for the top line and sampled from gradients of parameters for the bottom line. The dashed lines give the approximate slope of the scatter.

structures, such as ResNet-20 and transformer, and more complex datasets, such as CIFAR-100 and Multi30k, as shown in Fig. 5.

### B.3 THE SIMILARITY OF EIGENVECTORS BETWEEN THE HESSIAN MATRIX AND THE COVARIANCE MATRIX

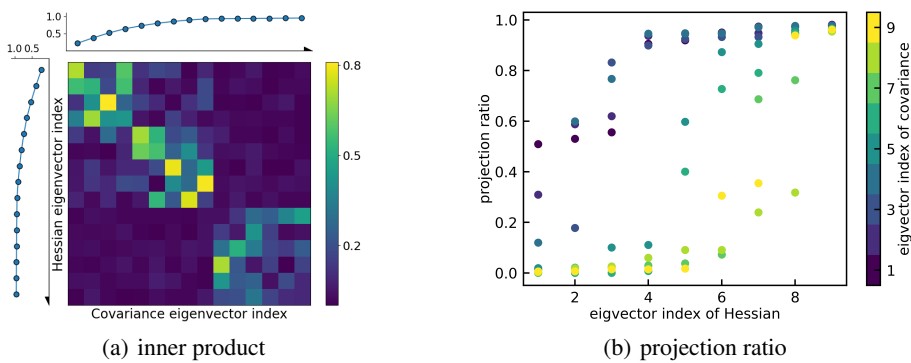

(a) inner product  (b) projection ratio

Figure 6: (a) The cosine similarity between the first 15 eigenvectors of the Hessian matrix and the covariance matrix. The color represents the value of cosine similarity, and the lines on the top and left represent the energy ratio of the first $k$ eigenvalues of the covariance matrix and the Hessian matrix, respectively. (b) The projection ratio of the first 9 eigenvectors of the covariance matrix in different eigendirections of the Hessian matrix.

We study the similarity of eigenvectors between the Hessian matrix and the covariance matrix. The model parameters are derived from the final model represented by the blue line in Fig. 3(a). In Fig. 6(a), we study the cosine similarity between the first 15 eigenvectors of the Hessian matrix and

the covariance matrix. The color represents the value of cosine similarity, and the lines on the top and left represent the energy ratio $E(k)$ of the first $k$ eigenvalues of the covariance matrix and the Hessian matrix respectively, that is, $E(k) = \Sigma_{i=1}^{k}(\lambda_i)^2 / \Sigma_{i=1}^{N}(\lambda_i)^2$, where $\lambda_i$ is the $i$-th eigenvalue, and $N$ is the number of the eigenvalues for the two matrices. It is easy to find that the energy ratio of the first 9 eigenvectors of the two matrices far exceeds the energy ratio of other eigenvectors. Meanwhile, the first 9 eigenvectors of the two matrices are highly similar, which implies the alignment property between the Hessian matrix and the covariance matrices in the eigenspace.

In order to further characterize the similarity between the first few eigenvectors of the two matrices, we study the projection of the first 9 eigenvectors of the covariance matrix in different eigendirections of the Hessian matrix. As shown in Fig. 6(b), we calculate the value of $\Sigma_{i=1}^{k}\alpha(\boldsymbol{v}_i(H), \boldsymbol{v}_j(\Sigma))^2$ for the index $k$ of the Hessian eigenvector and the index $j$ of the covariance eigenvector, where $\alpha(\boldsymbol{v}_i, \boldsymbol{v}_j)$ is the cosine similarity between the two vectors $\boldsymbol{v}_i$, $\boldsymbol{v}_j$. The energy proportions of the first nine eigenvectors of the covariance matrix in the first nine directions of the Hessian matrix are all greater than 0.95, which means that the similarity of the eigenvectors corresponding to the first few eigenvalues of the Hessian matrix and the covariance matrix is much higher than the similarity between them and other feature directions, which further confirms the alignment properties between the Hessian matrix and the covariance matrix in the eigenspace.

## C EXTENDED ANALYSIS OF DROPOUT UNDER COMPLEX NETWORK STRUCTURES AND SGD SETTING

In the main text, the theoretical part and experimental verification mainly focus on the settings of the two-layer network and GD. This is mainly due to the derivation of the specific forms of the drift term and covariance matrix. In fact, the SME framework in this work can be applied to more complex network structures and SGD settings without being limited to the explicit calculation of drift terms and covariance matrices. In the following, we conduct a detailed analysis of the feasibility of SME under complex network structures and SGD settings, and verify it through SME numerical simulation and the alignment between the Hessian matrix and covariance matrix under broader settings.

### C.1 FEASIBILITY ANALYSIS FOR NETWORK STRUCTURES AND SGD SETTING

The formulation of our stochastic modified equations relies on two critical components. Firstly, the algorithm updates parameters iteratively in a recursive manner, i.e., $\theta_N = F(\theta_{N-1}, \delta_N)$, thereby forming iterations as a time-homogeneous Markov chain. This characteristic is foundational for applying the stochastic modified equation approach to weakly approximate the dynamics of dropout. The second crucial component involves the utilization of Taylor's theorem with the Lagrange form of the remainder. In light of these two foundational components, we assert that the derivation and analysis of stochastic modified equations extend beyond the confines of two-layer neural networks and dropout. This approach is applicable to a broader spectrum of stochastic algorithms, including but not limited to SGD and ADAM, and extends naturally to deeper or more complex neural network architectures.

One might wonder why we only demonstrate our results limited to two-layer neural networks. For one thing, two-layer neural networks are indeed a meaningful step towards a thorough understanding of dropout. For another, the specific structure of two-layer neural networks, wherein parameters are 'decoupled', i.e., $\theta = q_r = (a_r, w_r)$, facilitates the demonstration of dropout's effect through the computation of the network output. Finally, for two-layer neural networks, we are able to give out an explicit expression for the modified loss, since we are able to calculate the first and second moment of the random variable $(\delta)_r$. In the case of multi-layer neural networks, if dropout is applied only to the outermost layer, we can still calculate the explicit expression for the modified loss, since its calculation involves solely the first and second moments of the random variable $(\delta)_r$. However, the situation becomes significantly more challenging when dropout is applied to the inner layers of the multi-layer neural network. In this scenario, obtaining a closed-form expression for the expectation $\mathbb{E}[h(\delta)]$, where $h$ is a highly nonlinear function with respect to $\delta$, becomes nearly impossible. Moreover, the computation of the covariance is even more infeasible in multi-layer neural networks compared to two-layer neural networks.

While the structural dynamics of dropout within nonlinear activations for deep neural networks remain uncertain, we assert that the same structure is retained if we apply dropout in a linear manner to the deep neural networks by computing $f_\theta(x) = \sum_{r=1}^m a_r(\delta)_r \sigma(w_r^T x^{[L]})$, where $x^{[L]}$ is the output function of a $L$-layer neural network for $L \geq 2$, the same structure remains, and the modified loss $L_S(\cdot)$ reads:

$$L_S(\theta) := \frac{1}{2n} \sum_{i=1}^n e_i^2 + \frac{1-p}{2np} \sum_{i=1}^n \sum_{r=1}^m a_r^2 \sigma(w_r^T x_i^{[L]})^2.$$

As for the SGD setting, given the nature of SGD as an unbiased estimator with respect to the full sample, we conjecture that an order-$1$ approximation utilizing SME for SGD combined with dropout shall be in the form:

$$d\theta_t = -\nabla_\theta L_S(\theta_t)dt + \widetilde{\Sigma}(\theta_t) dW_t,$$

wherein the drift term remains invariant regardless of GD or SGD, while the diffusion term $\widetilde{\Sigma}(\theta_t)$ combines noise from both dropout and SGD.

As for the scenario where dropout is applied within the nonlinear activations, this exact form of the SME remains unexplored. However, we design numerical experiments to verify the approximation ability of deeper network SMEs in the next subsection, where the drift term here is calculated numerically through multiple samplings on dropout variable $\delta$.

## C.2 Extended experiments for network structures and SGD setting

In this section, we verify the feasibility of the above analysis through numerical experiments. We all use first-order SME for approximation. Please refer to the Appendix A for specific experimental settings.

### C.2.1 SME simulation

In this subsection, we conduct an empirical validation of the feasibility analysis of SMEs for dropout. This validation is conducted through an exploration of the resemblance between the numerical simulation of the SME and the real-time training process of dropout. For the numerical simulation of the SME, unless otherwise specified, we employ the Euler-Maruyama method to approximate its dynamic evolution by the order-1 approximation. To mitigate the computational demands associated with the covariance matrix, we resize the MNIST data to $7 \times 7$, thereby reducing the number of network parameters involved in the simulations.

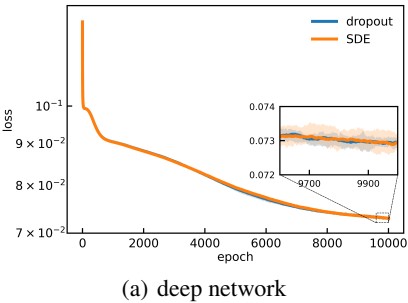
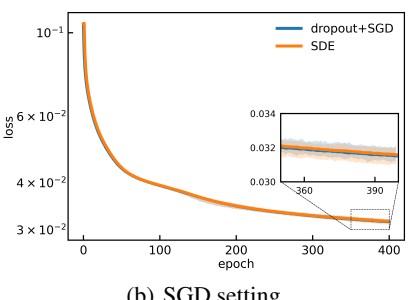

(a) deep network          (b) SGD setting

Figure 7: We train fully connected networks on MNIST. The curves and points are derived from the average results of 20 individual trials, each of which utilized the same initialization distribution. The training loss trajectory obtained by SME simulation or dropout training under four cases of different learning rates and dropout rates. The error bands, portrayed with greater transparency, are derived from the maximum and minimum loss values observed across these 20 random trials at each training step. (a) Four-layer network, each layer width is 40. The learning rate is 0.1 and $p = 0.5$. (b) Two-layer network training with SGD, the batch size is 32. The learning rate is 0.1 and $p = 0.8$.

As shown in Fig. 7, we compare the SME simulation results under deep network and SGD settings with dropout. For deep networks, we add a dropout layer after each hidden layer. Since the drift term cannot be calculated explicitly, we approximate the expectation by sampling the noise multiple times. This also limits the number of parameters in our model. For dropout under the SGD setting, we use the same drift term as under the GD setting, and calculate the noise structure from the combined noise of the two random algorithms, dropout and SGD.

### C.2.2 Hessian-variance alignment

In this section, we mainly study the impact of dropout of complex network structures on Hessian-variance alignment under the SGD setting. In the main text, we have conducted relevant experiments on complex network structures under the GD setting. To investigate the Hessian-Variance alignment relation for dropout with SGD, we study the cosine similarity quantity $\alpha(\boldsymbol{\theta}_t)$ between the covariance matrix $\boldsymbol{\Sigma}_t := \boldsymbol{\Sigma}(\boldsymbol{\theta}_t)$ and the Hessian matrix $\boldsymbol{H}_t := \boldsymbol{H}(\boldsymbol{\theta}_t)$ at each time step $t$ under the SGD setting. $\boldsymbol{\Sigma}_t$ is the covariance matrix of $\mathcal{D}_{\mathrm{grad}}$, a collection of gradients calculated with different dropout variables $\boldsymbol{\delta}$ and input data $\boldsymbol{x}$ sampled at the $t$th step. On the other hand, $\boldsymbol{H}_t$ is the Hessian of the loss function evaluated at the $t$th iteration. Then the crucial cosine similarity metric $\alpha(\boldsymbol{\theta}_t)$ is formally expressed as:

$$\alpha(\boldsymbol{\theta}_t) = \frac{\mathrm{Tr}(\boldsymbol{H}_t\boldsymbol{\Sigma}_t)}{\|\boldsymbol{H}_t\|_{\mathrm{F}}\|\boldsymbol{\Sigma}_t\|_{\mathrm{F}}} \tag{20}$$

As depicted in Fig. 8, it is evident that throughout the training process, $\alpha(\boldsymbol{\theta}_t)$ consistently attains values surpassing 0.85 in Fig. 8(a) and 0.65 in Fig. 8(b), and these observations hold true across

varying learning rates and dropout rates. It's worth noting that, based on 100 samples, the average cosine similarity between the two random matrices based on the selected parameters is only $8.6 \times 10^{-4}$. The eigenvalues of the two random matrices are derived from the eigenvalues of the Hessian matrix and covariance matrix of the selected parameters, and the corresponding eigenvectors are sampled from a normal distribution and normalized. The model parameters are derived from the final model represented by the blue line in Fig. 8(a). Consequently, the introduced noise is highly anisotropic in that it aligns well with the Hessian matrix across all directions. We acknowledged that due to computational constraints, this experiment limits the trace calculation to a subset of parameters, which can be effectively regarded as the projection of the Hessian and the noise into specific directions.

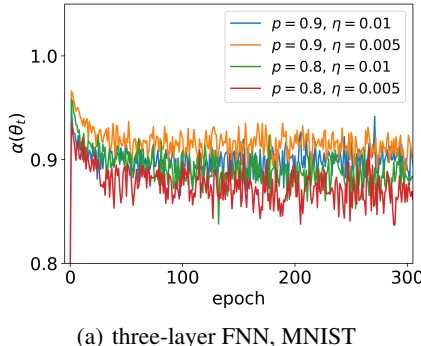
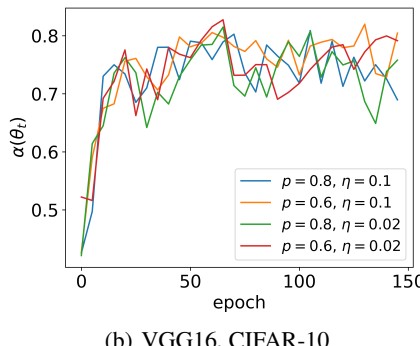

(a) three-layer FNN, MNIST          (b) VGG16, CIFAR-10

Figure 8: The cosine similarity $\alpha(\boldsymbol{\theta}_t)$ between the Hessian of the loss function and the covariance of the dropout noise at each training epoch $t$ for different choices of dropout rate and learning rate. (a) The FNN with size 784-50-50-10 is trained on the MNIST dataset with a batch size of 32. The dropout layer is added after the first hidden layer. (b) The VGG16 is trained on the CFIAR-10 dataset with a batch size of 32. The dropout layers are added after the first two convolutional layers of each block and the first fully-connected layer. The calculation of $\alpha(\boldsymbol{\theta}_t)$ is performed every five epochs.

# D    EFFECT OF NOISE FROM DIFFERENT TRAINING STRATEGIES ON RESULTS.

In this subsection, we compare the effect of noise from different training strategies on model output and generalization. We mainly compared three training strategies: SGD, dropout, and parameter noise injection. Parameter noise injection is well known as the explicit regularizer on the trace of the hessian (Orvieto et al., 2023). We first study the results of ReLU NNs training under three training strategies, as shown in Fig. 9(a). It is easy to see that the model output of GD+dropout has better smoothness and is more in line with expectations. In Fig. 9(b), we compare the generalization capabilities of three randomized algorithms. We follow the settings in Orvieto et al. (2023) and conduct experiments using MLP1 in the corresponding code, with the network size 784-500-500-500-10. We use a batch size of 128 for the three algorithms. The mean (solid line) and standard deviation (shaded range) were calculated from five independent experiments for each setting. For dropout, we add the dropout layer under each hidden layer, and we set $p = 0.8$. For parameter noise injection, we set the noise standard deviation $\sigma = 0.03$. At the same time, we compared the traces of the Hessian matrix of the three training strategies, which is a metric widely used to characterize flatness. As shown in Fig. 9(c), the trace of the Hessian matrix of dropout is smaller than that of the other two training strategies during the entire training process, suggesting a stronger tendency for the flat solution.

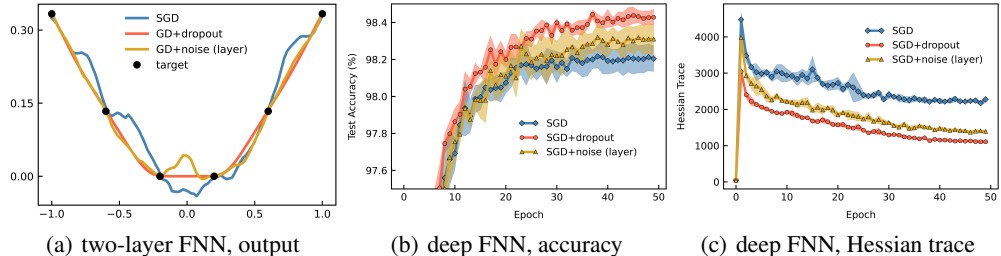

(a) two-layer FNN, output     (b) deep FNN, accuracy     (c) deep FNN, Hessian trace

Figure 9: (a) Two-layer ReLU NN output under different training strategies. The width of the hidden layer is 1000, and the learning rate is $1 \times 10^{-3}$. For SGD, the batch size is 1. For dropout, the dropout layer is added after the hidden layer with $p = 0.8$. For parameter noise injection, we use the layer noise with the noise standard deviation $\sigma = 0.001$. (b, c) The deep FNN trained on MNIST under different training strategies. The network size is 784-500-500-500-10, and the learning rate is 0.01. The batch size is 128. For dropout, the dropout layer is added after the hidden layers with $p = 0.8$. For parameter noise injection, we use the layer noise with the noise standard deviation $\sigma = 0.03$. (b) Test accuracy. (c) Trace of the Hessian matrix.

# E  EXTENDED EXPERIMENTS OF DROPOUT UNDER MEAN-FIELD LIMIT

In the above, we do not treat the network width as the variable under study, but treat it as a constant value of order one. However, if we take the mean-field limit of the two-layer network, i.e.,

$$f_{\boldsymbol{\theta}}(\boldsymbol{x}) := \frac{1}{m} \sum_{r=1}^{m} a_r \sigma(\boldsymbol{w}_r^{\mathsf{T}} \boldsymbol{x}),$$

where $m \to \infty$, then the modified equation has the following form:

$$L_{\mathcal{S}}(\boldsymbol{\theta}) := \frac{1}{2n} \sum_{i=1}^{n} e_i^2 + \frac{1-p}{2npm^2} \sum_{i=1}^{n} \sum_{r=1}^{m} a_r^2 \sigma(\boldsymbol{w}_r^{\mathsf{T}} \boldsymbol{x}_i)^2.$$

The additional regularity of the dropout modified equation disappears in the infinite width limit, which is contrary to our common sense. Intuitively speaking, if we take each neuron to be independent and identically distributed, even if we 'drop' some neurons, dropout does not seem to exert an impact on the training process. We carefully verified this by experimentally studying different wide networks with and without dropout. As shown in Fig. 10, for narrow networks, dropout will have a certain impact on the training process, and this impact will become insignificant when the network is wide enough.

We design experiments with and without dropout at different hidden layer widths under the mean-field setting. The quantity $R_1$ we study is defined as $\frac{1-p}{2nm^2p} \sum_{i=1}^{n} \sum_{r=1}^{m} a_r^2 \sigma(\boldsymbol{w}_r^{\mathsf{T}} \boldsymbol{x}_i)^2$. As shown in Fig. 10, for the wide network, the model loss trajectory is almost consistent with or without dropout, and the cosine similarity between the two model parameter vectors i.e., flatten all the parameters of the model into one vector, is 0.9999998, but there is no such similarity for the narrow network.

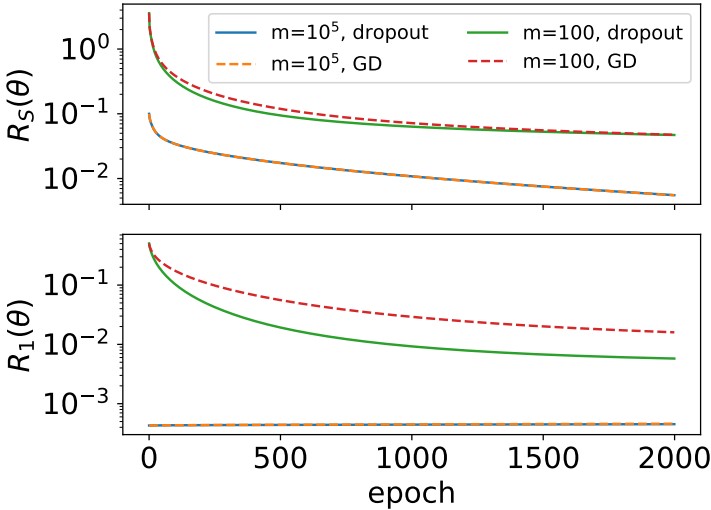

Figure 10: The two-layer network with and without dropout at different hidden layer widths $m$ under the mean-field setting. The learning rate is 0.01 for all experiments, and $p = 0.9$ for experiments with dropout. These settings can make the dropout noise have little effect on the results.

# F  PRELIMINARIES

## F.1  NOTATIONS

We adhere wherever possible to the following notation. Dimensional indices are written as subscripts with a bracket to avoid confusion with other sequential indices (e.g. time, iteration number), which do not have brackets. When more than one indices are present, we separate them with a comma, e.g. $x_{k,(i)}$ is the $i$-th coordinate of the vector $x_k$, the $k^{\text{th}}$ member of a sequence.

We set a special vector $(1, 1, 1, \ldots, 1)^\intercal$ by $\mathbf{1} := (1, 1, 1, \ldots, 1)^\intercal$ whose dimension varies. We set $n$ for the number of input samples, $m$ for the width of the neural network, and $D := m(d + 1)$ hereafter in this paper. We let $[n] = \{1, 2, \ldots, n\}$. We set $\mathcal{N}(\boldsymbol{\mu}, \boldsymbol{\Sigma})$ as the normal distribution with mean $\boldsymbol{\mu}$ and covariance $\boldsymbol{\Sigma}$. We denote $\otimes$ as the Kronecker tensor product, $\langle \cdot, \cdot \rangle$ for standard inner product between two vectors, and $\boldsymbol{A} : \boldsymbol{B}$ for the Frobenius inner product between two matrices $\boldsymbol{A}$ and $\boldsymbol{B}$. We denote vector $L^2$ norm as $\|\cdot\|_2$, vector or function $L_\infty$ norm as $\|\cdot\|_\infty$, function $L_1$ norm as $\|\cdot\|_1$, matrix infinity norm as $\|\cdot\|_{\infty \to \infty}$, matrix spectral (operator) norm as $\|\cdot\|_{2 \to 2}$, and matrix Frobenius norm as $\|\cdot\|_{\mathrm{F}}$ . Finally, we denote the set of continuous functions $f(\cdot) : \mathbb{R}^D \to \mathbb{R}$ possessing continuous derivatives of order up to and including $r$ by $\mathcal{C}^r(\mathbb{R}^D)$, and for a Polish space $\mathcal{X}$, we denote the space of bounded measurable functions by $\mathcal{B}_b(\mathcal{X})$, and the space of bounded continuous functions by $\mathcal{C}_b(\mathcal{X})$. In the mathematical discipline of general topology, a Polish space is a separable complete metric space.

## F.2  PROBLEM SETUP

For the empirical risk minimization problem given by the quadratic loss:

$$\min_{\boldsymbol{\theta}} R_{\mathcal{S}}(\boldsymbol{\theta}) = \frac{1}{2n} \sum_{i=1}^n (f_{\boldsymbol{\theta}}(\boldsymbol{x}_i) - y_i)^2 , \tag{21}$$

where $\mathcal{S} := \{(\boldsymbol{x}_i, y_i)\}_{i=1}^n$ is the training sample, $f_{\boldsymbol{\theta}}(\boldsymbol{x})$ is the prediction function, $\boldsymbol{\theta}$ are the parameters to be optimized over, and their dependence is modeled by a two-layer neural network (NN) with $m$ hidden neurons

$$f_{\boldsymbol{\theta}}(\boldsymbol{x}) := \sum_{r=1}^m a_r \sigma(\boldsymbol{w}_r^\intercal \boldsymbol{x}), \tag{22}$$

where $\boldsymbol{x} \in \mathbb{R}^d$, $\boldsymbol{\theta} = \text{vec}(\boldsymbol{\theta}_a, \boldsymbol{\theta}_w)$ with $\boldsymbol{\theta}_a = \text{vec}(\{a_r\}_{r=1}^m)$, $\boldsymbol{\theta}_w = \text{vec}(\{\boldsymbol{w}_r\}_{r=1}^m)$ is the set of parameters, $\sigma(\cdot)$ is the activation function applied coordinate-wisely to its input, and $\sigma$ is 1-Lipschitz with $\sigma \in \mathcal{C}^\infty(\mathbb{R})$. More precisely, $\boldsymbol{\theta} = \text{vec}(\{\boldsymbol{q}_r\}_{r=1}^m)$ whereas for each $r \in [m]$, $\boldsymbol{q}_r := (a_r, \boldsymbol{w}_r^\intercal)^\intercal$. We remark that the bias term $b_r$ can be incorporated by expanding $\boldsymbol{x}$ and $\boldsymbol{w}_r$ to $(\boldsymbol{x}^\intercal, 1)^\intercal$ and $(\boldsymbol{w}_r^\intercal, b_r)^\intercal$.

Given fixed learning rate $\eta > 0$, then at the $N$-th iteration, where

$$t_N := N\eta,$$

and a scaling vector $\boldsymbol{\delta}_N \in \mathbb{R}^m$ is sampled with independent random coordinates: For each $k \in [m]$,

$$(\boldsymbol{\delta}_N)_k = \begin{cases} \frac{1}{p} & \text{with probability } p, \\ 0 & \text{with probability } 1 - p, \end{cases} \tag{23}$$

and we observe that $\{\boldsymbol{\delta}_N\}_{N \geq 1}$ is an i.i.d. Bernulli sequence with $\mathbb{E}\boldsymbol{\delta}_1 = \mathbf{1}$, and naturally, with slight abuse of notations, the $\sigma$-fields $\mathcal{F}_N := \{\sigma(\boldsymbol{\delta}_1, \boldsymbol{\delta}_2, \cdots \boldsymbol{\delta}_N)\}$ forms a filtration.

We then apply dropout to two-layer NNs by computing

$$\boldsymbol{f}_{\boldsymbol{\theta}}(\boldsymbol{x}; \boldsymbol{\delta}) := \sum_{r=1}^m (\boldsymbol{\delta})_r a_r \sigma(\boldsymbol{w}_r^\intercal \boldsymbol{x}), \tag{24}$$

and we denote the empirical risk associated with dropout by

$$
\begin{aligned}
R_{\mathcal{S}}^{\mathrm{drop}}\left(\boldsymbol{\theta};\boldsymbol{\delta}\right) :&= \frac{1}{2n}\sum_{i=1}^{n}\left(\boldsymbol{f_{\theta}}(\boldsymbol{x}_i;\boldsymbol{\delta})-y_i\right)^2 \\
&= \frac{1}{2n}\sum_{i=1}^{n}\left(\sum_{r=1}^{m}(\boldsymbol{\delta})_r a_r \sigma(\boldsymbol{w}_r^{\mathsf{T}}\boldsymbol{x}_i)-y_i\right)^2.
\end{aligned}
\tag{25}
$$

We observe that the parameters at the $N$-th step are updated via back propagation as follows:

$$
\boldsymbol{\theta}_N = \boldsymbol{\theta}_{N-1} - \eta\nabla_{\boldsymbol{\theta}}R_{\mathcal{S}}^{\mathrm{drop}}\left(\boldsymbol{\theta}_{N-1};\boldsymbol{\delta}_N\right),
\tag{26}
$$

where $\boldsymbol{\theta}_0 := \boldsymbol{\theta}(0)$. Finally, we denote hereafter that for all $i\in[n]$,

$$
e_i^N := e_i(\boldsymbol{\theta}_{N-1};\boldsymbol{\delta}_N) := \boldsymbol{f_{\theta_{N-1}}}(\boldsymbol{x}_i;\boldsymbol{\delta}_N)-y_i,
$$

hence the empirical risk associated with dropout $R_{\mathcal{S}}^{\mathrm{drop}}\left(\boldsymbol{\theta}_{N-1};\boldsymbol{\delta}_N\right)$ can be written into

$$
R_{\mathcal{S}}^{\mathrm{drop}}\left(\boldsymbol{\theta}_{N-1};\boldsymbol{\delta}_N\right) = \frac{1}{2n}\sum_{i=1}^{n}\left(e_i^N\right)^2,
$$

thus the dropout iteration (26) reads

$$
\boldsymbol{\theta}_N - \boldsymbol{\theta}_{N-1} = -\eta\nabla_{\boldsymbol{\theta}}R_{\mathcal{S}}^{\mathrm{drop}}\left(\boldsymbol{\theta}_{N-1};\boldsymbol{\delta}_N\right) = -\frac{\eta}{n}\sum_{i=1}^{n}e_i^N\nabla_{\boldsymbol{\theta}}e_i^N,
$$

and we may proceed to the introduction of the SME approximation.

# G STOCHASTIC MODIFIED EQUATIONS FOR DROPOUT

## G.1 MODIFIED LOSS

Recall that the parameters at the $N$-th step are updated as follows:

$$\boldsymbol{\theta}_N = \boldsymbol{\theta}_{N-1} - \frac{\eta}{n} \sum_{i=1}^{n} e_i^N \nabla_{\boldsymbol{\theta}} e_i^N, \tag{27}$$

and since $\{\boldsymbol{\delta}_N\}_{N \geq 1}$ is an i.i.d. sequence, then the dropout iteration (27) updates the parameters in a recursion form of

$$\boldsymbol{\theta}_N = \boldsymbol{F}(\boldsymbol{\theta}_{N-1}, \boldsymbol{\delta}_N), \tag{28}$$

where $\boldsymbol{F}(\cdot, \cdot) : \mathbb{R}^D \times \mathbb{R}^m \to \mathbb{R}^D$ is a smooth ($\mathcal{C}^\infty$) function, and $\{\boldsymbol{\delta}_N\}_{N \geq 1}$ is a disturbance sequence on $\mathbb{R}^m$, whose marginal distribution possesses a density supported on an open subset of $\mathbb{R}^m$. Then, based on the results in Meyn and Tweedie (2012), the dropout iterations (27) forms a time-homogeneous Markov chain. Thus, we may misuse $\mathbb{E}[\cdot \mid \mathcal{F}_N]$, the conditional expectation given $\mathcal{F}_N$, with $\mathbb{E}_{\boldsymbol{\theta}_{N-1}}[\cdot]$, the conditional expectation given $\boldsymbol{\theta}_{N-1}$. Then, for each $k \in [m]$, the conditional expectation of the increment restricted to $\boldsymbol{q}_k$ reads

$$\mathbb{E}_{\boldsymbol{\theta}_{N-1}} \left[ \sum_{i=1}^{n} e_i^N \nabla_{\boldsymbol{q}_k} e_i^N \right] = \mathbb{E}_{\boldsymbol{\theta}_{N-1}} \left[ \sum_{i=1}^{n} e_i^N (\boldsymbol{\delta}_N)_k \nabla_{\boldsymbol{q}_k} \left( a_k \sigma(\boldsymbol{w}_k^\mathsf{T} \boldsymbol{x}_i) \right) \right],$$

and since

$$\begin{aligned}
\mathbb{E}_{\boldsymbol{\theta}_{N-1}} \left[ e_i^N (\boldsymbol{\delta}_N)_k \right] &= \mathbb{E}_{\boldsymbol{\theta}_{N-1}} \left[ \sum_{r=1, r \neq k}^{m} (\boldsymbol{\delta}_N)_r a_r \sigma(\boldsymbol{w}_r^\mathsf{T} \boldsymbol{x}_i) - y_i \right] \mathbb{E}_{\boldsymbol{\theta}_{N-1}} \left[ (\boldsymbol{\delta}_N)_k \right] \\
&\quad + \mathbb{E}_{\boldsymbol{\theta}_{N-1}} \left[ (\boldsymbol{\delta}_N)_k^2 \right] a_k \sigma(\boldsymbol{w}_k^\mathsf{T} \boldsymbol{x}_i) \\
&= \left( \sum_{r=1, r \neq k}^{m} a_r \sigma(\boldsymbol{w}_r^\mathsf{T} \boldsymbol{x}_i) - y_i \right) + \frac{1}{p} a_k \sigma(\boldsymbol{w}_k^\mathsf{T} \boldsymbol{x}_i) \\
&= \left( \sum_{r=1}^{m} a_r \sigma(\boldsymbol{w}_r^\mathsf{T} \boldsymbol{x}_i) - y_i \right) + \left( \frac{1}{p} - 1 \right) a_k \sigma(\boldsymbol{w}_k^\mathsf{T} \boldsymbol{x}_i).
\end{aligned}$$

For simplicity, given fixed $k \in [m]$, for any $i \in [n]$, we denote hereafter that

$$e_i := e_i(\boldsymbol{\theta}) := \sum_{r=1}^{m} a_r \sigma(\boldsymbol{w}_r^\mathsf{T} \boldsymbol{x}_i) - y_i,$$

$$e_{i, \backslash k} := e_{i, \backslash k}(\boldsymbol{\theta}) := \sum_{r=1, r \neq k}^{m} a_r \sigma(\boldsymbol{w}_r^\mathsf{T} \boldsymbol{x}_i) - y_i,$$

we remark that compared with $e_i^N$, $e_i$ and $e_{i, \backslash k}$ do not depend on the random variable $\boldsymbol{\delta}_N$. Then $\mathbb{E}_{\boldsymbol{\theta}_{N-1}} \left( e_i^N (\boldsymbol{\delta}_N)_k \right)$ can be written in short by

$$\begin{aligned}
\mathbb{E}_{\boldsymbol{\theta}_{N-1}} \left[ e_i^N (\boldsymbol{\delta}_N)_k \right] &= e_{i, \backslash k} + \frac{1}{p} a_k \sigma(\boldsymbol{w}_k^\mathsf{T} \boldsymbol{x}_i) \\
&= e_i + \left( \frac{1}{p} - 1 \right) a_k \sigma(\boldsymbol{w}_k^\mathsf{T} \boldsymbol{x}_i).
\end{aligned} \tag{29}$$

Hence for each $k \in [m]$, expectation of the increment restricted to $\boldsymbol{q}_k$ reads

$$\begin{aligned}
&\mathbb{E}_{\boldsymbol{\theta}_{N-1}} \left[ \sum_{i=1}^{n} e_i^N (\boldsymbol{\delta}_N)_k \nabla_{\boldsymbol{q}_k} \left( a_k \sigma(\boldsymbol{w}_k^\mathsf{T} \boldsymbol{x}_i) \right) \right] \\
&= \sum_{i=1}^{n} e_i \nabla_{\boldsymbol{q}_k} \left( a_k \sigma(\boldsymbol{w}_k^\mathsf{T} \boldsymbol{x}_i) \right) + \sum_{i=1}^{n} \left( \frac{1}{p} - 1 \right) a_k \sigma(\boldsymbol{w}_k^\mathsf{T} \boldsymbol{x}_i) \nabla_{\boldsymbol{q}_k} \left( a_k \sigma(\boldsymbol{w}_k^\mathsf{T} \boldsymbol{x}_i) \right),
\end{aligned}$$

then we define the *modified loss* $L_{\mathcal{S}}(\cdot) : \mathbb{R}^{m(d+1)} \to \mathbb{R}$ for dropout:

$$L_{\mathcal{S}}(\boldsymbol{\theta}) := \frac{1}{2n}\sum_{i=1}^{n}e_i^2 + \frac{1-p}{2np}\sum_{i=1}^{n}\sum_{r=1}^{m}a_r^2\sigma(\boldsymbol{w}_r^{\mathsf{T}}\boldsymbol{x}_i)^2, \tag{30}$$

since as $\boldsymbol{\theta}_{N-1}$ is given, then by taking the conditional expectation, increment of the dropout iteration (27) reads

$$\boldsymbol{\theta}_N - \boldsymbol{\theta}_{N-1} = -\eta\mathbb{E}_{\boldsymbol{\theta}_{N-1}}\left[\nabla_{\boldsymbol{\theta}}R_{\mathcal{S}}^{\mathrm{drop}}\left(\boldsymbol{\theta}_{N-1};\boldsymbol{\delta}_N\right)\right] = -\eta\nabla_{\boldsymbol{\theta}}L_{\mathcal{S}}(\boldsymbol{\theta})\big|_{\boldsymbol{\theta}=\boldsymbol{\theta}_{N-1}},$$

which implies that in the sense of expectations, $\{\boldsymbol{\theta}_N\}_{N\geq 0}$ follows close to the gradient descent trajectory of $L_{\mathcal{S}}(\boldsymbol{\theta})$ with fixed learning rate $\eta$.

### G.2 STOCHASTIC MODIFIED EQUATIONS

We then follow the strategy of Li et al. (2017) to derive the stochastic modified equations (SME) for dropout. Firstly, from the results in Section G.1, we observe that given $\boldsymbol{\theta}_{N-1}$,

$$\boldsymbol{\theta}_N - \boldsymbol{\theta}_{N-1} = -\eta\nabla_{\boldsymbol{\theta}}L_{\mathcal{S}}(\boldsymbol{\theta})\big|_{\boldsymbol{\theta}=\boldsymbol{\theta}_{N-1}} + \sqrt{\eta}\boldsymbol{V}(\boldsymbol{\theta}_{N-1}), \tag{31}$$

where $L_{\mathcal{S}}(\cdot) : \mathbb{R}^{m(d+1)} \to \mathbb{R}$ is the modified loss defined in (30), and $\boldsymbol{V}(\cdot) : \mathbb{R}^{m(d+1)} \to \mathbb{R}^{m(d+1)}$ is a $m(d+1)$-dimensional random vector, and when given $\boldsymbol{\theta}_{N-1}$, $\boldsymbol{V}(\boldsymbol{\theta}_{N-1})$ has mean $\boldsymbol{0}$ and covariance $\eta\boldsymbol{\Sigma}(\boldsymbol{\theta}_{N-1})$, where $\boldsymbol{\Sigma}(\cdot) : \mathbb{R}^{m(d+1)} \to \mathbb{R}^{m(d+1)\times m(d+1)}$ is the covariance of $\nabla_{\boldsymbol{\theta}}R_{\mathcal{S}}^{\mathrm{drop}}\left(\boldsymbol{\theta}_{N-1};\boldsymbol{\delta}_N\right)$. Recall that $\boldsymbol{\theta} = \mathrm{vec}(\{\boldsymbol{q}_r\}_{r=1}^m) = \mathrm{vec}\left(\{(a_r,\boldsymbol{w}_r)\}_{r=1}^m\right)$, and for any $k, r \in [m]$, we denote that

$$\boldsymbol{\Sigma}_{kr}(\boldsymbol{\theta}_{N-1}) := \mathrm{Cov}\left(\nabla_{\boldsymbol{q}_k}R_{\mathcal{S}}^{\mathrm{drop}}\left(\boldsymbol{\theta}_{N-1};\boldsymbol{\delta}_N\right), \nabla_{\boldsymbol{q}_r}R_{\mathcal{S}}^{\mathrm{drop}}\left(\boldsymbol{\theta}_{N-1};\boldsymbol{\delta}_N\right)\right),$$

then

$$\boldsymbol{\Sigma} = \begin{bmatrix} \boldsymbol{\Sigma}_{11} & \boldsymbol{\Sigma}_{12} & \cdots & \boldsymbol{\Sigma}_{1m} \\ \boldsymbol{\Sigma}_{21} & \boldsymbol{\Sigma}_{22} & \cdots & \boldsymbol{\Sigma}_{2m} \\ \vdots & \vdots & \vdots & \vdots \\ \boldsymbol{\Sigma}_{m1} & \boldsymbol{\Sigma}_{m2} & \cdots & \boldsymbol{\Sigma}_{mm} \end{bmatrix}.$$

For each $k \in [m]$, we obtain that

$$\boldsymbol{\Sigma}_{kk}(\boldsymbol{\theta}_{N-1}) = \mathrm{Cov}\left(\nabla_{\boldsymbol{q}_k}R_{\mathcal{S}}^{\mathrm{drop}}\left(\boldsymbol{\theta}_{N-1};\boldsymbol{\delta}_N\right), \nabla_{\boldsymbol{q}_k}R_{\mathcal{S}}^{\mathrm{drop}}\left(\boldsymbol{\theta}_{N-1};\boldsymbol{\delta}_N\right)\right)$$

$$= \left(\frac{1}{p}-1\right)\left(\frac{1}{n}\sum_{i=1}^{n}\left(e_{i,\backslash k} + \frac{1}{p}a_k\sigma(\boldsymbol{w}_k^{\mathsf{T}}\boldsymbol{x}_i)\right)\nabla_{\boldsymbol{q}_k}\left(a_k\sigma(\boldsymbol{w}_k^{\mathsf{T}}\boldsymbol{x}_i)\right)\right)$$

$$\otimes\left(\frac{1}{n}\sum_{i=1}^{n}\left(e_{i,\backslash k} + \frac{1}{p}a_k\sigma(\boldsymbol{w}_k^{\mathsf{T}}\boldsymbol{x}_i)\right)\nabla_{\boldsymbol{q}_k}\left(a_k\sigma(\boldsymbol{w}_k^{\mathsf{T}}\boldsymbol{x}_i)\right)\right)$$

$$+\left(\frac{1}{p^2}-\frac{1}{p}\right)\sum_{l=1,l\neq k}^{m}\left(\frac{1}{n}\sum_{i=1}^{n}a_l\sigma(\boldsymbol{w}_l^{\mathsf{T}}\boldsymbol{x}_i)\nabla_{\boldsymbol{q}_k}\left(a_k\sigma(\boldsymbol{w}_k^{\mathsf{T}}\boldsymbol{x}_i)\right)\right)$$

$$\otimes\left(\frac{1}{n}\sum_{i=1}^{n}a_l\sigma(\boldsymbol{w}_l^{\mathsf{T}}\boldsymbol{x}_i)\nabla_{\boldsymbol{q}_k}\left(a_k\sigma(\boldsymbol{w}_k^{\mathsf{T}}\boldsymbol{x}_i)\right)\right),$$

and for each $k, r \in [m]$ with $k \neq r$,

$$\boldsymbol{\Sigma}_{kr}(\boldsymbol{\theta}_{N-1}) = \mathrm{Cov}\left(\nabla_{\boldsymbol{q}_k} R_{\mathcal{S}}^{\mathrm{drop}}\left(\boldsymbol{\theta}_{N-1}; \boldsymbol{\delta}_N\right), \nabla_{\boldsymbol{q}_r} R_{\mathcal{S}}^{\mathrm{drop}}\left(\boldsymbol{\theta}_{N-1}; \boldsymbol{\delta}_N\right)\right)$$

$$= \left(\frac{1}{p} - 1\right)\left(\frac{1}{n}\sum_{i=1}^n\left(e_{i,\backslash k,\backslash r} + \frac{1}{p}a_k\sigma(\boldsymbol{w}_k^\mathsf{T}\boldsymbol{x}_i) + \frac{1}{p}a_r\sigma(\boldsymbol{w}_r^\mathsf{T}\boldsymbol{x}_i)\right)\nabla_{\boldsymbol{q}_k}\left(a_k\sigma(\boldsymbol{w}_k^\mathsf{T}\boldsymbol{x}_i)\right)\right)$$

$$\otimes\left(\frac{1}{n}\sum_{i=1}^n a_k\sigma(\boldsymbol{w}_k^\mathsf{T}\boldsymbol{x}_i)\nabla_{\boldsymbol{q}_r}\left(a_r\sigma(\boldsymbol{w}_r^\mathsf{T}\boldsymbol{x}_i)\right)\right)$$

$$+\left(\frac{1}{p} - 1\right)\left(\frac{1}{n}\sum_{i=1}^n a_r\sigma(\boldsymbol{w}_r^\mathsf{T}\boldsymbol{x}_i)\nabla_{\boldsymbol{q}_k}\left(a_k\sigma(\boldsymbol{w}_k^\mathsf{T}\boldsymbol{x}_i)\right)\right)$$

$$\otimes\left(\frac{1}{n}\sum_{i=1}^n\left(e_{i,\backslash k,\backslash r} + a_k\sigma(\boldsymbol{w}_k^\mathsf{T}\boldsymbol{x}_i) + \frac{1}{p}a_r\sigma(\boldsymbol{w}_r^\mathsf{T}\boldsymbol{x}_i)\right)\nabla_{\boldsymbol{q}_r}\left(a_r\sigma(\boldsymbol{w}_r^\mathsf{T}\boldsymbol{x}_i)\right)\right),$$

where we denote hereafter that

$$e_{i,\backslash k,\backslash r} := e_{i,\backslash k,\backslash r}(\boldsymbol{\theta}) := \sum_{l=1,l\neq k,l\neq r}^m a_l\sigma(\boldsymbol{w}_l^\mathsf{T}\boldsymbol{x}_i) - y_i,$$

and compared with $e_i^N$, $e_{i,\backslash k,\backslash r}$ still does not depend on the random variable $\boldsymbol{\delta}_N$. We remark that the expression above is consistent in that for the extreme case where $p = 1$, dropout 'degenerates' to gradient descent (GD), hence the covariance matrix degenerates to a zero matrix, i.e., $\boldsymbol{\Sigma} = \boldsymbol{0}_{D\times D}$. We remark that details for the derivation of $\boldsymbol{\Sigma}$ is deferred to Section J.

Now, as we consider the stochastic differential equation (SDE),

$$\mathrm{d}\boldsymbol{\Theta}_t = \boldsymbol{b}\left(\boldsymbol{\Theta}_t\right)\mathrm{d}t + \boldsymbol{\sigma}\left(\boldsymbol{\Theta}_t\right)\mathrm{d}\boldsymbol{W}_t, \quad \boldsymbol{\Theta}_0 = \boldsymbol{\Theta}(0), \tag{32}$$

where $\boldsymbol{W}_t$ is a standard $m(d+1)$-dimensional standard Wiener process, whose Euler–Maruyama discretization with step size $\eta > 0$ at the $N$-th step reads

$$\boldsymbol{\Theta}_{\eta N} = \boldsymbol{\Theta}_{\eta(N-1)} + \eta\boldsymbol{b}\left(\boldsymbol{\Theta}_{\eta(N-1)}\right) + \sqrt{\eta}\boldsymbol{\sigma}\left(\boldsymbol{\Theta}_{\eta(N-1)}\right)\boldsymbol{Z}_N,$$

where $\boldsymbol{Z}_N \sim \mathcal{N}(\boldsymbol{0}, \boldsymbol{I}_{m(d+1)})$ and $\boldsymbol{\Theta}_0 = \boldsymbol{\Theta}(0)$. Thus, if we set

$$\begin{aligned}\boldsymbol{b}\left(\boldsymbol{\Theta}\right) &:= -\nabla_{\boldsymbol{\Theta}}L_{\mathcal{S}}(\boldsymbol{\Theta}), \\ \boldsymbol{\sigma}\left(\boldsymbol{\Theta}\right) &:= \sqrt{\eta}\left(\boldsymbol{\Sigma}\left(\boldsymbol{\Theta}\right)\right)^{\frac{1}{2}}, \\ \boldsymbol{\Theta}_0 &:= \boldsymbol{\theta}_0,\end{aligned} \tag{33}$$

then we would expect (32) to be a 'good' approximation of (31) with the time identification $t = \eta N$. Based on the earlier work of Li et al. (2017), since the path of dropout and the counterpart of SDE are driven by noises sampled in different spaces. Firstly, notice that the stochastic process $\{\boldsymbol{\theta}_N\}_{N\geq 0}$ induces a probability measure on the product space $\mathbb{R}^D \times \mathbb{R}^D \times \cdots \times \mathbb{R}^D \times \cdots$, whereas $\{\boldsymbol{\Theta}_t\}_{t\geq 0}$ induces a probability measure on $\mathcal{C}\left([0,\infty),\mathbb{R}^D\right)$. To compare them, one can form a piece-wise linear interpolation of the former. Alternatively, as we do in this work, we sample a discrete number of points from the latter. Secondly, the process $\{\boldsymbol{\theta}_N\}_{N\geq 0}$ is adapted to the filtration generated by $\mathcal{F}_N$ whereas the process $\{\boldsymbol{\Theta}_t\}_{t\geq 0}$ is adapted to an independent Wiener filtration $\mathcal{F}_t$. Hence, it is not appropriate to compare individual sample paths. Rather, we define below a sense of *weak* approximations (Kloeden and Platen, 2011, Section 9.7) by comparing the distributions of the two processes.

To compare different discrete time approximations, we need to take the rate of weak convergence into consideration, and we also need to choose an appropriate class of functions as the space of test functions. We introduce the following set of smooth functions:

$$\mathcal{C}_b^M\left(\mathbb{R}^{m(d+1)}\right) = \left\{f \in \mathcal{C}^M\left(\mathbb{R}^{m(d+1)}\right) \,\middle|\, \|f\|_{\mathcal{C}^M} := \sum_{|\beta|\leq M}\left\|\mathrm{D}^\beta f\right\|_\infty < \infty\right\},$$

where D is the usual differential operator. We remark that $\mathcal{C}_b^M(\mathbb{R}^D)$ is a subset of $\mathcal{G}(\mathbb{R}^D)$, the class of functions with polynomial growth, which is chosen to be the space of test functions in previous works (Li et al., 2017; Kloeden and Platen, 2011; Malladi et al., 2022).

Before we proceed to the definition of weak approximation, to ensure the rigor and validity of our analysis, we shall assert an assumption regarding the existence and uniqueness of solutions to the SDE (32).

**Assumption 2.** *There exists $T^* > 0$, such that for any time $t \in [0, T^*]$, there exists a unique $t$-continuous solution $\Theta_t$ of the initial value problem:*

$$\mathrm{d}\Theta_t = \boldsymbol{b}(\Theta_t)\,\mathrm{d}t + \boldsymbol{\sigma}(\Theta_t)\,\mathrm{d}\boldsymbol{W}_t, \quad \Theta_0 = \Theta(0),$$

*with the property that $\Theta_t$ is adapted to the filtration $\mathcal{F}_t$ generated by $\boldsymbol{W}_s$ for all time $s \leq t$. Furthermore, for any $t \in [0, T^*]$,*

$$\mathbb{E}\int_0^t \|\Theta_s(\cdot)\|_2^2\,\mathrm{d}s < \infty.$$

*Moreover, we assume that the second, fourth and sixth moments of the solution to SDE (32) are uniformly bounded with respect to time $t$, i.e., for each $l \in [3]$, there exists $C(T^*, \Theta_0) > 0$, such that*

$$\sup_{0 \leq s \leq T^*} \mathbb{E}\|\Theta_s(\cdot)\|_2^{2l} \leq C(T^*, \Theta_0). \tag{34}$$

*As for the dropout iterations (27), we assume further that the second, fourth and sixth moments of the dropout iterations (27) are uniformly bounded with respect to the number of iterations $N$, i.e., let $0 < \eta < 1$, $T > 0$ and set $N_{T,\eta} := \lfloor \frac{T}{\eta} \rfloor$, then for each $l \in [3]$, there exists $T^* > 0$ and $\eta_0 > 0$, such that for any given learning rate $\eta \leq \eta_0$ and all $N \in [0 : N_{T^*,\eta}]$, there exists $C(T^*, \boldsymbol{\theta}_0, \eta_0) > 0$, such that*

$$\sup_{0 \leq N \leq [N_{T^*,\eta}]} \mathbb{E}\|\boldsymbol{\theta}_N\|_2^{2l} \leq C(T^*, \boldsymbol{\theta}_0, \eta_0). \tag{35}$$

We remark that if $\mathcal{G}(\mathbb{R}^D)$ is chosen to be the test functions in Li et al. (2019), then similar relations to (34) and (35) shall be imposed, except that in our cases, we only require the second, fourth and sixth moments to be uniformly bounded, while in their cases, all $2l$-moments are required for $l \geq 1$. Establishments of the validity of Assumption 2 regarding the existence and uniqueness of the SDE will be exhibited in Section I.

The definition of weak approximation is stated out as follows.

**Definition 6.** *The SDE (32) is an order $\alpha$ weak approximation to the dropout (27), if for every $g \in \mathcal{C}_b^M(\mathbb{R}^{m(d+1)})$, there exists $C > 0$ and $\eta_0 > 0$, such that given any $\eta \leq \eta_0$ and $T \leq T^*$, then for all $N \in [N_{T,\eta}]$,*

$$|\mathbb{E}g(\Theta_{\eta N}) - \mathbb{E}g(\boldsymbol{\theta}_N)| \leq C(T, g, \eta_0)\eta^\alpha. \tag{36}$$

# H    SEMIGROUP AND PROOF DETAILS FOR THE MAIN THEOREM

In this section, we use a semigroup approach (Feng et al., 2018) to study the time-homogeneous Markov chains (processes) formed by dropout.

## H.1    DISCRETE AND CONTINUOUS SEMIGROUP

**Definition 7.** *A* Markov operator *over a Polish space $\mathcal{X}$ is a bounded linear operator $\mathcal{P} : \mathcal{B}_b(\mathcal{X}) \to \mathcal{B}_b(\mathcal{X})$ satisfying*

- *$\mathcal{P}\mathbf{1} = \mathbf{1}$;*

- *$\mathcal{P}\varphi$ is positive whenever $\varphi$ is positive;*

- *If a sequence $\{\varphi_n\} \subset \mathcal{B}_b(\mathcal{X})$ converges pointwise to an element $\varphi \in \mathcal{B}_b(\mathcal{X})$, then $\mathcal{P}\varphi_n$ converges pointwise to $\mathcal{P}\varphi$;*

To demonstrate further inequalities that Markov operators satisfy, we offer the following proposition

**Proposition 1.** *A Markov operator $\mathcal{P} : \mathcal{B}_b(\mathcal{X}) \to \mathcal{B}_b(\mathcal{X})$ over a Polish space $\mathcal{X}$ satisfies*

- *$(\mathcal{P}f(\boldsymbol{x}))^+ \leq \mathcal{P}f^+(\boldsymbol{x})$;*

- *$(\mathcal{P}f(\boldsymbol{x}))^- \leq \mathcal{P}f^-(\boldsymbol{x})$;*

- *$|\mathcal{P}f(\boldsymbol{x})| \leq \mathcal{P}|f(\boldsymbol{x})|$.*

*Moreover, if the Polish space $\mathcal{X}$ is equipped with a measure $\mu$, a function $f : \mathcal{X} \to \mathbb{R}$ is said to be an element of $\mathcal{L}^1(\mathcal{X})$ if*

$$\int_{\mathcal{X}} |f| \mathrm{d}\mu < \infty.$$

*Then for every $f \in \mathcal{L}^1(\mathcal{X})$, the following holds*

- *$\|\mathcal{P}f\|_1 \leq \|f\|_1$.*

In mathematics, the positive part of a real function is defined by the formula

$$f^+(\boldsymbol{x}) = \max(f(\boldsymbol{x}), 0) = \begin{cases} f(\boldsymbol{x}) & \text{if } f(\boldsymbol{x}) > 0, \\ 0 & \text{otherwise.} \end{cases}$$

Similarly, the negative part of $f$ is defined as

$$f^-(\boldsymbol{x}) = \max(-f(\boldsymbol{x}), 0) = -\min(f(\boldsymbol{x}), 0) = \begin{cases} -f(\boldsymbol{x}) & \text{if } f(\boldsymbol{x}) < 0, \\ 0 & \text{otherwise.} \end{cases}$$

We proceed to the proof for Proposition 1

*Proof.* From the definition of $f^+$ and $f^-$, it follows that

$$(\mathcal{P}f)^+ = \left(\mathcal{P}f^+ - \mathcal{P}f^-\right)^+ = \max\left(0, \mathcal{P}f^+ - \mathcal{P}f^-\right)$$
$$\leq \max\left(0, \mathcal{P}f^+\right) = \mathcal{P}f^+.$$

Similarly, we obtain that

$$(\mathcal{P}f)^- = \left(\mathcal{P}f^+ - \mathcal{P}f^-\right)^- = \max\left(0, \mathcal{P}f^- - \mathcal{P}f^+\right)$$
$$\leq \max\left(0, \mathcal{P}f^-\right) = \mathcal{P}f^-.$$

Hence for the last inequality

$$|\mathcal{P}f| = (\mathcal{P}f)^+ + (\mathcal{P}f)^-$$
$$\leq \mathcal{P}f^+ + \mathcal{P}f^-$$
$$= \mathcal{P}\left(f^+ + f^-\right) = \mathcal{P}|f|.$$

Finally, by integrating the above relation over $\mathcal{X}$, we obtain that

$$
\begin{aligned}
\|\mathcal{P}f\|_1 &= \int_{\mathcal{X}} |\mathcal{P}f|\,\mathrm{d}\mu \\
&\leq \int_{\mathcal{X}} \mathcal{P}\,|f|\,\mathrm{d}\mu = \int_{\mathcal{X}} |f|\,\mathrm{d}\mu = \|f\|_1 .
\end{aligned}
\tag{37}
$$

$\square$

Inequality (37) is extremely important, and any operator $\mathcal{P}$ that satisfies it is called a contraction. This relation is known as the contractive property of $\mathcal{P}$. To illustrate its power, note that for any $f \in \mathcal{L}^1(\mathcal{X})$, we have

$$
\|\mathcal{P}^n f\|_1 = \left\| \mathcal{P} \circ \mathcal{P}^{n-1} f \right\|_1 \leq \left\| \mathcal{P}^{n-1} f \right\|_1 .
$$

As we consider Markov processes with continuous time, it is natural to consider a family of Markov operators indexed by time. We call such a family a Markov semigroup (Hairer, 2008), provided that it satisfies the relation

$$
\mathcal{P}_{t+s} = \mathcal{P}_t \circ \mathcal{P}_s, \quad \text{for any time } s, t > 0.
\tag{38}
$$

And if given $A \in \mathcal{B}(\mathcal{X})$, where $\mathcal{B}(\mathcal{X})$ is the Borel $\sigma$-algebra on $\mathcal{X}$, and given any two times $s < t$, if the following holds almost surely

$$
\mathbb{P}\left( \boldsymbol{X}_t \in A \mid \boldsymbol{X}_s \right) = \left( \mathcal{P}_{t-s} \mathbf{1}_A \right) \left( \boldsymbol{X}_s \right),
$$

then we call $\boldsymbol{X}_t$ a time-homogeneous Markov process with semigroup $\{\mathcal{P}_t\}_{t \geq 0}$.

In our case for dropout, we set the Polish space $\mathcal{X} = \mathbb{R}^D$, and since $\mathcal{C}_b^M(\mathbb{R}^D) \subset \mathcal{B}_b(\mathbb{R}^D)$, then WLOG we fix $g \in \mathcal{C}_b^M(\mathbb{R}^D)$ and define

$$
\mathcal{P}_\eta g(\tilde{\boldsymbol{\theta}}) := \mathbb{E}\left[ g\left( \tilde{\boldsymbol{\theta}} - \eta \nabla_{\boldsymbol{\theta}} R_{\mathcal{S}}^{\mathrm{drop}}\left( \boldsymbol{\theta}; \boldsymbol{\delta} \right) |_{\boldsymbol{\theta} = \tilde{\boldsymbol{\theta}}} \right) \right].
\tag{39}
$$

We conclude that the dropout iterations (27) forms a time-homogeneous Markov chain with discrete Markov semigroup $\left\{ \mathcal{P}_\eta^n \right\}_{n \geq 0}$.

As for the SDE (32), based on Assumption 2 and combined with the results in (Hairer, 2008, Example 2.11), the Markov semigroup $\{\mathcal{P}_t\}_{t \geq 0}$ associated to the solutions of the SDE reads: For any $g \in \mathcal{B}_b(\mathbb{R}^D)$,

$$
\partial_t \mathcal{P}_t g = \mathcal{L} \mathcal{P}_t g,
$$

where $\mathcal{L}$ is termed the *generator* of the diffusion process (32), which reads

$$
\mathcal{L}g := \langle \boldsymbol{b}, \nabla_{\boldsymbol{\Theta}} g \rangle + \frac{1}{2} \boldsymbol{\sigma}\boldsymbol{\sigma}^{\mathsf{T}} : \nabla_{\boldsymbol{\Theta}}^2 g.
\tag{40}
$$

Moreover, for a fixed test function $g \in \mathcal{C}_b^M(\mathbb{R}^D)$, then for any two times $s, t \geq 0$,

$$
\mathcal{P}_t g(\boldsymbol{\Theta}_s) := \exp(t\mathcal{L}) g(\boldsymbol{\Theta}_s) := \mathbb{E}_{\boldsymbol{\Theta}_s}\left[ g(\boldsymbol{\Theta}_{t+s}) \right],
\tag{41}
$$

and $\{\mathcal{P}_t\}_{t \geq 0}$ forms a continuous Markov semigroup for the SDE (32).

### H.2 SEMIGROUP EXPANSION WITH ACCURACY OF ORDER ONE

Our results are essentially based on Itô-Taylor expansions (Kloeden and Platen, 2011) or Taylor's theorem with the Lagrange form of the remainder (Li et al., 2019, Lemma 27).

**Theorem 2** (Order-1 accuracy). *Fix time $T \leq T^*$, if we choose*

$$
\boldsymbol{b}\left( \boldsymbol{\Theta} \right) := -\nabla_{\boldsymbol{\Theta}} L_{\mathcal{S}}(\boldsymbol{\Theta}),
$$

$$
\boldsymbol{\sigma}\left( \boldsymbol{\Theta} \right) := \sqrt{\eta}\left( \boldsymbol{\Sigma}\left( \boldsymbol{\Theta} \right) \right)^{\frac{1}{2}},
$$

*then for all $t \in [0, T]$, the stochastic processes $\boldsymbol{\Theta}_t$ satisfying*

$$
\mathrm{d}\boldsymbol{\Theta}_t = \boldsymbol{b}\left( \boldsymbol{\Theta}_t \right)\mathrm{d}t + \boldsymbol{\sigma}\left( \boldsymbol{\Theta}_t \right)\mathrm{d}\boldsymbol{W}_t, \quad \boldsymbol{\Theta}_0 = \boldsymbol{\Theta}(0),
\tag{42}
$$

*is an order-1 approximation of dropout (27), i.e., given any test function $g \in \mathcal{C}_b^4(\mathbb{R}^D)$, there exists $\eta_0 > 0$ and $C(T, \|g\|_{C^4}, \eta_0) > 0$, such that for any $\eta \leq \eta_0$ and $T \leq T^*$, and for all $N \in [N_{T,\eta}]$, the following holds:*

$$
\left| \mathbb{E}g(\boldsymbol{\theta}_N) - \mathbb{E}g(\boldsymbol{\Theta}_{\eta N}) \right| \leq C(T, \|g\|_{C^4}, \boldsymbol{\theta}_0, \eta_0)\boldsymbol{\delta},
\tag{43}
$$

*where $\boldsymbol{\theta}_0 = \boldsymbol{\Theta}_0$.*

*Proof.* By application of Taylor's theorem with the Lagrange form of the remainder, we have that for some $\alpha \geq 1$,

$$g(\boldsymbol{\vartheta}) - g(\tilde{\boldsymbol{\vartheta}}) = \sum_{s=1}^{\alpha} \frac{1}{s!} \sum_{i_1,\ldots,i_j=1}^{D} \prod_{j=1}^{s} \left[ \boldsymbol{\vartheta}_{(i_j)} - \tilde{\boldsymbol{\vartheta}}_{(i_j)} \right] \frac{\partial^s g}{\partial \boldsymbol{\vartheta}_{(i_1)} \ldots \partial \boldsymbol{\vartheta}_{(i_j)}}(\tilde{\boldsymbol{\vartheta}})$$

$$+ \frac{1}{(\alpha+1)!} \sum_{i_1,\ldots,i_j=1}^{D} \prod_{j=1}^{\alpha+1} \left[ \boldsymbol{\vartheta}_{(i_j)} - \tilde{\boldsymbol{\vartheta}}_{(i_j)} \right] \frac{\partial^{\alpha+1} g}{\partial \boldsymbol{\vartheta}_{(i_1)} \ldots \partial \boldsymbol{\vartheta}_{(i_j)}}(\gamma \boldsymbol{\vartheta} + (1-\gamma)\tilde{\boldsymbol{\vartheta}}),$$

for some $\gamma \in (0,1)$. We adopt the Einstein's summation convention, where repeated (spatial) indices are summed, i.e.,

$$\boldsymbol{x}_{(i)} \boldsymbol{x}_{(i)} := \sum_{i=1}^{D} \boldsymbol{x}_{(i)} \boldsymbol{x}_{(i)}.$$

As we choose $\boldsymbol{\vartheta} := \boldsymbol{\theta}_1$, $\tilde{\boldsymbol{\vartheta}} := \boldsymbol{\theta}_0$ and $\alpha = 1$, then we obtain that

$$g(\boldsymbol{\theta}_1) - g(\boldsymbol{\theta}_0) = \langle \nabla_{\boldsymbol{\theta}} g(\boldsymbol{\theta}_0), \boldsymbol{\theta}_1 - \boldsymbol{\theta}_0 \rangle$$

$$+ \frac{1}{2} \nabla_{\boldsymbol{\theta}}^2 g(\gamma \boldsymbol{\theta}_1 + (1-\gamma)\boldsymbol{\theta}_0) : (\boldsymbol{\theta}_1 - \boldsymbol{\theta}_0) \otimes (\boldsymbol{\theta}_1 - \boldsymbol{\theta}_0)$$

$$= \langle \nabla_{\boldsymbol{\theta}} g(\boldsymbol{\theta}_0), \boldsymbol{\theta}_1 - \boldsymbol{\theta}_0 \rangle + \frac{1}{2} \nabla_{\boldsymbol{\theta}}^2 g(\tilde{\boldsymbol{\theta}}_0) : (\boldsymbol{\theta}_1 - \boldsymbol{\theta}_0) \otimes (\boldsymbol{\theta}_1 - \boldsymbol{\theta}_0),$$

where $\tilde{\boldsymbol{\theta}}_0 := \gamma \boldsymbol{\theta}_1 + (1-\gamma)\boldsymbol{\theta}_0$, and we observe that since

$$\boldsymbol{\theta}_1 - \boldsymbol{\theta}_0 = -\eta \nabla_{\boldsymbol{\theta}} L_{\mathcal{S}}(\boldsymbol{\theta})\big|_{\boldsymbol{\theta}=\boldsymbol{\theta}_0} + \sqrt{\eta} \boldsymbol{V}(\boldsymbol{\theta}_0),$$

then

$$\mathbb{E} g(\boldsymbol{\theta}_1) - \mathbb{E} g(\boldsymbol{\theta}_0) = \langle \nabla_{\boldsymbol{\theta}} g(\boldsymbol{\theta}_0), \mathbb{E}\boldsymbol{\theta}_1 - \mathbb{E}\boldsymbol{\theta}_0 \rangle + \frac{1}{2} \mathbb{E}\left[ \nabla_{\boldsymbol{\theta}}^2 g(\tilde{\boldsymbol{\theta}}_0) : (\boldsymbol{\theta}_1 - \boldsymbol{\theta}_0) \otimes (\boldsymbol{\theta}_1 - \boldsymbol{\theta}_0) \right]$$

$$= -\eta \left\langle \nabla_{\boldsymbol{\theta}} g(\boldsymbol{\theta}_0), \nabla_{\boldsymbol{\theta}} L_{\mathcal{S}}(\boldsymbol{\theta})\big|_{\boldsymbol{\theta}=\boldsymbol{\theta}_0} \right\rangle + E_\eta^1(\boldsymbol{\theta}_0),$$

where the remainder term $E_\eta^1(\cdot) : \mathbb{R}^D \to \mathbb{R}$, whose expression reads

$$E_\eta^1(\boldsymbol{\theta}_0) := \frac{1}{2} \mathbb{E}\left[ \nabla_{\boldsymbol{\theta}}^2 g(\tilde{\boldsymbol{\theta}}_0) : (\boldsymbol{\theta}_1 - \boldsymbol{\theta}_0) \otimes (\boldsymbol{\theta}_1 - \boldsymbol{\theta}_0) \right], \tag{44}$$

and we remark that $\tilde{\boldsymbol{\theta}}_0$ and $\boldsymbol{\theta}_1$ are implicitly defined by $\boldsymbol{\theta}_0$. Then, directly from Assumption 2, we obtain that

$$E_\eta^1(\boldsymbol{\theta}_0) = \frac{1}{2} \mathbb{E}\left[ \nabla_{\boldsymbol{\theta}}^2 g(\tilde{\boldsymbol{\theta}}_0) : (\boldsymbol{\theta}_1 - \boldsymbol{\theta}_0) \otimes (\boldsymbol{\theta}_1 - \boldsymbol{\theta}_0) \right]$$

$$\leq \frac{1}{2} \|g\|_{C^4} \mathbb{E} \|\boldsymbol{\theta}_1 - \boldsymbol{\theta}_0\|_2^2 = \eta^2 \|g\|_{C^4} \mathbb{E}\left[ \left\| \nabla_{\boldsymbol{\theta}} R_{\mathcal{S}}^{\mathrm{drop}}(\boldsymbol{\theta}_0; \boldsymbol{\delta}_1) \right\|_2^2 \right]$$

$$\leq \eta^2 \|g\|_{C^4} C(T^*, \boldsymbol{\theta}_0, \eta_0),$$

since $\nabla_{\boldsymbol{\theta}} L_{\mathcal{S}}(\boldsymbol{\theta})$ and $\boldsymbol{\Sigma}(\boldsymbol{\theta})$ can be bounded above by the second and fourth moments of the dropout iteration (27).

We observe that

$$\boldsymbol{\Theta}_\eta - \boldsymbol{\Theta}_0 = \int_0^\eta \boldsymbol{b}(\boldsymbol{\Theta}_s)\mathrm{d}s + \int_0^\eta \boldsymbol{\sigma}(\boldsymbol{\Theta}_s)\mathrm{d}\boldsymbol{W}_s.$$

As we choose $\boldsymbol{\vartheta} := \boldsymbol{\Theta}_\eta$, $\tilde{\boldsymbol{\vartheta}} := \boldsymbol{\Theta}_0$ and $\alpha = 1$, then we obtain that

$$g(\boldsymbol{\Theta}_\eta) - g(\boldsymbol{\Theta}_0) = \langle \nabla_{\boldsymbol{\Theta}} g(\boldsymbol{\Theta}_0), \boldsymbol{\Theta}_\eta - \boldsymbol{\Theta}_0 \rangle$$

$$+ \frac{1}{2} \nabla_{\boldsymbol{\Theta}}^2 g(\widetilde{\boldsymbol{\Theta}}_0) : (\boldsymbol{\Theta}_\eta - \boldsymbol{\Theta}_0) \otimes (\boldsymbol{\Theta}_\eta - \boldsymbol{\Theta}_0),$$

where

$$\widetilde{\boldsymbol{\Theta}}_0 := \gamma \boldsymbol{\Theta}_\eta + (1-\gamma)\boldsymbol{\Theta}_0,$$

for some $\gamma \in (0, 1)$. Then

$$\mathbb{E}g(\boldsymbol{\Theta}_\eta) - \mathbb{E}g(\boldsymbol{\Theta}_0)$$

$$= \langle \nabla_{\boldsymbol{\Theta}}g(\boldsymbol{\Theta}_0), \mathbb{E}\boldsymbol{\Theta}_\eta - \mathbb{E}\boldsymbol{\Theta}_0 \rangle + \frac{1}{2}\mathbb{E}\left[\nabla_{\boldsymbol{\Theta}}^2 g(\widetilde{\boldsymbol{\Theta}}_0) : (\boldsymbol{\Theta}_\eta - \boldsymbol{\Theta}_0) \otimes (\boldsymbol{\Theta}_\eta - \boldsymbol{\Theta}_0)\right]$$

$$= \left\langle \nabla_{\boldsymbol{\Theta}}g(\boldsymbol{\Theta}_0), \int_0^\eta \mathbb{E}[\boldsymbol{b}(\boldsymbol{\Theta}_s)]\mathrm{d}s \right\rangle + \frac{1}{2}\mathbb{E}\left[\nabla_{\boldsymbol{\Theta}}^2 g(\widetilde{\boldsymbol{\Theta}}_0) : (\boldsymbol{\Theta}_\eta - \boldsymbol{\Theta}_0) \otimes (\boldsymbol{\Theta}_\eta - \boldsymbol{\Theta}_0)\right],$$

and since

$$\langle \nabla_{\boldsymbol{\Theta}}g(\boldsymbol{\Theta}_0), \mathbb{E}[\boldsymbol{b}(\boldsymbol{\Theta}_s)] \rangle = \langle \nabla_{\boldsymbol{\Theta}}g(\boldsymbol{\Theta}_0), \mathbb{E}[\boldsymbol{b}(\boldsymbol{\Theta}_0)] \rangle + \int_0^s \mathcal{L}\langle \nabla_{\boldsymbol{\Theta}}g(\boldsymbol{\Theta}_0), \boldsymbol{b}\rangle(\boldsymbol{\Theta}_v)\mathrm{d}v,$$

then we obtain that

$$\mathbb{E}g(\boldsymbol{\Theta}_\eta) - \mathbb{E}g(\boldsymbol{\Theta}_0) = \eta\langle \nabla_{\boldsymbol{\Theta}}g(\boldsymbol{\Theta}_0), \mathbb{E}[\boldsymbol{b}(\boldsymbol{\Theta}_0)] \rangle + \int_0^\eta \int_0^s \mathcal{L}\langle \nabla_{\boldsymbol{\Theta}}g(\boldsymbol{\Theta}_0), \boldsymbol{b}\rangle(\boldsymbol{\Theta}_v)\mathrm{d}v\mathrm{d}s$$

$$+ \frac{1}{2}\mathbb{E}\left[\nabla_{\boldsymbol{\Theta}}^2 g(\widetilde{\boldsymbol{\Theta}}_0) : (\boldsymbol{\Theta}_\eta - \boldsymbol{\Theta}_0) \otimes (\boldsymbol{\Theta}_\eta - \boldsymbol{\Theta}_0)\right]$$

$$= \eta\langle \nabla_{\boldsymbol{\Theta}}g(\boldsymbol{\Theta}_0), \boldsymbol{b}(\boldsymbol{\Theta}_0) \rangle + \eta^2 \bar{E}_\eta^1(\boldsymbol{\Theta}_0),$$

where the remainder term $\bar{E}_\eta^1(\cdot) : \mathbb{R}^D \to \mathbb{R}$, whose expression reads

$$\bar{E}_\eta^1(\boldsymbol{\Theta}_0) := \int_0^\eta \int_0^s \mathcal{L}\langle \nabla_{\boldsymbol{\Theta}}g(\boldsymbol{\Theta}_0), \boldsymbol{b}\rangle(\boldsymbol{\Theta}_v)\mathrm{d}v\mathrm{d}s$$

$$+ \frac{1}{2}\mathbb{E}\left[\nabla_{\boldsymbol{\Theta}}^2 g(\widetilde{\boldsymbol{\Theta}}_0) : (\boldsymbol{\Theta}_\eta - \boldsymbol{\Theta}_0) \otimes (\boldsymbol{\Theta}_\eta - \boldsymbol{\Theta}_0)\right], \tag{45}$$

and we remark that $\widetilde{\boldsymbol{\Theta}}_0$ and $\boldsymbol{\Theta}_\eta$ are implicitly defined by $\boldsymbol{\Theta}_0$. As we choose

$$\boldsymbol{b}(\boldsymbol{\Theta}) = -\nabla_{\boldsymbol{\Theta}}L_{\mathcal{S}}(\boldsymbol{\Theta}),$$

$$\boldsymbol{\sigma}(\boldsymbol{\Theta}) = \sqrt{\eta}\,(\boldsymbol{\Sigma}(\boldsymbol{\Theta}))^{\frac{1}{2}},$$

then we carry out the computation for $\mathcal{L}\langle \nabla_{\boldsymbol{\Theta}}g(\boldsymbol{\Theta}_0), \boldsymbol{b}\rangle(\boldsymbol{\Theta}_v)$,

$$\mathcal{L}\langle \nabla_{\boldsymbol{\Theta}}g(\boldsymbol{\Theta}_0), \boldsymbol{b}\rangle(\boldsymbol{\Theta}_v) = \langle \nabla_{\boldsymbol{\Theta}}L_{\mathcal{S}}(\boldsymbol{\Theta}_v), \nabla_{\boldsymbol{\Theta}}\langle \nabla_{\boldsymbol{\Theta}}g(\boldsymbol{\Theta}_0), \nabla_{\boldsymbol{\Theta}}L_{\mathcal{S}}(\boldsymbol{\Theta})\rangle|_{\boldsymbol{\Theta}=\boldsymbol{\Theta}_v}\rangle$$

$$+ \frac{\eta}{2}\boldsymbol{\Sigma}(\boldsymbol{\Theta}_v) : \nabla_{\boldsymbol{\Theta}}^2(\langle \nabla_{\boldsymbol{\Theta}}g(\boldsymbol{\Theta}_0), \nabla_{\boldsymbol{\Theta}}L_{\mathcal{S}}(\boldsymbol{\Theta})\rangle)|_{\boldsymbol{\Theta}=\boldsymbol{\Theta}_v},$$

since $\nabla_{\boldsymbol{\Theta}}L_{\mathcal{S}}(\boldsymbol{\Theta}), \nabla_{\boldsymbol{\Theta}}^2 L_{\mathcal{S}}(\boldsymbol{\Theta}), \nabla_{\boldsymbol{\Theta}}^3 L_{\mathcal{S}}(\boldsymbol{\Theta})$ and $\boldsymbol{\Sigma}(\boldsymbol{\Theta})$ can be bounded above by the second, fourth and sixth moments of the solution to SDE (32), hence we may apply the mean value theorem to (45) and obtain that

$$\left|\bar{E}_\eta^1(\boldsymbol{\Theta}_0)\right| = \left|\int_0^\eta s\mathcal{L}\langle \nabla_{\boldsymbol{\Theta}}g(\boldsymbol{\Theta}_0), \boldsymbol{b}\rangle(\widetilde{\boldsymbol{\Theta}}_s)\mathrm{d}s + \frac{1}{2}\mathbb{E}\left[\nabla_{\boldsymbol{\Theta}}^2 g(\widetilde{\boldsymbol{\Theta}}_0) : (\boldsymbol{\Theta}_\eta - \boldsymbol{\Theta}_0) \otimes (\boldsymbol{\Theta}_\eta - \boldsymbol{\Theta}_0)\right]\right|$$

$$\leq \int_0^\eta s\|g\|_{C^4}C(T^*, \boldsymbol{\Theta}_0)\mathrm{d}s + \frac{1}{2}\|g\|_{C^4}\mathbb{E}\|\boldsymbol{\Theta}_\eta - \boldsymbol{\Theta}_0\|_2^2$$

$$\leq \frac{\eta^2}{2}\|g\|_{C^4}C(T^*, \boldsymbol{\Theta}_0) + \|g\|_{C^4}\mathbb{E}\left\|\int_0^\eta \boldsymbol{b}(\boldsymbol{\Theta}_s)\mathrm{d}s + \int_0^\eta \boldsymbol{\sigma}(\boldsymbol{\Theta}_s)\mathrm{d}\boldsymbol{W}_s\right\|_2^2$$

$$\leq \frac{\eta^2}{2}\|g\|_{C^4}C(T^*, \boldsymbol{\Theta}_0) + 2\|g\|_{C^4}\mathbb{E}\left\|\int_0^\eta \boldsymbol{b}(\boldsymbol{\Theta}_s)\mathrm{d}s\right\|_2^2$$

$$+ 2\|g\|_{C^4}\mathbb{E}\left\|\int_0^\eta \boldsymbol{\sigma}(\boldsymbol{\Theta}_s)\mathrm{d}\boldsymbol{W}_s\right\|_2^2$$

$$\leq \frac{\eta^2}{2}\|g\|_{C^4}C(T^*, \boldsymbol{\Theta}_0) + 2\|g\|_{C^4}\eta^2\mathbb{E}\left\|\nabla_{\boldsymbol{\Theta}}L_{\mathcal{S}}(\widetilde{\boldsymbol{\Theta}}_0)\right\|_2^2$$

$$+ 2\|g\|_{C^4}\mathbb{E}\int_0^\eta \|\boldsymbol{\sigma}(\boldsymbol{\Theta}_s)\|_{\mathrm{F}}^2\,\mathrm{d}s$$

$$\leq \frac{\eta^2}{2}\|g\|_{C^4}C(T^*, \boldsymbol{\Theta}_0) + 2\|g\|_{C^4}\eta^2\mathbb{E}\left\|\nabla_{\boldsymbol{\Theta}}L_{\mathcal{S}}(\widetilde{\boldsymbol{\Theta}}_0)\right\|_2^2$$

$$+ 2\|g\|_{C^4}\eta\mathbb{E}\left[\eta\left\|\boldsymbol{\Sigma}(\widetilde{\boldsymbol{\Theta}}_0)\right\|_{\mathrm{F}}\right] \leq \eta^2\|g\|_{C^4}C(T^*, \boldsymbol{\Theta}_0).$$

To sum up for now,

$$|\mathbb{E}g(\boldsymbol{\theta}_1) - \mathbb{E}g(\boldsymbol{\Theta}_\eta)| = \left|\mathbb{E}g(\boldsymbol{\theta}_0) - \eta\left\langle\nabla_{\boldsymbol{\theta}}g(\boldsymbol{\theta}_0), \nabla_{\boldsymbol{\theta}}L_\mathcal{S}(\boldsymbol{\theta})\big|_{\boldsymbol{\theta}=\boldsymbol{\theta}_0}\right\rangle + E_\eta^1(\boldsymbol{\theta}_0)\right.$$
$$\left. - \mathbb{E}g(\boldsymbol{\Theta}_0) - \eta\langle\nabla_{\boldsymbol{\Theta}}g(\boldsymbol{\Theta}_0), \boldsymbol{b}(\boldsymbol{\Theta}_0)\rangle + \bar{E}_\eta^1(\boldsymbol{\Theta}_0)\right|,$$

since $\boldsymbol{\theta}_0 = \boldsymbol{\Theta}_0$ and $\boldsymbol{b}(\boldsymbol{\Theta}_0) = -\nabla_{\boldsymbol{\Theta}}L_\mathcal{S}(\boldsymbol{\Theta})\big|_{\boldsymbol{\theta}=\boldsymbol{\theta}_0}$, thus

$$
\begin{aligned}
\left|\mathcal{P}_\eta^1 g(\boldsymbol{\theta}_0) - \mathcal{P}_\eta g(\boldsymbol{\Theta}_0)\right| &= |\mathbb{E}g(\boldsymbol{\theta}_1) - \mathbb{E}g(\boldsymbol{\Theta}_\eta)| \\
&\le \left|E_\eta^1(\boldsymbol{\theta}_0)\right| + \left|\bar{E}_\eta^1(\boldsymbol{\Theta}_0)\right| \\
&\le \eta^2\|g\|_{C^4}C(T^*,\boldsymbol{\theta}_0,\eta_0) + \eta^2\|g\|_{C^4}C(T^*,\boldsymbol{\Theta}_0) \\
&= \mathcal{O}(\eta^2).
\end{aligned}
\tag{46}
$$

For the $N$-th step iteration, since

$$|\mathbb{E}g(\boldsymbol{\theta}_N) - \mathbb{E}g(\boldsymbol{\Theta}_{\eta N})| = \left|\mathcal{P}_\eta^N g(\boldsymbol{\theta}_0) - \mathcal{P}_{\eta N}g(\boldsymbol{\Theta}_0)\right|,$$

and the RHS of the above equation can be written into a telescoping sum as

$$\mathcal{P}_\eta^N g(\boldsymbol{\theta}_0) - \mathcal{P}_{\eta N}g(\boldsymbol{\Theta}_0) = \sum_{l=1}^N \left(\mathcal{P}_\eta^{N-l+1}\circ\mathcal{P}_{(l-1)\eta}g(\boldsymbol{\theta}_0) - \mathcal{P}_\eta^{N-l}\circ\mathcal{P}_{l\eta}g(\boldsymbol{\Theta}_0)\right),$$

hence by application of Proposition 1, we obtain that

$$
\begin{aligned}
|\mathbb{E}g(\boldsymbol{\theta}_N) - \mathbb{E}g(\boldsymbol{\Theta}_{\eta N})| &\le \sum_{l=1}^N \left|\mathcal{P}_\eta^{N-l+1}\circ\mathcal{P}_{(l-1)\eta}g(\boldsymbol{\theta}_0) - \mathcal{P}_\eta^{N-l}\circ\mathcal{P}_{l\eta}g(\boldsymbol{\Theta}_0)\right| \\
&\le \sum_{l=1}^N \left|\mathcal{P}_\eta^{N-l}\circ\left(\mathcal{P}_\eta^1\circ\mathcal{P}_{(l-1)\eta} - \mathcal{P}_\eta\circ\mathcal{P}_{(l-1)\eta}\right)g(\boldsymbol{\Theta}_0)\right|,
\end{aligned}
$$

since $\left(\mathcal{P}_\eta^1\circ\mathcal{P}_{(l-1)\eta} - \mathcal{P}_\eta\circ\mathcal{P}_{(l-1)\eta}\right)g(\boldsymbol{\Theta}_0)$ can be regarded as $\mathcal{L}^1(\mathbb{R}^D)$ if we choose measure $\mu$ to be the delta measure concentrated on $\boldsymbol{\Theta}_0$. i.e.,

$$\mu := \delta_{\boldsymbol{\Theta}_0},$$

hence by the conctration property of Markov operators, we obtain further that

$$
\begin{aligned}
|\mathbb{E}g(\boldsymbol{\theta}_N) - \mathbb{E}g(\boldsymbol{\Theta}_{\eta N})| &\le \sum_{l=1}^N \left|\left(\mathcal{P}_\eta^1\circ\mathcal{P}_{(l-1)\eta} - \mathcal{P}_\eta\circ\mathcal{P}_{(l-1)\eta}\right)g(\boldsymbol{\Theta}_0)\right| \\
&\le \sum_{l=1}^N \left|\mathcal{P}_\eta^1 g(\boldsymbol{\Theta}_{(l-1)\eta}) - \mathcal{P}_\eta g(\boldsymbol{\Theta}_{(l-1)\eta})\right|.
\end{aligned}
$$

By taking expectation conditioned on $\boldsymbol{\Theta}_{(l-1)\eta}$, then similar to the relation (46), the following holds

$$
\begin{aligned}
\left|\mathcal{P}_\eta^1 g(\boldsymbol{\Theta}_{(l-1)\eta}) - \mathcal{P}_\eta g(\boldsymbol{\Theta}_{(l-1)\eta})\right| &= \mathbb{E}\left[\left[|\mathbb{E}g(\boldsymbol{\theta}_l) - \mathbb{E}g(\boldsymbol{\Theta}_\eta l)|\Big|\boldsymbol{\Theta}_{(l-1)\eta}\right]\right] \\
&\le \mathbb{E}\left|E_\eta^1(\boldsymbol{\Theta}_{(l-1)\eta})\right| + \mathbb{E}\left|\bar{E}_\eta^1(\boldsymbol{\Theta}_{(l-1)\eta})\right| \\
&\le \eta^2\|g\|_{C^4}C(T^*,\boldsymbol{\theta}_0,\eta_0) + \eta^2\|g\|_{C^4}C(T^*,\boldsymbol{\Theta}_0) \\
&= \mathcal{O}(\eta^2).
\end{aligned}
$$

We remark that the last line of the above relation is essentially based on Assumption 2, since $\mathbb{E}\left|E_\eta^1(\boldsymbol{\Theta}_{(l-1)\eta})\right|$ and $\mathbb{E}\left|\bar{E}_\eta^1(\boldsymbol{\Theta}_{(l-1)\eta})\right|$ can be bounded above by the second, fourth and sixth moments of the solution to SDE (32), hence we may apply dominated convergence theorem to obtain the last line of the above relation.

To sum up, as

$$\left|\mathcal{P}_\eta^N g(\boldsymbol{\theta}_0) - \mathcal{P}_{\eta N}g(\boldsymbol{\Theta}_0)\right| \le \sum_{l=1}^N \left|\mathcal{P}_\eta^{N-l+1}\circ\mathcal{P}_{(l-1)\eta}g(\boldsymbol{\theta}_0) - \mathcal{P}_\eta^{N-l}\circ\mathcal{P}_{l\eta}g(\boldsymbol{\Theta}_0)\right| = N\mathcal{O}(\eta^2),$$

hence for $N = N_{T,\eta}$,

$$\left|\mathcal{P}_\eta^N g(\boldsymbol{\theta}_0) - \mathcal{P}_{\eta N}g(\boldsymbol{\Theta}_0)\right| = N\mathcal{O}(\eta^2) = N\eta\mathcal{O}(\eta) \le T\mathcal{O}(\eta) = \mathcal{O}(\eta).$$

$\square$

## H.3 Semigroup Expansion with Accuracy of Order Two

**Theorem 3** (Order-2 accuracy). *Fix time $T \leq T^*$, if we choose*

$$\boldsymbol{b}(\boldsymbol{\Theta}) = -\nabla_{\boldsymbol{\Theta}} \left( L_{\mathcal{S}}(\boldsymbol{\Theta}) + \frac{\eta}{4} \|\nabla_{\boldsymbol{\Theta}} L_{\mathcal{S}}(\boldsymbol{\Theta})\|_2^2 \right),$$

$$\boldsymbol{\sigma}(\boldsymbol{\Theta}) = \sqrt{\eta} \left( \boldsymbol{\Sigma}(\boldsymbol{\Theta}) \right)^{\frac{1}{2}},$$

*then for all $t \in [0, T]$, the stochastic processes $\boldsymbol{\Theta}_t$ satisfying*

$$\mathrm{d}\boldsymbol{\Theta}_t = \boldsymbol{b}(\boldsymbol{\Theta}_t)\,\mathrm{d}t + \boldsymbol{\sigma}(\boldsymbol{\Theta}_t)\,\mathrm{d}\boldsymbol{W}_t, \quad \boldsymbol{\Theta}_0 = \boldsymbol{\Theta}(0), \tag{47}$$

*is an order-2 approximation of dropout (27), i.e., given any test function $g \in \mathcal{C}_b^6(\mathbb{R}^D)$, there exists $\eta_0 > 0$ and $C(T, \|g\|_{C^6}, \eta_0) > 0$, such that for any $\eta \leq \eta_0$ and $T \leq T^*$, and for all $N \in [N_{T,\eta}]$, the following holds:*

$$|\mathbb{E}g(\boldsymbol{\theta}_N) - \mathbb{E}g(\boldsymbol{\Theta}_{\eta N})| \leq C(T, \|g\|_{C^6}, \boldsymbol{\theta}_0, \eta_0)\boldsymbol{\delta}, \tag{48}$$

*where $\boldsymbol{\theta}_0 = \boldsymbol{\Theta}_0$.*

*Proof.* By application of Taylor's theorem with the Lagrange form of the remainder, we have that for some $\alpha \geq 1$,

$$g(\boldsymbol{\vartheta}) - g(\tilde{\boldsymbol{\vartheta}}) = \sum_{s=1}^{\alpha} \frac{1}{s!} \sum_{i_1,\ldots,i_j=1}^{D} \prod_{j=1}^{s} \left[ \boldsymbol{\vartheta}_{(i_j)} - \tilde{\boldsymbol{\vartheta}}_{(i_j)} \right] \frac{\partial^s g}{\partial \boldsymbol{\vartheta}_{(i_1)} \ldots \partial \boldsymbol{\vartheta}_{(i_j)}}(\tilde{\boldsymbol{\vartheta}})$$

$$+ \frac{1}{(\alpha+1)!} \sum_{i_1,\ldots,i_j=1}^{D} \prod_{j=1}^{\alpha+1} \left[ \boldsymbol{\vartheta}_{(i_j)} - \tilde{\boldsymbol{\vartheta}}_{(i_j)} \right] \frac{\partial^{\alpha+1} g}{\partial \boldsymbol{\vartheta}_{(i_1)} \ldots \partial \boldsymbol{\vartheta}_{(i_j)}}(\gamma\boldsymbol{\vartheta} + (1-\gamma)\tilde{\boldsymbol{\vartheta}}),$$

for some $\gamma \in (0, 1)$.

As we choose $\boldsymbol{\vartheta} := \boldsymbol{\theta}_1$, $\tilde{\boldsymbol{\vartheta}} := \boldsymbol{\theta}_0$ and $\alpha = 2$, with slight misuse of the Frobenius inner product notation, we obtain that

$$g(\boldsymbol{\theta}_1) - g(\boldsymbol{\theta}_0) = \langle \nabla_{\boldsymbol{\theta}} g(\boldsymbol{\theta}_0), \boldsymbol{\theta}_1 - \boldsymbol{\theta}_0 \rangle + \frac{1}{2} \nabla_{\boldsymbol{\theta}}^2 g(\boldsymbol{\theta}_0) : (\boldsymbol{\theta}_1 - \boldsymbol{\theta}_0) \otimes (\boldsymbol{\theta}_1 - \boldsymbol{\theta}_0)$$

$$+ \frac{1}{6} \nabla_{\boldsymbol{\theta}}^3 g(\gamma\boldsymbol{\theta}_1 + (1-\gamma)\boldsymbol{\theta}_0) : (\boldsymbol{\theta}_1 - \boldsymbol{\theta}_0) \otimes (\boldsymbol{\theta}_1 - \boldsymbol{\theta}_0) \otimes (\boldsymbol{\theta}_1 - \boldsymbol{\theta}_0)$$

$$= \langle \nabla_{\boldsymbol{\theta}} g(\boldsymbol{\theta}_0), \boldsymbol{\theta}_1 - \boldsymbol{\theta}_0 \rangle + \frac{1}{2} \nabla_{\boldsymbol{\theta}}^2 g(\boldsymbol{\theta}_0) : (\boldsymbol{\theta}_1 - \boldsymbol{\theta}_0) \otimes (\boldsymbol{\theta}_1 - \boldsymbol{\theta}_0)$$

$$+ \frac{1}{6} \nabla_{\boldsymbol{\theta}}^3 g(\tilde{\boldsymbol{\theta}}_0) : (\boldsymbol{\theta}_1 - \boldsymbol{\theta}_0) \otimes (\boldsymbol{\theta}_1 - \boldsymbol{\theta}_0) \otimes (\boldsymbol{\theta}_1 - \boldsymbol{\theta}_0),$$

where $\tilde{\boldsymbol{\theta}}_0 := \gamma\boldsymbol{\theta}_1 + (1-\gamma)\boldsymbol{\theta}_0$, and we observe that since

$$\boldsymbol{\theta}_1 - \boldsymbol{\theta}_0 = -\eta\nabla_{\boldsymbol{\theta}} L_{\mathcal{S}}(\boldsymbol{\theta})\big|_{\boldsymbol{\theta}=\boldsymbol{\theta}_0} + \sqrt{\eta}\boldsymbol{V}(\boldsymbol{\theta}_0),$$

then

$$\mathbb{E}g(\boldsymbol{\theta}_1) - \mathbb{E}g(\boldsymbol{\theta}_0) = \langle \nabla_{\boldsymbol{\theta}} g(\boldsymbol{\theta}_0), \mathbb{E}\boldsymbol{\theta}_1 - \mathbb{E}\boldsymbol{\theta}_0 \rangle + \frac{1}{2}\nabla_{\boldsymbol{\theta}}^2 g(\boldsymbol{\theta}_0) : \mathbb{E}\left[ (\boldsymbol{\theta}_1 - \boldsymbol{\theta}_0) \otimes (\boldsymbol{\theta}_1 - \boldsymbol{\theta}_0) \right]$$

$$+ \frac{1}{6}\mathbb{E}\left[ \nabla_{\boldsymbol{\theta}}^3 g(\tilde{\boldsymbol{\theta}}_0) : (\boldsymbol{\theta}_1 - \boldsymbol{\theta}_0) \otimes (\boldsymbol{\theta}_1 - \boldsymbol{\theta}_0) \otimes (\boldsymbol{\theta}_1 - \boldsymbol{\theta}_0) \right]$$

$$= -\eta\left\langle \nabla_{\boldsymbol{\theta}} g(\boldsymbol{\theta}_0), \nabla_{\boldsymbol{\theta}} L_{\mathcal{S}}(\boldsymbol{\theta})\big|_{\boldsymbol{\theta}=\boldsymbol{\theta}_0} \right\rangle$$

$$+ \frac{\eta^2}{2}\nabla_{\boldsymbol{\theta}}^2 g(\boldsymbol{\theta}_0) : \left( \nabla_{\boldsymbol{\theta}} L_{\mathcal{S}}(\boldsymbol{\theta})\big|_{\boldsymbol{\theta}=\boldsymbol{\theta}_0} \otimes \nabla_{\boldsymbol{\theta}} L_{\mathcal{S}}(\boldsymbol{\theta})\big|_{\boldsymbol{\theta}=\boldsymbol{\theta}_0} + \boldsymbol{\Sigma}(\boldsymbol{\theta}_0) \right)$$

$$+ E_\eta^2(\boldsymbol{\theta}_0),$$

where the remainder term $E_\eta^2(\cdot) : \mathbb{R}^D \to \mathbb{R}$, whose expression reads

$$E_\eta^2(\boldsymbol{\theta}_0) := \frac{1}{6}\mathbb{E}\left[ \nabla_{\boldsymbol{\theta}}^3 g(\tilde{\boldsymbol{\theta}}_0) : (\boldsymbol{\theta}_1 - \boldsymbol{\theta}_0) \otimes (\boldsymbol{\theta}_1 - \boldsymbol{\theta}_0) \otimes (\boldsymbol{\theta}_1 - \boldsymbol{\theta}_0) \right], \tag{49}$$

and we remark that $\tilde{\boldsymbol{\theta}}_0$ and $\boldsymbol{\theta}_1$ are implicitly defined by $\boldsymbol{\theta}_0$. Then, directly from Assumption 2, we obtain that

$$E_\eta^2(\boldsymbol{\theta}_0) \leq \frac{1}{6} \|g\|_{C^6} \mathbb{E} \|\boldsymbol{\theta}_1 - \boldsymbol{\theta}_0\|_2^3 = \eta^3 \|g\|_{C^6} \mathbb{E}\left[\left\|\nabla_{\boldsymbol{\theta}} R_{\mathcal{S}}^{\mathrm{drop}}(\boldsymbol{\theta}_0; \boldsymbol{\delta}_1)\right\|_2^3\right]$$

$$\leq \eta^3 \|g\|_{C^6} C(T^*, \boldsymbol{\theta}_0, \eta_0),$$

since $\nabla_{\boldsymbol{\theta}} L_{\mathcal{S}}(\boldsymbol{\theta})$ and $\boldsymbol{\Sigma}(\boldsymbol{\theta})$ can be bounded above by the second and fourth moments of the dropout iteration (27).

We observe that

$$\boldsymbol{\Theta}_\eta - \boldsymbol{\Theta}_0 = \int_0^\eta \boldsymbol{b}(\boldsymbol{\Theta}_s)\mathrm{d}s + \int_0^\eta \boldsymbol{\sigma}(\boldsymbol{\Theta}_s)\mathrm{d}\boldsymbol{W}_s.$$

As we choose $\boldsymbol{\vartheta} := \boldsymbol{\Theta}_\eta$, $\tilde{\boldsymbol{\vartheta}} := \boldsymbol{\Theta}_0$ and $\alpha = 3$, then we obtain that

$$g(\boldsymbol{\Theta}_\eta) - g(\boldsymbol{\Theta}_0) = \langle \nabla_{\boldsymbol{\Theta}} g(\boldsymbol{\Theta}_0), \boldsymbol{\Theta}_\eta - \boldsymbol{\Theta}_0\rangle$$

$$+ \frac{1}{2} \nabla_{\boldsymbol{\Theta}}^2 g(\boldsymbol{\Theta}_0) : (\boldsymbol{\Theta}_\eta - \boldsymbol{\Theta}_0) \otimes (\boldsymbol{\Theta}_\eta - \boldsymbol{\Theta}_0)$$

$$+ \frac{1}{6} \nabla_{\boldsymbol{\Theta}}^3 g(\boldsymbol{\Theta}_0) : (\boldsymbol{\Theta}_\eta - \boldsymbol{\Theta}_0) \otimes (\boldsymbol{\Theta}_\eta - \boldsymbol{\Theta}_0) \otimes (\boldsymbol{\Theta}_\eta - \boldsymbol{\Theta}_0)$$

$$+ \frac{1}{24} \nabla_{\boldsymbol{\Theta}}^4 g(\widetilde{\boldsymbol{\Theta}}_0) : (\boldsymbol{\Theta}_\eta - \boldsymbol{\Theta}_0) \otimes (\boldsymbol{\Theta}_\eta - \boldsymbol{\Theta}_0) \otimes (\boldsymbol{\Theta}_\eta - \boldsymbol{\Theta}_0) \otimes (\boldsymbol{\Theta}_\eta - \boldsymbol{\Theta}_0),$$

where

$$\widetilde{\boldsymbol{\Theta}}_0 := \gamma \boldsymbol{\Theta}_\eta + (1 - \gamma) \boldsymbol{\Theta}_0,$$

for some $\gamma \in (0, 1)$. Then

$$\mathbb{E}g(\boldsymbol{\Theta}_\eta) - \mathbb{E}g(\boldsymbol{\Theta}_0)$$

$$= \langle \nabla_{\boldsymbol{\Theta}} g(\boldsymbol{\Theta}_0), \mathbb{E}\boldsymbol{\Theta}_\eta - \mathbb{E}\boldsymbol{\Theta}_0\rangle + \frac{1}{2} \nabla_{\boldsymbol{\Theta}}^2 g(\boldsymbol{\Theta}_0) : \mathbb{E}\left[(\boldsymbol{\Theta}_\eta - \boldsymbol{\Theta}_0) \otimes (\boldsymbol{\Theta}_\eta - \boldsymbol{\Theta}_0)\right]$$

$$+ \frac{1}{6} \nabla_{\boldsymbol{\Theta}}^3 g(\boldsymbol{\Theta}_0) : \mathbb{E}\left[(\boldsymbol{\Theta}_\eta - \boldsymbol{\Theta}_0) \otimes (\boldsymbol{\Theta}_\eta - \boldsymbol{\Theta}_0) \otimes (\boldsymbol{\Theta}_\eta - \boldsymbol{\Theta}_0)\right]$$

$$+ \frac{1}{24} \mathbb{E}\left[\nabla_{\boldsymbol{\Theta}}^4 g(\widetilde{\boldsymbol{\Theta}}_0) : (\boldsymbol{\Theta}_\eta - \boldsymbol{\Theta}_0) \otimes (\boldsymbol{\Theta}_\eta - \boldsymbol{\Theta}_0) \otimes (\boldsymbol{\Theta}_\eta - \boldsymbol{\Theta}_0) \otimes (\boldsymbol{\Theta}_\eta - \boldsymbol{\Theta}_0)\right]$$

$$= \left\langle \nabla_{\boldsymbol{\Theta}} g(\boldsymbol{\Theta}_0), \int_0^\eta \mathbb{E}[\boldsymbol{b}(\boldsymbol{\Theta}_s)]\mathrm{d}s \right\rangle + \frac{1}{2} \nabla_{\boldsymbol{\Theta}}^2 g(\boldsymbol{\Theta}_0) : \mathbb{E}\left[(\boldsymbol{\Theta}_\eta - \boldsymbol{\Theta}_0) \otimes (\boldsymbol{\Theta}_\eta - \boldsymbol{\Theta}_0)\right]$$

$$+ \frac{1}{6} \nabla_{\boldsymbol{\Theta}}^3 g(\boldsymbol{\Theta}_0) : \mathbb{E}\left[(\boldsymbol{\Theta}_\eta - \boldsymbol{\Theta}_0) \otimes (\boldsymbol{\Theta}_\eta - \boldsymbol{\Theta}_0) \otimes (\boldsymbol{\Theta}_\eta - \boldsymbol{\Theta}_0)\right]$$

$$+ \frac{1}{24} \mathbb{E}\left[\nabla_{\boldsymbol{\Theta}}^4 g(\widetilde{\boldsymbol{\Theta}}_0) : (\boldsymbol{\Theta}_\eta - \boldsymbol{\Theta}_0) \otimes (\boldsymbol{\Theta}_\eta - \boldsymbol{\Theta}_0) \otimes (\boldsymbol{\Theta}_\eta - \boldsymbol{\Theta}_0) \otimes (\boldsymbol{\Theta}_\eta - \boldsymbol{\Theta}_0)\right],$$

and since

$$\langle \nabla_{\boldsymbol{\Theta}} g(\boldsymbol{\Theta}_0), \mathbb{E}[\boldsymbol{b}(\boldsymbol{\Theta}_s)]\rangle = \langle \nabla_{\boldsymbol{\Theta}} g(\boldsymbol{\Theta}_0), \mathbb{E}[\boldsymbol{b}(\boldsymbol{\Theta}_0)]\rangle + \int_0^s \mathcal{L} \langle \nabla_{\boldsymbol{\Theta}} g(\boldsymbol{\Theta}_0), \boldsymbol{b}\rangle (\boldsymbol{\Theta}_v)\mathrm{d}v,$$

then we obtain that

$$\mathbb{E}g(\boldsymbol{\Theta}_\eta) - \mathbb{E}g(\boldsymbol{\Theta}_0) = \eta \langle \nabla_{\boldsymbol{\Theta}} g(\boldsymbol{\Theta}_0), \mathbb{E}[\boldsymbol{b}(\boldsymbol{\Theta}_0)]\rangle + \int_0^\eta \int_0^s \mathcal{L} \langle \nabla_{\boldsymbol{\Theta}} g(\boldsymbol{\Theta}_0), \boldsymbol{b}\rangle (\boldsymbol{\Theta}_v)\mathrm{d}v\mathrm{d}s$$

$$+ \frac{1}{2} \nabla_{\boldsymbol{\Theta}}^2 g(\boldsymbol{\Theta}_0) : \mathbb{E}\left[(\boldsymbol{\Theta}_\eta - \boldsymbol{\Theta}_0) \otimes (\boldsymbol{\Theta}_\eta - \boldsymbol{\Theta}_0)\right]$$

$$+ \frac{1}{6} \nabla_{\boldsymbol{\Theta}}^3 g(\boldsymbol{\Theta}_0) : \mathbb{E}\left[(\boldsymbol{\Theta}_\eta - \boldsymbol{\Theta}_0) \otimes (\boldsymbol{\Theta}_\eta - \boldsymbol{\Theta}_0) \otimes (\boldsymbol{\Theta}_\eta - \boldsymbol{\Theta}_0)\right]$$

$$+ \frac{1}{24} \mathbb{E}\left[\nabla_{\boldsymbol{\Theta}}^4 g(\widetilde{\boldsymbol{\Theta}}_0) : (\boldsymbol{\Theta}_\eta - \boldsymbol{\Theta}_0) \otimes (\boldsymbol{\Theta}_\eta - \boldsymbol{\Theta}_0) \otimes (\boldsymbol{\Theta}_\eta - \boldsymbol{\Theta}_0) \otimes (\boldsymbol{\Theta}_\eta - \boldsymbol{\Theta}_0)\right],$$

and once again since

$$\mathcal{L} \langle \nabla_{\boldsymbol{\Theta}} g(\boldsymbol{\Theta}_0), \boldsymbol{b}\rangle (\boldsymbol{\Theta}_v) = \mathcal{L} \langle \nabla_{\boldsymbol{\Theta}} g(\boldsymbol{\Theta}_0), \boldsymbol{b}\rangle (\boldsymbol{\Theta}_0) + \int_0^v \mathcal{L}\left(\mathcal{L} \langle \nabla_{\boldsymbol{\Theta}} g(\boldsymbol{\Theta}_0), \boldsymbol{b}\rangle\right)(\boldsymbol{\Theta}_u)\mathrm{d}u,$$

then we obtain that

$$\mathbb{E}g(\boldsymbol{\Theta}_\eta) - \mathbb{E}g(\boldsymbol{\Theta}_0) = \eta \langle \nabla_{\boldsymbol{\Theta}} g(\boldsymbol{\Theta}_0), \mathbb{E}[\boldsymbol{b}(\boldsymbol{\Theta}_0)] \rangle + \int_0^\eta \int_0^s \mathcal{L} \langle \nabla_{\boldsymbol{\Theta}} g(\boldsymbol{\Theta}_0), \boldsymbol{b} \rangle (\boldsymbol{\Theta}_v) \mathrm{d}v \mathrm{d}s$$

$$+ \frac{1}{2} \nabla_{\boldsymbol{\Theta}}^2 g(\boldsymbol{\Theta}_0) : \mathbb{E}\left[(\boldsymbol{\Theta}_\eta - \boldsymbol{\Theta}_0) \otimes (\boldsymbol{\Theta}_\eta - \boldsymbol{\Theta}_0)\right]$$

$$+ \frac{1}{6} \nabla_{\boldsymbol{\Theta}}^3 g(\boldsymbol{\Theta}_0) : \mathbb{E}\left[(\boldsymbol{\Theta}_\eta - \boldsymbol{\Theta}_0) \otimes (\boldsymbol{\Theta}_\eta - \boldsymbol{\Theta}_0) \otimes (\boldsymbol{\Theta}_\eta - \boldsymbol{\Theta}_0)\right]$$

$$+ \frac{1}{24} \mathbb{E}\left[\nabla_{\boldsymbol{\Theta}}^4 g(\widetilde{\boldsymbol{\Theta}}_0) : (\boldsymbol{\Theta}_\eta - \boldsymbol{\Theta}_0) \otimes (\boldsymbol{\Theta}_\eta - \boldsymbol{\Theta}_0) \otimes (\boldsymbol{\Theta}_\eta - \boldsymbol{\Theta}_0) \otimes (\boldsymbol{\Theta}_\eta - \boldsymbol{\Theta}_0)\right]$$

$$= \eta \langle \nabla_{\boldsymbol{\Theta}} g(\boldsymbol{\Theta}_0), \mathbb{E}[\boldsymbol{b}(\boldsymbol{\Theta}_0)] \rangle + \int_0^\eta \int_0^s \mathcal{L} \langle \nabla_{\boldsymbol{\Theta}} g(\boldsymbol{\Theta}_0), \boldsymbol{b} \rangle (\boldsymbol{\Theta}_0) \mathrm{d}v \mathrm{d}s$$

$$+ \int_0^\eta \int_0^s \int_0^v \mathcal{L} \left(\mathcal{L} \langle \nabla_{\boldsymbol{\Theta}} g(\boldsymbol{\Theta}_0), \boldsymbol{b} \rangle\right) (\boldsymbol{\Theta}_u) \mathrm{d}u \mathrm{d}v \mathrm{d}s$$

$$+ \frac{1}{2} \nabla_{\boldsymbol{\Theta}}^2 g(\boldsymbol{\Theta}_0) : \mathbb{E}\left[(\boldsymbol{\Theta}_\eta - \boldsymbol{\Theta}_0) \otimes (\boldsymbol{\Theta}_\eta - \boldsymbol{\Theta}_0)\right]$$

$$+ \frac{1}{6} \nabla_{\boldsymbol{\Theta}}^3 g(\boldsymbol{\Theta}_0) : \mathbb{E}\left[(\boldsymbol{\Theta}_\eta - \boldsymbol{\Theta}_0) \otimes (\boldsymbol{\Theta}_\eta - \boldsymbol{\Theta}_0) \otimes (\boldsymbol{\Theta}_\eta - \boldsymbol{\Theta}_0)\right]$$

$$+ \frac{1}{24} \mathbb{E}\left[\nabla_{\boldsymbol{\Theta}}^4 g(\widetilde{\boldsymbol{\Theta}}_0) : (\boldsymbol{\Theta}_\eta - \boldsymbol{\Theta}_0) \otimes (\boldsymbol{\Theta}_\eta - \boldsymbol{\Theta}_0) \otimes (\boldsymbol{\Theta}_\eta - \boldsymbol{\Theta}_0) \otimes (\boldsymbol{\Theta}_\eta - \boldsymbol{\Theta}_0)\right]$$

$$= \eta \langle \nabla_{\boldsymbol{\Theta}} g(\boldsymbol{\Theta}_0), \mathbb{E}[\boldsymbol{b}(\boldsymbol{\Theta}_0)] \rangle + \frac{\eta^2}{2} \mathcal{L} \langle \nabla_{\boldsymbol{\Theta}} g(\boldsymbol{\Theta}_0), \boldsymbol{b} \rangle (\boldsymbol{\Theta}_0)$$

$$+ \frac{1}{2} \nabla_{\boldsymbol{\Theta}}^2 g(\boldsymbol{\Theta}_0) : \mathbb{E}\left[(\boldsymbol{\Theta}_\eta - \boldsymbol{\Theta}_0) \otimes (\boldsymbol{\Theta}_\eta - \boldsymbol{\Theta}_0)\right] + \bar{E}_\eta^2(\boldsymbol{\Theta}_0),$$

where the remainder term $\bar{E}_\eta^2(\cdot) : \mathbb{R}^D \to \mathbb{R}$, whose expression reads

$$\bar{E}_\eta^2(\boldsymbol{\Theta}_0) := \int_0^\eta \int_0^s \int_0^v \mathcal{L} \left(\mathcal{L} \langle \nabla_{\boldsymbol{\Theta}} g(\boldsymbol{\Theta}_0), \boldsymbol{b} \rangle\right) (\boldsymbol{\Theta}_u) \mathrm{d}u \mathrm{d}v \mathrm{d}s$$

$$+ \frac{1}{6} \nabla_{\boldsymbol{\Theta}}^3 g(\boldsymbol{\Theta}_0) : \mathbb{E}\left[(\boldsymbol{\Theta}_\eta - \boldsymbol{\Theta}_0) \otimes (\boldsymbol{\Theta}_\eta - \boldsymbol{\Theta}_0) \otimes (\boldsymbol{\Theta}_\eta - \boldsymbol{\Theta}_0)\right] \quad (50)$$

$$+ \frac{1}{24} \mathbb{E}\left[\nabla_{\boldsymbol{\Theta}}^4 g(\widetilde{\boldsymbol{\Theta}}_0) : (\boldsymbol{\Theta}_\eta - \boldsymbol{\Theta}_0) \otimes (\boldsymbol{\Theta}_\eta - \boldsymbol{\Theta}_0) \otimes (\boldsymbol{\Theta}_\eta - \boldsymbol{\Theta}_0) \otimes (\boldsymbol{\Theta}_\eta - \boldsymbol{\Theta}_0)\right],$$

and we remark that $\widetilde{\boldsymbol{\Theta}}_0$ and $\boldsymbol{\Theta}_\eta$ are implicitly defined by $\boldsymbol{\Theta}_0$. As we choose

$$\boldsymbol{b}(\boldsymbol{\Theta}) = -\nabla_{\boldsymbol{\Theta}} \left(L_{\mathcal{S}}(\boldsymbol{\Theta}) + \frac{\eta}{4} \|\nabla_{\boldsymbol{\Theta}} L_{\mathcal{S}}(\boldsymbol{\Theta})\|_2^2\right),$$

$$\boldsymbol{\sigma}(\boldsymbol{\Theta}) = \sqrt{\eta} \left(\boldsymbol{\Sigma}(\boldsymbol{\Theta})\right)^{\frac{1}{2}},$$

then we carry out the computation for $\mathcal{L}\left(\mathcal{L} \langle \nabla_{\boldsymbol{\Theta}} g(\boldsymbol{\Theta}_0), \boldsymbol{b} \rangle\right) (\boldsymbol{\Theta}_u)$,

$$\mathcal{L}\left(\mathcal{L} \langle \nabla_{\boldsymbol{\Theta}} g(\boldsymbol{\Theta}_0), \boldsymbol{b} \rangle\right) (\boldsymbol{\Theta}_u)$$

$$= \mathcal{L}\left(\langle \boldsymbol{b}, \nabla_{\boldsymbol{\Theta}} \left(\langle \nabla_{\boldsymbol{\Theta}} g(\boldsymbol{\Theta}_0), \boldsymbol{b} \rangle\right)\rangle\right) (\boldsymbol{\Theta}_u) + \mathcal{L}\left(\frac{\eta}{2} \boldsymbol{\Sigma} : \nabla_{\boldsymbol{\Theta}}^2 \left(\langle \nabla_{\boldsymbol{\Theta}} g(\boldsymbol{\Theta}_0), \boldsymbol{b} \rangle\right)\right) (\boldsymbol{\Theta}_u)$$

$$= \langle \boldsymbol{b}, \nabla_{\boldsymbol{\Theta}} \left(\langle \boldsymbol{b}, \nabla_{\boldsymbol{\Theta}} \left(\langle \nabla_{\boldsymbol{\Theta}} g(\boldsymbol{\Theta}_0), \boldsymbol{b} \rangle\right)\rangle\right)\rangle + \frac{\eta}{2} \boldsymbol{\Sigma} : \nabla_{\boldsymbol{\Theta}} \left(\langle \boldsymbol{b}, \nabla_{\boldsymbol{\Theta}}^2 \left(\langle \nabla_{\boldsymbol{\Theta}} g(\boldsymbol{\Theta}_0), \boldsymbol{b} \rangle\right)\rangle\right)$$

$$+ \frac{\eta}{2} \langle \boldsymbol{b}, \nabla_{\boldsymbol{\Theta}} \left(\boldsymbol{\Sigma} : \nabla_{\boldsymbol{\Theta}}^2 \left(\langle \nabla_{\boldsymbol{\Theta}} g(\boldsymbol{\Theta}_0), \boldsymbol{b} \rangle\right)\right)\rangle + \frac{\eta^2}{4} \boldsymbol{\Sigma} : \nabla_{\boldsymbol{\Theta}}^2 \left(\boldsymbol{\Sigma} : \nabla_{\boldsymbol{\Theta}}^2 \left(\langle \nabla_{\boldsymbol{\Theta}} g(\boldsymbol{\Theta}_0), \boldsymbol{b} \rangle\right)\right)$$

$$= \boldsymbol{b}^{\mathsf{T}} \nabla_{\boldsymbol{\Theta}} \left(\boldsymbol{b}^{\mathsf{T}} \nabla_{\boldsymbol{\Theta}} \boldsymbol{b} \nabla_{\boldsymbol{\Theta}} g(\boldsymbol{\Theta}_0)\right) (\boldsymbol{\Theta}_u) + \eta R_\eta(\boldsymbol{\Theta}_u)$$

$$= \left\langle \nabla_{\boldsymbol{\Theta}} L_{\mathcal{S}}(\boldsymbol{\Theta}_u), \nabla_{\boldsymbol{\Theta}} \left(\left\langle \frac{1}{2} \nabla_{\boldsymbol{\Theta}} \left(\|\nabla_{\boldsymbol{\Theta}} L_{\mathcal{S}}(\boldsymbol{\Theta}_u)\|_2^2\right), \nabla_{\boldsymbol{\Theta}} g(\boldsymbol{\Theta}_0) \right\rangle\right)\right\rangle + \eta R'_\eta(\boldsymbol{\Theta}_u),$$

since $\nabla_{\boldsymbol{\Theta}} L_{\mathcal{S}}(\boldsymbol{\Theta})$, $\nabla_{\boldsymbol{\Theta}}^2 L_{\mathcal{S}}(\boldsymbol{\Theta})$, $\nabla_{\boldsymbol{\Theta}}^3 L_{\mathcal{S}}(\boldsymbol{\Theta})$, $\boldsymbol{\Sigma}(\boldsymbol{\Theta})$, $R_\eta(\boldsymbol{\Theta}_u)$ and $R'_\eta(\boldsymbol{\Theta}_u)$ can be bounded above by the second, fourth and sixth moments of the solution to SDE (32). Moreover, we observe that

$$
\mathbb{E}\left[(\boldsymbol{\Theta}_\eta - \boldsymbol{\Theta}_0) \otimes (\boldsymbol{\Theta}_\eta - \boldsymbol{\Theta}_0) \otimes (\boldsymbol{\Theta}_\eta - \boldsymbol{\Theta}_0)\right]
$$
$$
=\mathbb{E}\left[\left(\int_0^\eta \boldsymbol{b}(\boldsymbol{\Theta}_s)\mathrm{d}s + \int_0^\eta \boldsymbol{\sigma}(\boldsymbol{\Theta}_s)\mathrm{d}\boldsymbol{W}_s\right) \otimes \left(\int_0^\eta \boldsymbol{b}(\boldsymbol{\Theta}_s)\mathrm{d}s + \int_0^\eta \boldsymbol{\sigma}(\boldsymbol{\Theta}_s)\mathrm{d}\boldsymbol{W}_s\right)\right.
$$
$$
\left. \otimes \left(\int_0^\eta \boldsymbol{b}(\boldsymbol{\Theta}_s)\mathrm{d}s + \int_0^\eta \boldsymbol{\sigma}(\boldsymbol{\Theta}_s)\mathrm{d}\boldsymbol{W}_s\right)\right],
$$

and its entry can be categorized into four types. The first one is the pure drift part, i.e.,

$$
\int_0^\eta \boldsymbol{b}(\boldsymbol{\Theta}_s)\mathrm{d}s \otimes \int_0^\eta \boldsymbol{b}(\boldsymbol{\Theta}_s)\mathrm{d}s \otimes \int_0^\eta \boldsymbol{b}(\boldsymbol{\Theta}_s)\mathrm{d}s,
$$

then by application of the mean value theorem and the fact that $\nabla_{\boldsymbol{\Theta}} L_{\mathcal{S}}(\boldsymbol{\Theta})$, $\nabla_{\boldsymbol{\Theta}}^2 L_{\mathcal{S}}(\boldsymbol{\Theta})$, $\nabla_{\boldsymbol{\Theta}}^3 L_{\mathcal{S}}(\boldsymbol{\Theta})$, and $\boldsymbol{\Sigma}(\boldsymbol{\Theta})$ can be bounded above by the second, fourth and sixth moments of the solution to SDE (32), we obtain that

$$
\mathbb{E}\int_0^\eta \boldsymbol{b}(\boldsymbol{\Theta}_s)\mathrm{d}s \otimes \int_0^\eta \boldsymbol{b}(\boldsymbol{\Theta}_s)\mathrm{d}s \otimes \int_0^\eta \boldsymbol{b}(\boldsymbol{\Theta}_s)\mathrm{d}s
$$
$$
=\eta^3 \mathbb{E}\boldsymbol{b}(\widetilde{\boldsymbol{\Theta}}_s) \otimes \boldsymbol{b}(\widetilde{\boldsymbol{\Theta}}_s) \otimes \boldsymbol{b}(\widetilde{\boldsymbol{\Theta}}_s) = \mathcal{O}(\eta^3).
$$

The second one is the pure noise part, i.e.,

$$
\left(\int_0^\eta \boldsymbol{\sigma}(\boldsymbol{\Theta}_s)\mathrm{d}\boldsymbol{W}_s\right) \otimes \left(\int_0^\eta \boldsymbol{\sigma}(\boldsymbol{\Theta}_s)\mathrm{d}\boldsymbol{W}_s\right) \otimes \left(\int_0^\eta \boldsymbol{\sigma}(\boldsymbol{\Theta}_s)\mathrm{d}\boldsymbol{W}_s\right),
$$

and as the odd moments of zero mean Gaussian variables are zero, hence we have

$$
\mathbb{E}\left[\left(\int_0^\eta \boldsymbol{\sigma}(\boldsymbol{\Theta}_s)\mathrm{d}\boldsymbol{W}_s\right) \otimes \left(\int_0^\eta \boldsymbol{\sigma}(\boldsymbol{\Theta}_s)\mathrm{d}\boldsymbol{W}_s\right) \otimes \left(\int_0^\eta \boldsymbol{\sigma}(\boldsymbol{\Theta}_s)\mathrm{d}\boldsymbol{W}_s\right)\right] = \boldsymbol{0},
$$

the third and fourth one are both of the mixed part, for the third one

$$
\int_0^\eta \boldsymbol{b}(\boldsymbol{\Theta}_s)\mathrm{d}s \otimes \int_0^\eta \boldsymbol{b}(\boldsymbol{\Theta}_s)\mathrm{d}s \otimes \left(\int_0^\eta \boldsymbol{\sigma}(\boldsymbol{\Theta}_s)\mathrm{d}\boldsymbol{W}_s\right),
$$

whose expectation is of course zero since the drift part and the noise part is independent, and the fact the odd moments of zero mean Gaussian variables are zero, and for the fourth one

$$
\int_0^\eta \boldsymbol{b}(\boldsymbol{\Theta}_s)\mathrm{d}s \otimes \left(\int_0^\eta \boldsymbol{\sigma}(\boldsymbol{\Theta}_s)\mathrm{d}\boldsymbol{W}_s\right) \otimes \left(\int_0^\eta \boldsymbol{\sigma}(\boldsymbol{\Theta}_s)\mathrm{d}\boldsymbol{W}_s\right),
$$

we obtain that

$$
\mathbb{E}\left[\int_0^\eta \boldsymbol{b}(\boldsymbol{\Theta}_s)\mathrm{d}s \otimes \left(\int_0^\eta \boldsymbol{\sigma}(\boldsymbol{\Theta}_s)\mathrm{d}\boldsymbol{W}_s\right) \otimes \left(\int_0^\eta \boldsymbol{\sigma}(\boldsymbol{\Theta}_s)\mathrm{d}\boldsymbol{W}_s\right)\right]
$$
$$
=\eta \mathbb{E}\boldsymbol{b}(\widetilde{\boldsymbol{\Theta}}_s) \otimes \mathbb{E}\left[\left(\int_0^\eta \boldsymbol{\sigma}(\boldsymbol{\Theta}_s)\mathrm{d}\boldsymbol{W}_s\right) \otimes \left(\int_0^\eta \boldsymbol{\sigma}(\boldsymbol{\Theta}_s)\mathrm{d}\boldsymbol{W}_s\right)\right] = \mathcal{O}(\eta^3).
$$

As we denote

$$
\bar{R}^3(\boldsymbol{\Theta}_0) := \mathbb{E}\left[(\boldsymbol{\Theta}_\eta - \boldsymbol{\Theta}_0) \otimes (\boldsymbol{\Theta}_\eta - \boldsymbol{\Theta}_0) \otimes (\boldsymbol{\Theta}_\eta - \boldsymbol{\Theta}_0)\right],
$$

then we obtain that

$$
\left\|\mathrm{vec}(\bar{R}^3(\boldsymbol{\Theta}_0))\right\|_2 \leq \eta^3 C(T^*, \boldsymbol{\Theta}_0).
$$

Hence we may apply the mean value theorem to (50) and obtain that

$$
\begin{aligned}
\left| \bar{E}_\eta^2(\mathbf{\Theta}_0) \right| = \Bigg| & \int_0^\eta \int_0^s v \mathcal{L}\left( \mathcal{L}\left\langle \nabla_{\mathbf{\Theta}} g(\mathbf{\Theta}_0), \boldsymbol{b} \right\rangle \right)(\widetilde{\mathbf{\Theta}}_u) \mathrm{d}v \mathrm{d}s \\
& + \frac{1}{6} \nabla_{\mathbf{\Theta}}^3 g(\mathbf{\Theta}_0) : \mathbb{E}\left[ (\mathbf{\Theta}_\eta - \mathbf{\Theta}_0) \otimes (\mathbf{\Theta}_\eta - \mathbf{\Theta}_0) \otimes (\mathbf{\Theta}_\eta - \mathbf{\Theta}_0) \right] \\
& + \frac{1}{24} \mathbb{E}\left[ \nabla_{\mathbf{\Theta}}^4 g(\widetilde{\mathbf{\Theta}}_0) : (\mathbf{\Theta}_\eta - \mathbf{\Theta}_0) \otimes (\mathbf{\Theta}_\eta - \mathbf{\Theta}_0) \otimes (\mathbf{\Theta}_\eta - \mathbf{\Theta}_0) \otimes (\mathbf{\Theta}_\eta - \mathbf{\Theta}_0) \right] \Bigg| \\
\leq & \int_0^\eta \int_0^s v \left\| g \right\|_{C^6} C(T^*, \mathbf{\Theta}_0) \mathrm{d}v \mathrm{d}s + \frac{1}{6} \left\| g \right\|_{C^6} \eta^3 C(T^*, \mathbf{\Theta}_0) \\
& + \frac{1}{24} \left\| g \right\|_{C^6} \left\| \mathbf{\Theta}_\eta - \mathbf{\Theta}_0 \right\|_2^4 \\
= & \frac{\eta^3}{6} \left\| g \right\|_{C^6} C(T^*, \mathbf{\Theta}_0) + \frac{1}{6} \left\| g \right\|_{C^6} \eta^3 C(T^*, \mathbf{\Theta}_0) \\
& + \frac{1}{24} \left\| g \right\|_{C^6} \mathbb{E} \left\| \int_0^\eta \boldsymbol{b}(\mathbf{\Theta}_s) \mathrm{d}s + \int_0^\eta \boldsymbol{\sigma}(\mathbf{\Theta}_s) \mathrm{d}\boldsymbol{W}_s \right\|_2^4 \\
\leq & \, \eta^3 \left\| g \right\|_{C^6} C(T^*, \mathbf{\Theta}_0) + \frac{1}{6} \left\| g \right\|_{C^6} \eta^3 C(T^*, \mathbf{\Theta}_0) \\
& + \frac{4}{24} \left\| g \right\|_{C^6} \eta^3 \mathbb{E} \left\| \nabla_{\mathbf{\Theta}} L_{\mathcal{S}}(\widetilde{\mathbf{\Theta}}_0) \right\|_2^2 + \frac{4}{24} \left\| g \right\|_{C^6} \mathbb{E} \left\| \int_0^\eta \boldsymbol{\sigma}(\mathbf{\Theta}_s) \mathrm{d}\boldsymbol{W}_s \right\|_2^4 \\
\leq & \, \eta^3 \left\| g \right\|_{C^6} C(T^*, \mathbf{\Theta}_0) + \frac{1}{6} \left\| g \right\|_{C^6} \eta^3 C(T^*, \mathbf{\Theta}_0) \\
& + \frac{4}{24} \left\| g \right\|_{C^6} \eta^3 \mathbb{E} \left\| \nabla_{\mathbf{\Theta}} L_{\mathcal{S}}(\widetilde{\mathbf{\Theta}}_0) \right\|_2^2 + \frac{C}{24} \left\| g \right\|_{C^6} \mathbb{E} \int_0^\eta \left\| \boldsymbol{\sigma}(\mathbf{\Theta}_s) \right\|_{\mathrm{F}}^4 \mathrm{d}s \\
\leq & \, \eta^3 \left\| g \right\|_{C^6} C(T^*, \mathbf{\Theta}_0) + \frac{1}{6} \left\| g \right\|_{C^6} \eta^3 C(T^*, \mathbf{\Theta}_0) \\
& + \eta^3 \left\| g \right\|_{C^6} \mathbb{E} \left\| \nabla_{\mathbf{\Theta}} L_{\mathcal{S}}(\widetilde{\mathbf{\Theta}}_0) \right\|_2^2 + C \left\| g \right\|_{C^6} \eta \mathbb{E} \left[ \eta^2 \left\| \mathbf{\Sigma}(\widetilde{\mathbf{\Theta}}_0) \right\|_{\mathrm{F}}^2 \right] \\
\leq & \, \eta^3 \left\| g \right\|_{C^6} C(T^*, \mathbf{\Theta}_0).
\end{aligned}
$$

We remark that for the last but third line we apply the Burkholder-Davis-Gundy inequality. To sum up for now,

$$
\begin{aligned}
\mathbb{E} g(\boldsymbol{\theta}_1) - \mathbb{E} g(\boldsymbol{\theta}_0) = & -\eta \left\langle \nabla_{\boldsymbol{\theta}} g(\boldsymbol{\theta}_0), \nabla_{\boldsymbol{\theta}} L_{\mathcal{S}}(\boldsymbol{\theta}) \big|_{\boldsymbol{\theta} = \boldsymbol{\theta}_0} \right\rangle \\
& + \frac{\eta^2}{2} \nabla_{\boldsymbol{\theta}}^2 g(\boldsymbol{\theta}_0) : \left( \nabla_{\boldsymbol{\theta}} L_{\mathcal{S}}(\boldsymbol{\theta}) \big|_{\boldsymbol{\theta} = \boldsymbol{\theta}_0} \otimes \nabla_{\boldsymbol{\theta}} L_{\mathcal{S}}(\boldsymbol{\theta}) \big|_{\boldsymbol{\theta} = \boldsymbol{\theta}_0} + \mathbf{\Sigma}(\boldsymbol{\theta}_0) \right) + E_\eta^2(\boldsymbol{\theta}_0),
\end{aligned}
$$

and

$$\mathbb{E}g(\boldsymbol{\Theta}_\eta) - \mathbb{E}g(\boldsymbol{\Theta}_0) = \eta \left\langle \nabla_{\boldsymbol{\Theta}} g(\boldsymbol{\Theta}_0), \mathbb{E}[\boldsymbol{b}(\boldsymbol{\Theta}_0)] \right\rangle + \frac{\eta^2}{2} \mathcal{L} \left\langle \nabla_{\boldsymbol{\Theta}} g(\boldsymbol{\Theta}_0), \boldsymbol{b} \right\rangle (\boldsymbol{\Theta}_0)$$

$$+ \frac{1}{2} \nabla_{\boldsymbol{\Theta}}^2 g(\boldsymbol{\Theta}_0) : \mathbb{E}\left[(\boldsymbol{\Theta}_\eta - \boldsymbol{\Theta}_0) \otimes (\boldsymbol{\Theta}_\eta - \boldsymbol{\Theta}_0)\right] + \bar{E}_\eta^2(\boldsymbol{\Theta}_0)$$

$$= \eta \left\langle \nabla_{\boldsymbol{\Theta}} g(\boldsymbol{\Theta}_0), \mathbb{E}[\boldsymbol{b}(\boldsymbol{\Theta}_0)] \right\rangle + \frac{\eta^2}{2} \mathcal{L} \left\langle \nabla_{\boldsymbol{\Theta}} g(\boldsymbol{\Theta}_0), \boldsymbol{b} \right\rangle (\boldsymbol{\Theta}_0)$$

$$+ \frac{1}{2} \nabla_{\boldsymbol{\Theta}}^2 g(\boldsymbol{\Theta}_0) : \mathbb{E}\left[\left(\int_0^\eta \boldsymbol{b}(\boldsymbol{\Theta}_s)\mathrm{d}s + \int_0^\eta \boldsymbol{\sigma}(\boldsymbol{\Theta}_s)\mathrm{d}\boldsymbol{W}_s\right)\right.$$

$$\left.\otimes \left(\int_0^\eta \boldsymbol{b}(\boldsymbol{\Theta}_s)\mathrm{d}s + \int_0^\eta \boldsymbol{\sigma}(\boldsymbol{\Theta}_s)\mathrm{d}\boldsymbol{W}_s\right)\right] + \bar{E}_\eta^2(\boldsymbol{\Theta}_0)$$

$$= \eta \left\langle \nabla_{\boldsymbol{\Theta}} g(\boldsymbol{\Theta}_0), \mathbb{E}[\boldsymbol{b}(\boldsymbol{\Theta}_0)] \right\rangle + \frac{\eta^2}{2} \mathcal{L} \left\langle \nabla_{\boldsymbol{\Theta}} g(\boldsymbol{\Theta}_0), \boldsymbol{b} \right\rangle (\boldsymbol{\Theta}_0)$$

$$+ \frac{1}{2} \nabla_{\boldsymbol{\Theta}}^2 g(\boldsymbol{\Theta}_0) : \mathbb{E}\left[\int_0^\eta \boldsymbol{b}(\boldsymbol{\Theta}_s)\mathrm{d}s \otimes \int_0^\eta \boldsymbol{b}(\boldsymbol{\Theta}_s)\mathrm{d}s\right]$$

$$+ \frac{1}{2} \nabla_{\boldsymbol{\Theta}}^2 g(\boldsymbol{\Theta}_0) : \mathbb{E}\left[\int_0^\eta \boldsymbol{\sigma}(\boldsymbol{\Theta}_s)\mathrm{d}\boldsymbol{W}_s \otimes \int_0^\eta \boldsymbol{\sigma}(\boldsymbol{\Theta}_s)\mathrm{d}\boldsymbol{W}_s\right] + \bar{E}_\eta^2(\boldsymbol{\Theta}_0)$$

$$= \eta \left\langle \nabla_{\boldsymbol{\Theta}} g(\boldsymbol{\Theta}_0), \mathbb{E}[\boldsymbol{b}(\boldsymbol{\Theta}_0)] \right\rangle + \frac{\eta^2}{2} \mathcal{L} \left\langle \nabla_{\boldsymbol{\Theta}} g(\boldsymbol{\Theta}_0), \boldsymbol{b} \right\rangle (\boldsymbol{\Theta}_0)$$

$$+ \frac{1}{2} \nabla_{\boldsymbol{\Theta}}^2 g(\boldsymbol{\Theta}_0) : \mathbb{E}\left[\int_0^\eta \int_0^\eta \boldsymbol{b}(\boldsymbol{\Theta}_s) \otimes \boldsymbol{b}(\boldsymbol{\Theta}_u)\mathrm{d}s\mathrm{d}u\right]$$

$$+ \frac{1}{2} \nabla_{\boldsymbol{\Theta}}^2 g(\boldsymbol{\Theta}_0) : \mathbb{E}\left[\int_0^\eta \boldsymbol{\sigma}(\boldsymbol{\Theta}_s)\mathrm{d}\boldsymbol{W}_s \otimes \int_0^\eta \boldsymbol{\sigma}(\boldsymbol{\Theta}_s)\mathrm{d}\boldsymbol{W}_s\right] + \bar{E}_\eta^2(\boldsymbol{\Theta}_0),$$

we observe that

$$\frac{1}{2} \nabla_{\boldsymbol{\Theta}}^2 g(\boldsymbol{\Theta}_0) : \mathbb{E}\left[\int_0^\eta \boldsymbol{\sigma}(\boldsymbol{\Theta}_s)\mathrm{d}\boldsymbol{W}_s \otimes \int_0^\eta \boldsymbol{\sigma}(\boldsymbol{\Theta}_s)\mathrm{d}\boldsymbol{W}_s\right]$$

$$= \mathbb{E}\left[\int_0^\eta \frac{1}{2} \nabla_{\boldsymbol{\Theta}}^2 g(\boldsymbol{\Theta}_0) : \boldsymbol{\sigma}\boldsymbol{\sigma}^\intercal(\boldsymbol{\Theta}_s)\mathrm{d}s\right]$$

$$= \frac{\eta}{2} \mathbb{E}\left[\int_0^\eta \nabla_{\boldsymbol{\Theta}}^2 g(\boldsymbol{\Theta}_0) : \boldsymbol{\Sigma}(\boldsymbol{\Theta}_s)\mathrm{d}s\right],$$

thus

$$\mathbb{E}g(\boldsymbol{\Theta}_\eta) - \mathbb{E}g(\boldsymbol{\Theta}_0) = \eta \left\langle \nabla_{\boldsymbol{\Theta}} g(\boldsymbol{\Theta}_0), \mathbb{E}[\boldsymbol{b}(\boldsymbol{\Theta}_0)] \right\rangle + \frac{\eta^2}{2} \mathcal{L} \left\langle \nabla_{\boldsymbol{\Theta}} g(\boldsymbol{\Theta}_0), \boldsymbol{b} \right\rangle (\boldsymbol{\Theta}_0)$$

$$+ \frac{1}{2} \nabla_{\boldsymbol{\Theta}}^2 g(\boldsymbol{\Theta}_0) : \mathbb{E}\left[\int_0^\eta \int_0^\eta \boldsymbol{b}(\boldsymbol{\Theta}_s) \otimes \boldsymbol{b}(\boldsymbol{\Theta}_u)\mathrm{d}s\mathrm{d}u\right]$$

$$+ \frac{\eta}{2} \mathbb{E}\left[\int_0^\eta \nabla_{\boldsymbol{\Theta}}^2 g(\boldsymbol{\Theta}_0) : \boldsymbol{\Sigma}(\boldsymbol{\Theta}_s)\mathrm{d}s\right] + \bar{E}_\eta^2(\boldsymbol{\Theta}_0).$$

Since

$$\nabla_{\boldsymbol{\Theta}}^2 g(\boldsymbol{\Theta}_0) : \mathbb{E}\left[\boldsymbol{b}(\boldsymbol{\Theta}_s) \otimes \boldsymbol{b}(\boldsymbol{\Theta}_u)\right]$$

$$= \nabla_{\boldsymbol{\Theta}}^2 g(\boldsymbol{\Theta}_0) : \mathbb{E}[\boldsymbol{b}(\boldsymbol{\Theta}_s) \otimes \boldsymbol{b}(\boldsymbol{\Theta}_0)] + \int_0^u \mathcal{L}\left(\nabla_{\boldsymbol{\Theta}}^2 g(\boldsymbol{\Theta}_0) : \boldsymbol{b}(\boldsymbol{\Theta}_s) \otimes \boldsymbol{b}(\boldsymbol{\Theta}_v)\right)\mathrm{d}v$$

$$= \nabla_{\boldsymbol{\Theta}}^2 g(\boldsymbol{\Theta}_0) : \mathbb{E}[\boldsymbol{b}(\boldsymbol{\Theta}_0) \otimes \boldsymbol{b}(\boldsymbol{\Theta}_0)] + \int_0^s \mathcal{L}\left(\nabla_{\boldsymbol{\Theta}}^2 g(\boldsymbol{\Theta}_0) : \boldsymbol{b}(\boldsymbol{\Theta}_w) \otimes \boldsymbol{b}(\boldsymbol{\Theta}_0)\right)\mathrm{d}w$$

$$+ \int_0^u \mathcal{L}\left(\nabla_{\boldsymbol{\Theta}}^2 g(\boldsymbol{\Theta}_0) : \boldsymbol{b}(\boldsymbol{\Theta}_s) \otimes \boldsymbol{b}(\boldsymbol{\Theta}_v)\right)\mathrm{d}v,$$

and since

$$\nabla_{\boldsymbol{\Theta}}^2 g(\boldsymbol{\Theta}_0) : \mathbb{E}\left[\boldsymbol{\Sigma}(\boldsymbol{\Theta}_s)\right]$$
$$= \nabla_{\boldsymbol{\Theta}}^2 g(\boldsymbol{\Theta}_0) : \mathbb{E}\left[\boldsymbol{\Sigma}(\boldsymbol{\Theta}_0)\right] + \int_0^s \mathcal{L}\left(\nabla_{\boldsymbol{\Theta}}^2 g(\boldsymbol{\Theta}_0) : \boldsymbol{\Sigma}(\boldsymbol{\Theta}_s)\right) \mathrm{d}v,$$

we are one step away to finish our proof,

$$\mathbb{E}g(\boldsymbol{\Theta}_\eta) - \mathbb{E}g(\boldsymbol{\Theta}_0) = \eta \left\langle \nabla_{\boldsymbol{\Theta}} g(\boldsymbol{\Theta}_0), \mathbb{E}[\boldsymbol{b}(\boldsymbol{\Theta}_0)]\right\rangle + \frac{\eta^2}{2}\mathcal{L}\left\langle \nabla_{\boldsymbol{\Theta}} g(\boldsymbol{\Theta}_0), \boldsymbol{b}\right\rangle(\boldsymbol{\Theta}_0)$$
$$+ \frac{1}{2}\nabla_{\boldsymbol{\Theta}}^2 g(\boldsymbol{\Theta}_0) : \mathbb{E}\left[\int_0^\eta \int_0^\eta \boldsymbol{b}(\boldsymbol{\Theta}_0) \otimes \boldsymbol{b}(\boldsymbol{\Theta}_0)\mathrm{d}s\mathrm{d}u\right]$$
$$+ \frac{\eta}{2}\mathbb{E}\left[\int_0^\eta \nabla_{\boldsymbol{\Theta}}^2 g(\boldsymbol{\Theta}_0) : \boldsymbol{\Sigma}(\boldsymbol{\Theta}_0)\mathrm{d}s\right] + \bar{E}_\eta^2(\boldsymbol{\Theta}_0),$$

where we misuse our notations for $\bar{E}_\eta^2(\boldsymbol{\Theta}_0)$, and the term

$$\int_0^\eta \int_0^\eta \int_0^s \mathcal{L}\left(\nabla_{\boldsymbol{\Theta}}^2 g(\boldsymbol{\Theta}_0) : \boldsymbol{b}(\boldsymbol{\Theta}_w) \otimes \boldsymbol{b}(\boldsymbol{\Theta}_0)\right)\mathrm{d}w\mathrm{d}s\mathrm{d}u$$
$$+ \int_0^\eta \int_0^\eta \int_0^u \mathcal{L}\left(\nabla_{\boldsymbol{\Theta}}^2 g(\boldsymbol{\Theta}_0) : \boldsymbol{b}(\boldsymbol{\Theta}_s) \otimes \boldsymbol{b}(\boldsymbol{\Theta}_v)\right)\mathrm{d}v\mathrm{d}s\mathrm{d}u$$
$$+ \int_0^\eta \int_0^s \mathcal{L}\left(\nabla_{\boldsymbol{\Theta}}^2 g(\boldsymbol{\Theta}_0) : \boldsymbol{\Sigma}(\boldsymbol{\Theta}_s)\right)\mathrm{d}v\mathrm{d}s,$$

is included, and $\bar{E}_\eta^2(\boldsymbol{\Theta}_0)$ is still of order $\mathcal{O}(\eta^3)$ by similar reasoning and we omit its demonstration. Thus

$$\mathbb{E}g(\boldsymbol{\Theta}_\eta) - \mathbb{E}g(\boldsymbol{\Theta}_0) = \eta \left\langle \nabla_{\boldsymbol{\Theta}} g(\boldsymbol{\Theta}_0), \mathbb{E}[\boldsymbol{b}(\boldsymbol{\Theta}_0)]\right\rangle + \frac{\eta^2}{2}\left\langle \boldsymbol{b}(\boldsymbol{\Theta}_0), \nabla_{\boldsymbol{\Theta}}\left\langle \nabla_{\boldsymbol{\Theta}} g(\boldsymbol{\Theta}_0), \boldsymbol{b}\right\rangle(\boldsymbol{\Theta}_0)\right\rangle$$
$$+ \frac{\eta^3}{2}\boldsymbol{\Sigma}(\boldsymbol{\Theta}_0) : \nabla_{\boldsymbol{\Theta}}^2\left\langle \nabla_{\boldsymbol{\Theta}} g(\boldsymbol{\Theta}_0), \boldsymbol{b}\right\rangle(\boldsymbol{\Theta}_0)$$
$$+ \frac{\eta^2}{2}\nabla_{\boldsymbol{\Theta}}^2 g(\boldsymbol{\Theta}_0) : \mathbb{E}\left[\boldsymbol{b}(\boldsymbol{\Theta}_0) \otimes \boldsymbol{b}(\boldsymbol{\Theta}_0)\right]$$
$$+ \frac{\eta^2}{2}\mathbb{E}\left[\nabla_{\boldsymbol{\Theta}}^2 g(\boldsymbol{\Theta}_0) : \boldsymbol{\Sigma}(\boldsymbol{\Theta}_0)\right] + \bar{E}_\eta^2(\boldsymbol{\Theta}_0),$$

and recall that since we choose

$$\boldsymbol{b}(\boldsymbol{\Theta}) = -\nabla_{\boldsymbol{\Theta}}\left(L_{\mathcal{S}}(\boldsymbol{\Theta}) + \frac{\eta}{4}\left\|\nabla_{\boldsymbol{\Theta}}L_{\mathcal{S}}(\boldsymbol{\Theta})\right\|_2^2\right),$$
$$\boldsymbol{\sigma}(\boldsymbol{\Theta}) = \sqrt{\eta}\left(\boldsymbol{\Sigma}(\boldsymbol{\Theta})\right)^{\frac{1}{2}},$$

then

$$\mathbb{E}g(\boldsymbol{\Theta}_\eta) - \mathbb{E}g(\boldsymbol{\Theta}_0) = -\eta \left\langle \nabla_{\boldsymbol{\Theta}} g(\boldsymbol{\Theta}_0), \nabla_{\boldsymbol{\Theta}}\left(L_{\mathcal{S}}(\boldsymbol{\Theta})\right)|_{\boldsymbol{\Theta}=\boldsymbol{\Theta}_0}\right\rangle$$
$$- \frac{\eta^2}{4}\left\langle \nabla_{\boldsymbol{\Theta}} g(\boldsymbol{\Theta}_0), \nabla_{\boldsymbol{\Theta}}\left(\left\|\nabla_{\boldsymbol{\Theta}}L_{\mathcal{S}}(\boldsymbol{\Theta})\right\|_2^2\right)|_{\boldsymbol{\Theta}=\boldsymbol{\Theta}_0}\right\rangle$$
$$+ \frac{\eta^2}{2}\left\langle \nabla_{\boldsymbol{\Theta}}\left(L_{\mathcal{S}}(\boldsymbol{\Theta})\right)|_{\boldsymbol{\Theta}=\boldsymbol{\Theta}_0}, \nabla_{\boldsymbol{\Theta}}\left\langle \nabla_{\boldsymbol{\Theta}} g(\boldsymbol{\Theta}_0), \nabla_{\boldsymbol{\Theta}}\left(L_{\mathcal{S}}(\boldsymbol{\Theta})\right)\right\rangle|_{\boldsymbol{\Theta}=\boldsymbol{\Theta}_0}\right\rangle$$
$$+ \frac{\eta^2}{2}\nabla_{\boldsymbol{\Theta}}^2 g(\boldsymbol{\Theta}_0) : \left(\nabla_{\boldsymbol{\Theta}}\left(L_{\mathcal{S}}(\boldsymbol{\Theta})\right)|_{\boldsymbol{\Theta}=\boldsymbol{\Theta}_0}\right) \otimes \nabla_{\boldsymbol{\Theta}}\left(L_{\mathcal{S}}(\boldsymbol{\Theta})\right)|_{\boldsymbol{\Theta}=\boldsymbol{\Theta}_0}$$
$$+ \frac{\eta^2}{2}\nabla_{\boldsymbol{\Theta}}^2 g(\boldsymbol{\Theta}_0) : \boldsymbol{\Sigma}(\boldsymbol{\Theta}_0) + \bar{E}_\eta^2(\boldsymbol{\Theta}_0)$$
$$= -\eta \left\langle \nabla_{\boldsymbol{\Theta}} g(\boldsymbol{\Theta}_0), \nabla_{\boldsymbol{\Theta}}\left(L_{\mathcal{S}}(\boldsymbol{\Theta})\right)|_{\boldsymbol{\Theta}=\boldsymbol{\Theta}_0}\right\rangle$$
$$+ \frac{\eta^2}{2}\nabla_{\boldsymbol{\Theta}}^2 g(\boldsymbol{\Theta}_0) : \left(\nabla_{\boldsymbol{\Theta}}\left(L_{\mathcal{S}}(\boldsymbol{\Theta})\right)|_{\boldsymbol{\Theta}=\boldsymbol{\Theta}_0}\right) \otimes \nabla_{\boldsymbol{\Theta}}\left(L_{\mathcal{S}}(\boldsymbol{\Theta})\right)|_{\boldsymbol{\Theta}=\boldsymbol{\Theta}_0}$$
$$+ \frac{\eta^2}{2}\nabla_{\boldsymbol{\Theta}}^2 g(\boldsymbol{\Theta}_0) : \boldsymbol{\Sigma}(\boldsymbol{\Theta}_0) + \bar{E}_\eta^2(\boldsymbol{\Theta}_0),$$

thus, we have

$$
\begin{aligned}
|\mathbb{E}g(\boldsymbol{\theta}_1) - \mathbb{E}g(\boldsymbol{\Theta}_\eta)| = \Big| & \mathbb{E}g(\boldsymbol{\theta}_0) - \eta \left\langle \nabla_{\boldsymbol{\theta}} g(\boldsymbol{\theta}_0), \nabla_{\boldsymbol{\theta}} L_{\mathcal{S}}(\boldsymbol{\theta})\big|_{\boldsymbol{\theta}=\boldsymbol{\theta}_0} \right\rangle \\
& + \frac{\eta^2}{2} \nabla^2_{\boldsymbol{\theta}} g(\boldsymbol{\theta}_0) : \left( \nabla_{\boldsymbol{\theta}} L_{\mathcal{S}}(\boldsymbol{\theta})\big|_{\boldsymbol{\theta}=\boldsymbol{\theta}_0} \otimes \nabla_{\boldsymbol{\theta}} L_{\mathcal{S}}(\boldsymbol{\theta})\big|_{\boldsymbol{\theta}=\boldsymbol{\theta}_0} + \boldsymbol{\Sigma}(\boldsymbol{\theta}_0) \right) \\
& + E^2_\eta(\boldsymbol{\theta}_0) \\
& - \mathbb{E}g(\boldsymbol{\Theta}_0) + \eta \left\langle \nabla_{\boldsymbol{\Theta}} g(\boldsymbol{\Theta}_0), \nabla_{\boldsymbol{\Theta}} \left( L_{\mathcal{S}}(\boldsymbol{\Theta}) \right) |_{\boldsymbol{\Theta}=\boldsymbol{\Theta}_0} \right\rangle \\
& - \frac{\eta^2}{2} \nabla^2_{\boldsymbol{\Theta}} g(\boldsymbol{\Theta}_0) : \left( \nabla_{\boldsymbol{\Theta}} \left( L_{\mathcal{S}}(\boldsymbol{\Theta}) \right) |_{\boldsymbol{\Theta}=\boldsymbol{\Theta}_0} \right) \otimes \nabla_{\boldsymbol{\Theta}} \left( L_{\mathcal{S}}(\boldsymbol{\Theta}) \right) |_{\boldsymbol{\Theta}=\boldsymbol{\Theta}_0}) \\
& - \frac{\eta^2}{2} \nabla^2_{\boldsymbol{\Theta}} g(\boldsymbol{\Theta}_0) : \boldsymbol{\Sigma}(\boldsymbol{\Theta}_0) + \bar{E}^2_\eta(\boldsymbol{\Theta}_0) \Big| \\
\leq & \left| E^2_\eta(\boldsymbol{\theta}_0) \right| + \left| \bar{E}^2_\eta(\boldsymbol{\Theta}_0) \right| \\
\leq & \eta^3 \left\| g \right\|_{C^6} C(T^*, \boldsymbol{\theta}_0, \eta_0) + \eta^3 \left\| g \right\|_{C^6} C(T^*, \boldsymbol{\Theta}_0) \\
= & \mathcal{O}(\eta^3).
\end{aligned}
$$

For the $N$-th step iteration, since

$$
|\mathbb{E}g(\boldsymbol{\theta}_N) - \mathbb{E}g(\boldsymbol{\Theta}_{\eta N})| = \left| \mathcal{P}^N_\eta g(\boldsymbol{\theta}_0) - \mathcal{P}_{\eta N} g(\boldsymbol{\Theta}_0) \right|,
$$

and the RHS of the above equation can be written into a telescoping sum as

$$
\mathcal{P}^N_\eta g(\boldsymbol{\theta}_0) - \mathcal{P}_{\eta N} g(\boldsymbol{\Theta}_0) = \sum_{l=1}^{N} \left( \mathcal{P}^{N-l+1}_\eta \circ \mathcal{P}_{(l-1)\eta} g(\boldsymbol{\theta}_0) - \mathcal{P}^{N-l}_\eta \circ \mathcal{P}_{l\eta} g(\boldsymbol{\Theta}_0) \right),
$$

hence by application of Proposition 1, we obtain that

$$
\begin{aligned}
|\mathbb{E}g(\boldsymbol{\theta}_N) - \mathbb{E}g(\boldsymbol{\Theta}_{\eta N})| & \leq \sum_{l=1}^{N} \left| \mathcal{P}^{N-l+1}_\eta \circ \mathcal{P}_{(l-1)\eta} g(\boldsymbol{\theta}_0) - \mathcal{P}^{N-l}_\eta \circ \mathcal{P}_{l\eta} g(\boldsymbol{\Theta}_0) \right| \\
& \leq \sum_{l=1}^{N} \left| \mathcal{P}^{N-l}_\eta \circ \left( \mathcal{P}^1_\eta \circ \mathcal{P}_{(l-1)\eta} - \mathcal{P}_\eta \circ \mathcal{P}_{(l-1)\eta} \right) g(\boldsymbol{\Theta}_0) \right|,
\end{aligned}
$$

since $\left( \mathcal{P}^1_\eta \circ \mathcal{P}_{(l-1)\eta} - \mathcal{P}_\eta \circ \mathcal{P}_{(l-1)\eta} \right) g(\boldsymbol{\Theta}_0)$ can be regarded as $\mathcal{L}^1(\mathbb{R}^D)$ if we choose measure $\mu$ to be the delta measure concentrated on $\boldsymbol{\Theta}_0$. i.e.,

$$
\mu := \delta_{\boldsymbol{\Theta}_0},
$$

hence by the conctration property of Markov operators, we obtain further that

$$
\begin{aligned}
|\mathbb{E}g(\boldsymbol{\theta}_N) - \mathbb{E}g(\boldsymbol{\Theta}_{\eta N})| & \leq \sum_{l=1}^{N} \left| \left( \mathcal{P}^1_\eta \circ \mathcal{P}_{(l-1)\eta} - \mathcal{P}_\eta \circ \mathcal{P}_{(l-1)\eta} \right) g(\boldsymbol{\Theta}_0) \right| \\
& \leq \sum_{l=1}^{N} \left| \mathcal{P}^1_\eta g(\boldsymbol{\Theta}_{(l-1)\eta}) - \mathcal{P}_\eta g(\boldsymbol{\Theta}_{(l-1)\eta}) \right|.
\end{aligned}
$$

By taking expectation conditioned on $\boldsymbol{\Theta}_{(l-1)\eta}$, then similar to the relation (46), the following holds

$$
\begin{aligned}
\left| \mathcal{P}^1_\eta g(\boldsymbol{\Theta}_{(l-1)\eta}) - \mathcal{P}_\eta g(\boldsymbol{\Theta}_{(l-1)\eta}) \right| & = \mathbb{E} \left[ \left[ |\mathbb{E}g(\boldsymbol{\theta}_l) - \mathbb{E}g(\boldsymbol{\Theta}_\eta l)| \Big| \boldsymbol{\Theta}_{(l-1)\eta} \right] \right] \\
& \leq \mathbb{E} \left| E^2_\eta(\boldsymbol{\Theta}_{(l-1)\eta}) \right| + \mathbb{E} \left| \bar{E}^2_\eta(\boldsymbol{\Theta}_{(l-1)\eta}) \right| \\
& \leq \eta^3 \left\| g \right\|_{C^6} C(T^*, \boldsymbol{\theta}_0, \eta_0) + \eta^3 \left\| g \right\|_{C^6} C(T^*, \boldsymbol{\Theta}_0) \\
& = \mathcal{O}(\eta^3).
\end{aligned}
$$

We remark that the last line of the above relation is essentially based on Assumption 2, since $\mathbb{E} \left| E^2_\eta(\boldsymbol{\Theta}_{(l-1)\eta}) \right|$ and $\mathbb{E} \left| \bar{E}^2_\eta(\boldsymbol{\Theta}_{(l-1)\eta}) \right|$ can be bounded above by the second, fourth and sixth moments of the solution to SDE (32), hence we may apply dominated convergence theorem to obtain the last line of the above relation.

To sum up, as

$$\left|\mathcal{P}_\eta^N g(\boldsymbol{\theta}_0) - \mathcal{P}_{\eta N} g(\boldsymbol{\Theta}_0)\right| \leq \sum_{l=1}^{N} \left|\mathcal{P}_\eta^{N-l+1} \circ \mathcal{P}_{(l-1)\eta} g(\boldsymbol{\theta}_0) - \mathcal{P}_\eta^{N-l} \circ \mathcal{P}_{l\eta} g(\boldsymbol{\Theta}_0)\right| = N\mathcal{O}(\eta^3),$$

hence for $N = N_{T,\eta}$,

$$\left|\mathcal{P}_\eta^N g(\boldsymbol{\theta}_0) - \mathcal{P}_{\eta N} g(\boldsymbol{\Theta}_0)\right| = N\mathcal{O}(\eta^3) = N\eta\mathcal{O}(\eta) \leq T\mathcal{O}(\eta^2) = \mathcal{O}(\eta^2).$$

$\square$

# I  VALIDATION FOR ASSUMPTION 1

In this section, we endeavor to demonstrate the validity of Assumption 1. We begin this section by making some estimates on the modified loss $L_{\mathcal{S}}$ and covariance $\mathbf{\Sigma}$.

## I.1  ESTIMATES ON MODIFIED LOSS AND COVARIANCE

For the modified loss, recall that $\boldsymbol{\theta} = \mathrm{vec}(\{\boldsymbol{q}_r\}_{r=1}^m) = \mathrm{vec}\left(\{(a_r, \boldsymbol{w}_r)\}_{r=1}^m\right)$, as we have

$$\nabla_{\boldsymbol{q}_k} L_{\mathcal{S}}(\boldsymbol{\Theta}) = \frac{1}{n}\sum_{i=1}^n e_i \nabla_{\boldsymbol{q}_k}\left(a_k \sigma(\boldsymbol{w}_k^\mathsf{T}\boldsymbol{x}_i)\right) + \frac{1-p}{np}\sum_{i=1}^n a_k \sigma(\boldsymbol{w}_k^\mathsf{T}\boldsymbol{x}_i)\nabla_{\boldsymbol{q}_k}\left(a_k \sigma(\boldsymbol{w}_k^\mathsf{T}\boldsymbol{x}_i)\right),$$

and under the usual convention that for all $i \in [n]$,

$$\frac{1}{c} \le \|\boldsymbol{x}_i\|_2, \quad |y_i| \le c,$$

where $c$ is some universal constant, and that $\sigma(0) = 0$, we obtain that

$$|e_i| = \left|\sum_{r=1}^m a_r \sigma(\boldsymbol{w}_r^\mathsf{T}\boldsymbol{x}_i) - y_i\right|$$

$$\le 1 + \sum_{r=1}^m |a_r|\,\|\boldsymbol{w}_r\|_2$$

$$\le 1 + \frac{1}{2}\sum_{r=1}^m \left(|a_r|^2 + \|\boldsymbol{w}_r\|_2^2\right)$$

$$\le 1 + \|\boldsymbol{\Theta}\|_2^2,$$

hence

$$\|\nabla_{\boldsymbol{q}_k} L_{\mathcal{S}}(\boldsymbol{\Theta})\|_2 \le \left(1 + \|\boldsymbol{\Theta}\|_2^2\right)\|\boldsymbol{q}_k\|_2 + \frac{1-p}{p}\|\boldsymbol{q}_k\|_2^3,$$

thus we have

$$\|\nabla_{\boldsymbol{\Theta}} L_{\mathcal{S}}(\boldsymbol{\Theta})\|_2 \le \left(1 + \|\boldsymbol{\Theta}\|_2^2\right)\|\boldsymbol{\Theta}\|_2 + \frac{1-p}{p}\|\boldsymbol{\Theta}\|_2^3$$

$$\le C_p(1 + \|\boldsymbol{\Theta}\|_2^3).$$

Moreover, since

$$\nabla_{\boldsymbol{\Theta}}^2 L_S(\boldsymbol{\Theta}) = \frac{1}{n}\sum_{i=1}^n \left(\nabla_{\boldsymbol{\Theta}} e_i \otimes \nabla_{\boldsymbol{\Theta}} e_i + e_i \nabla_{\boldsymbol{\Theta}}^2 e_i\right)$$

$$+ \frac{1-p}{np}\sum_{i=1}^n \mathrm{diag}\left\{\nabla_{\boldsymbol{q}_k}^2\left(a_k^2 \sigma(\boldsymbol{w}_k^\mathsf{T}\boldsymbol{x}_i)^2\right)\right\},$$

as we denote only for now $\times$ as matrix multiplication,

$$\nabla_{\boldsymbol{\Theta}}^2 L_S(\boldsymbol{\Theta})\nabla_{\boldsymbol{\Theta}} L_{\mathcal{S}}(\boldsymbol{\Theta})$$

$$= \left(\frac{1}{n}\sum_{i=1}^n \left(\nabla_{\boldsymbol{\Theta}} e_i \otimes \nabla_{\boldsymbol{\Theta}} e_i + e_i \nabla_{\boldsymbol{\Theta}}^2 e_i\right) + \frac{1-p}{np}\sum_{i=1}^n \mathrm{diag}\left\{\nabla_{\boldsymbol{q}_k}^2\left(a_k^2 \sigma(\boldsymbol{w}_k^\mathsf{T}\boldsymbol{x}_i)^2\right)\right\}\right)$$

$$\times \left(\frac{1}{n}\sum_{i=1}^n e_i \nabla_{\boldsymbol{\Theta}} e_i + \frac{1-p}{np}\sum_{i=1}^n \nabla_{\boldsymbol{\Theta}}\left(a_k^2 \sigma(\boldsymbol{w}_k^\mathsf{T}\boldsymbol{x}_i)^2\right)\right),$$

then the components in $\nabla_{\boldsymbol{\Theta}}^2 L_S(\boldsymbol{\Theta})\nabla_{\boldsymbol{\Theta}} L_{\mathcal{S}}(\boldsymbol{\Theta})$ can be categorized into six different types: Firstly,

$$\|\left(\nabla_{\boldsymbol{\Theta}} e_i \otimes \nabla_{\boldsymbol{\Theta}} e_i\right)e_j \nabla_{\boldsymbol{\Theta}} e_j\|_2$$

$$\le |e_j|\,\|\nabla_{\boldsymbol{\Theta}} e_i\|_2^2\,\|\nabla_{\boldsymbol{\Theta}} e_j\|_2$$

$$\le \left(1 + \|\boldsymbol{\Theta}\|_2^2\right)\|\boldsymbol{\Theta}\|_2^3$$

$$\le \left(1 + \|\boldsymbol{\Theta}\|_2^5\right).$$

Secondly,

$$\left\| \left( e_i \nabla^2_{\Theta} e_i \right) e_j \nabla_{\Theta} e_j \right\|_2$$
$$\leq \left( 1 + \|\Theta\|_2^2 \right)^2 \left\| \nabla^2_{\Theta} e_i \right\|_{2 \to 2} \left\| \nabla_{\Theta} e_j \right\|_2$$
$$\leq \left( 1 + \|\Theta\|_2^4 \right) \|\Theta\|_2^2$$
$$\leq \left( 1 + \|\Theta\|_2^6 \right).$$

Thirdly,

$$\left\| \left( \mathrm{diag}\left\{ \nabla^2_{\boldsymbol{q}_k} \left( a_k^2 \sigma(\boldsymbol{w}_k^\mathsf{T} \boldsymbol{x}_i)^2 \right) \right\} \right) e_j \nabla_{\Theta} e_j \right\|_2$$
$$\leq \left( 1 + \|\Theta\|_2^2 \right) \left\| \mathrm{diag}\left\{ \nabla^2_{\boldsymbol{q}_k} \left( a_k^2 \sigma(\boldsymbol{w}_k^\mathsf{T} \boldsymbol{x}_i)^2 \right) \right\} \right\|_{2 \to 2} \|\Theta\|_2$$
$$\leq \left( 1 + \|\Theta\|_2^2 \right) \left( 1 + \|\Theta\|_2^3 \right) \|\Theta\|_2$$
$$\leq \left( 1 + \|\Theta\|_2^6 \right).$$

Fourthly,

$$\left\| \left( \nabla_{\Theta} e_i \otimes \nabla_{\Theta} e_i \right) \nabla_{\Theta} \left( a_k^2 \sigma(\boldsymbol{w}_k^\mathsf{T} \boldsymbol{x}_j)^2 \right) \right\|_2$$
$$\leq \left\| \nabla_{\Theta} e_i \right\|_2^2 \|\Theta\|_2^3$$
$$\leq \left( 1 + \|\Theta\|_2^5 \right).$$

Fifthly,

$$\left\| \left( e_i \nabla^2_{\Theta} e_i \right) \nabla_{\Theta} \left( a_k^2 \sigma(\boldsymbol{w}_k^\mathsf{T} \boldsymbol{x}_j)^2 \right) \right\|_2$$
$$\leq \left( 1 + \|\Theta\|_2^2 \right) \left\| \nabla^2_{\Theta} e_i \right\|_{2 \to 2} \|\Theta\|_2^3$$
$$\leq \left( 1 + \|\Theta\|_2^2 \right) \|\Theta\|_2^4$$
$$\leq \left( 1 + \|\Theta\|_2^6 \right).$$

Finally,

$$\left\| \left( \mathrm{diag}\left\{ \nabla^2_{\boldsymbol{q}_k} \left( a_k^2 \sigma(\boldsymbol{w}_k^\mathsf{T} \boldsymbol{x}_i)^2 \right) \right\} \right) \nabla_{\Theta} \left( a_k^2 \sigma(\boldsymbol{w}_k^\mathsf{T} \boldsymbol{x}_j)^2 \right) \right\|_2$$
$$\leq \left\| \mathrm{diag}\left\{ \nabla^2_{\boldsymbol{q}_k} \left( a_k^2 \sigma(\boldsymbol{w}_k^\mathsf{T} \boldsymbol{x}_i)^2 \right) \right\} \right\|_{2 \to 2} \|\Theta\|_2^3$$
$$\leq \left( 1 + \|\Theta\|_2^3 \right) \|\Theta\|_2^3$$
$$\leq \left( 1 + \|\Theta\|_2^6 \right).$$

To sum up, for the drift term $\boldsymbol{b}(\Theta)$, regardless of the choice of first order or second order accuracy, we obtain that

$$\|\boldsymbol{b}(\Theta)\|_2 \leq 1 + \|\Theta\|_2^6.$$

As for the covariance $\Sigma$, recall that $\boldsymbol{\theta} = \mathrm{vec}(\{\boldsymbol{q}_r\}_{r=1}^m) = \mathrm{vec}\left( \{(a_r, \boldsymbol{w}_r)\}_{r=1}^m \right)$, then we obtain that the covariance $\Sigma$ reads

$$\Sigma = \begin{bmatrix} \Sigma_{11} & \Sigma_{12} & \cdots & \Sigma_{1m} \\ \Sigma_{21} & \Sigma_{22} & \cdots & \Sigma_{2m} \\ \vdots & \vdots & \vdots & \vdots \\ \Sigma_{m1} & \Sigma_{m2} & \cdots & \Sigma_{mm} \end{bmatrix}.$$

For each $k \in [m]$, we obtain that

$$
\boldsymbol{\Sigma}_{kk}(\boldsymbol{\Theta}) = \left(\frac{1}{p} - 1\right) \left(\frac{1}{n} \sum_{i=1}^{n} \left(e_{i,\backslash k} + \frac{1}{p} a_k \sigma(\boldsymbol{w}_k^\mathsf{T} \boldsymbol{x}_i)\right) \nabla_{\boldsymbol{q}_k} \left(a_k \sigma(\boldsymbol{w}_k^\mathsf{T} \boldsymbol{x}_i)\right)\right)
$$
$$
\otimes \left(\frac{1}{n} \sum_{i=1}^{n} \left(e_{i,\backslash k} + \frac{1}{p} a_k \sigma(\boldsymbol{w}_k^\mathsf{T} \boldsymbol{x}_i)\right) \nabla_{\boldsymbol{q}_k} \left(a_k \sigma(\boldsymbol{w}_k^\mathsf{T} \boldsymbol{x}_i)\right)\right)
$$
$$
+ \left(\frac{1}{p^2} - \frac{1}{p}\right) \sum_{l=1,l\neq k}^{m} \left(\frac{1}{n} \sum_{i=1}^{n} a_l \sigma(\boldsymbol{w}_l^\mathsf{T} \boldsymbol{x}_i) \nabla_{\boldsymbol{q}_k} \left(a_k \sigma(\boldsymbol{w}_k^\mathsf{T} \boldsymbol{x}_i)\right)\right)
$$
$$
\otimes \left(\frac{1}{n} \sum_{i=1}^{n} a_l \sigma(\boldsymbol{w}_l^\mathsf{T} \boldsymbol{x}_i) \nabla_{\boldsymbol{q}_k} \left(a_k \sigma(\boldsymbol{w}_k^\mathsf{T} \boldsymbol{x}_i)\right)\right),
$$

and for each $k, r \in [m]$ with $k \neq r$,

$$
\boldsymbol{\Sigma}_{kr}(\boldsymbol{\Theta}) = \left(\frac{1}{p} - 1\right) \left(\frac{1}{n} \sum_{i=1}^{n} \left(e_{i,\backslash k,\backslash r} + \frac{1}{p} a_k \sigma(\boldsymbol{w}_k^\mathsf{T} \boldsymbol{x}_i) + \frac{1}{p} a_r \sigma(\boldsymbol{w}_r^\mathsf{T} \boldsymbol{x}_i)\right) \nabla_{\boldsymbol{q}_k} \left(a_k \sigma(\boldsymbol{w}_k^\mathsf{T} \boldsymbol{x}_i)\right)\right)
$$
$$
\otimes \left(\frac{1}{n} \sum_{i=1}^{n} a_k \sigma(\boldsymbol{w}_k^\mathsf{T} \boldsymbol{x}_i) \nabla_{\boldsymbol{q}_r} \left(a_r \sigma(\boldsymbol{w}_r^\mathsf{T} \boldsymbol{x}_i)\right)\right)
$$
$$
+ \left(\frac{1}{p} - 1\right) \left(\frac{1}{n} \sum_{i=1}^{n} a_r \sigma(\boldsymbol{w}_r^\mathsf{T} \boldsymbol{x}_i) \nabla_{\boldsymbol{q}_k} \left(a_k \sigma(\boldsymbol{w}_k^\mathsf{T} \boldsymbol{x}_i)\right)\right)
$$
$$
\otimes \left(\frac{1}{n} \sum_{i=1}^{n} \left(e_{i,\backslash k,\backslash r} + a_k \sigma(\boldsymbol{w}_k^\mathsf{T} \boldsymbol{x}_i) + \frac{1}{p} a_r \sigma(\boldsymbol{w}_r^\mathsf{T} \boldsymbol{x}_i)\right) \nabla_{\boldsymbol{q}_r} \left(a_r \sigma(\boldsymbol{w}_r^\mathsf{T} \boldsymbol{x}_i)\right)\right),
$$

hence we obtain that

$$
\|\boldsymbol{\Sigma}_{kk}(\boldsymbol{\Theta})\|_{\mathrm{F}}^2 \leq C_p \left(\left|e_{i,\backslash k} + \frac{1}{p} a_k \sigma(\boldsymbol{w}_k^\mathsf{T} \boldsymbol{x}_i)\right|^2 + \sum_{l=1,l\neq k}^{m} a_l^2 \sigma(\boldsymbol{w}_l^\mathsf{T} \boldsymbol{x}_i)^2\right) \|\nabla_{\boldsymbol{\Theta}} e_i\|_2^2
$$
$$
\leq C_p (1 + \|\boldsymbol{\Theta}\|_2^2)^2 \|\boldsymbol{\Theta}\|_2^2
$$
$$
\leq (1 + \|\boldsymbol{\Theta}\|_2^6),
$$

and by similar reasoning

$$
\|\boldsymbol{\Sigma}_{kr}(\boldsymbol{\Theta})\|_{\mathrm{F}}^2 \leq (1 + \|\boldsymbol{\Theta}\|_2^6).
$$

## I.2 EXISTENCE, UNIQUENESS AND MOMENT ESTIMATES OF THE SOLUTION TO SDE

Existence of the solution to SDE (32) is proved by a truncation procedure: For each $M \geq 1$, define the truncation function

$$
\boldsymbol{b}_M(\boldsymbol{\Theta}) := \begin{cases} \boldsymbol{b}(\boldsymbol{\Theta}) & \text{if } \|\boldsymbol{\Theta}\|_2 \leq M, \\ \boldsymbol{b}(M \frac{\boldsymbol{\Theta}}{\|\boldsymbol{\Theta}\|_2}) & \text{if } \|\boldsymbol{\Theta}\|_2 > M. \end{cases}
$$

We also perform similar truncation to $\boldsymbol{\sigma}(\boldsymbol{\Theta})$ and obtain its truncation $\boldsymbol{\sigma}_M(\boldsymbol{\Theta})$. Then $\boldsymbol{b}_M$ and $\boldsymbol{\sigma}_M$ satisfy the Lipschitz condition and the linear growth condition, hence by application of the classical results (Oksendal, 2013, Theorem 5.2.1) in SDE, there exists a unique solution $\boldsymbol{\Theta}_M(\cdot)$ to the truncated SDE

$$
\mathrm{d}\boldsymbol{\Theta}_t = \boldsymbol{b}_M(\boldsymbol{\Theta}_t)\,\mathrm{d}t + \boldsymbol{\sigma}_M(\boldsymbol{\Theta}_t)\,\mathrm{d}\boldsymbol{W}_t, \quad \boldsymbol{\Theta}_0 = \boldsymbol{\Theta}(0). \tag{51}
$$

We may choose $M$ large enough, such that

$$
\|\boldsymbol{\Theta}_0\|_2 < M,
$$

and the solution to SDE (32) coincides with the solution to SDE (51) at least for a period of time $T^* > 0$ since $\|\boldsymbol{\Theta}_0\|_2 < M$. We remark that $T^*$ is the desired time in Assumption 2. We also remark that not only for any time $t \in [0, T^*]$, the second, fourth and sixth moments of the solution to SDE

(32) are uniformly bounded with respect to time $t$, but also that for any time $t \in [0, T^*]$, all moments of the solution to SDE (32) are uniformly bounded with respect to time $t$.

At this point, it is important to discuss that we prove is that for fixed time $T$, we can take the learning rate $\eta > 0$ small enough so that the SME is a good approximation of the distribution of the dropout iterates. What we did not prove is that for fixed $\eta$, the approximations hold for arbitrary time $T$. In particular, it is not hard to construct systems where for fixed $\eta$, both the SME and the asymptotic expansion fails when time $T$ is large enough.

### I.3 MOMENT ESTIMATES OF THE DROPOUT ITERATION

Recall that the dropout iteration reads

$$\boldsymbol{\theta}_N = \boldsymbol{\theta}_{N-1} - \eta \nabla_{\boldsymbol{\theta}} R_{\mathcal{S}}^{\mathrm{drop}}\left(\boldsymbol{\theta}_{N-1}; \boldsymbol{\delta}_N\right),$$

then we obtain that

$$\mathbb{E}\left\|\boldsymbol{\theta}_N\right\|_2^{2l} = \mathbb{E}\left\|\boldsymbol{\theta}_{N-1}\right\|_2^{2l} - 2l\eta\mathbb{E}\left[\left\|\boldsymbol{\theta}_{N-1}\right\|_2^{2l-2}\left\langle\boldsymbol{\theta}_{N-1}, \nabla_{\boldsymbol{\theta}} R_{\mathcal{S}}^{\mathrm{drop}}\left(\boldsymbol{\theta}_{N-1}; \boldsymbol{\delta}_N\right)\right\rangle\right] + \mathcal{O}(\eta^2),$$

then for learning rate $\eta$ small enough, we observe that $\{\mathbb{E}\left\|\boldsymbol{\theta}_N\right\|_2^{2l}\}_{N \geq 0}$ follows close to the trajectory of a ordinary differential equation (ODE). Moreover, from the estimates obtained in Section I.1,

$$\left\|\boldsymbol{\theta}_{N-1}\right\|_2^{2l-2}\left\langle\boldsymbol{\theta}_{N-1}, \nabla_{\boldsymbol{\theta}} R_{\mathcal{S}}^{\mathrm{drop}}\left(\boldsymbol{\theta}_{N-1}; \boldsymbol{\delta}_N\right)\right\rangle$$
$$\leq \left\|\boldsymbol{\theta}_{N-1}\right\|_2^{2l-1}\left\|\nabla_{\boldsymbol{\theta}} R_{\mathcal{S}}^{\mathrm{drop}}\left(\boldsymbol{\theta}_{N-1}; \boldsymbol{\delta}_N\right)\right\|_2$$
$$= \left\|\boldsymbol{\theta}_{N-1}\right\|_2^{2l-1}\left|e_i^N\right|\left\|\nabla_{\boldsymbol{\theta}} e_i^N\right\|_2$$
$$\leq \left\|\boldsymbol{\theta}_{N-1}\right\|_2^{2l-1} C_p(1 + \left\|\boldsymbol{\theta}_{N-1}\right\|_2^2)\left\|\boldsymbol{\theta}_{N-1}\right\|_2$$
$$\leq C_p(1 + \left\|\boldsymbol{\theta}_{N-1}\right\|_2^{2l+2}),$$

we remark that as the above estimates hold almost surely, then for learning rate $\eta$ small enough, we may apply Gronwall inequality to $\{\mathbb{E}\left\|\boldsymbol{\theta}_N\right\|_2^{2l}\}_{N \geq 0}$ and shows that for some $N^*$, all moments of the dropout iterations are uniformly bounded with respect to $N$, since for the ODE

$$\frac{\mathrm{d}u}{\mathrm{d}t} = 1 + u^{1+\lambda}, \quad u_0 := u(0), \tag{52}$$

with $\lambda > 0$. There exists time $T^* > 0$, such that for any time $t \in [0, T^*]$, its solution $\{u_t\}_{t \geq 0}$ is uniformly bounded with respect to time $t$. And since for small enough learning rate, all moments of the dropout iterations $\{\mathbb{E}\left\|\boldsymbol{\theta}_N\right\|_2^{2l}\}_{N \geq 0}$ follows close to the trajectory of ODEs of (52) type, hence all these moments are also uniformly bounded with respect to $N$.

## J  EXPLICIT FORM AND DERIVATION ON THE COVARIANCE

In this section, we present the expression for $\boldsymbol{\Sigma}$. As $\boldsymbol{\theta} = \text{vec}(\{\boldsymbol{q}_r\}_{r=1}^m) = \text{vec}\left(\{(a_r, \boldsymbol{w}_r)\}_{r=1}^m\right)$, then covariance of $\nabla_{\boldsymbol{\theta}} R_{\mathcal{S}}^{\text{drop}}\left(\boldsymbol{\theta}_{N-1}; \boldsymbol{\delta}_N\right)$ equals to $\boldsymbol{\Sigma}(\boldsymbol{\theta}_{N-1})$. We denote

$$\boldsymbol{\Sigma}_{kr}(\boldsymbol{\theta}_{N-1}) := \text{Cov}\left(\nabla_{\boldsymbol{q}_k} R_{\mathcal{S}}^{\text{drop}}\left(\boldsymbol{\theta}_{N-1}; \boldsymbol{\delta}_N\right), \nabla_{\boldsymbol{q}_r} R_{\mathcal{S}}^{\text{drop}}\left(\boldsymbol{\theta}_{N-1}; \boldsymbol{\delta}_N\right)\right),$$

then

$$\boldsymbol{\Sigma} = \left[\begin{array}{cccc} \boldsymbol{\Sigma}_{11} & \boldsymbol{\Sigma}_{12} & \cdots & \boldsymbol{\Sigma}_{1m} \\ \boldsymbol{\Sigma}_{21} & \boldsymbol{\Sigma}_{22} & \cdots & \boldsymbol{\Sigma}_{2m} \\ \vdots & \vdots & \vdots & \vdots \\ \boldsymbol{\Sigma}_{m1} & \boldsymbol{\Sigma}_{m2} & \cdots & \boldsymbol{\Sigma}_{mm} \end{array}\right].$$

For each $k \in [m]$, we obtain that

$$\begin{aligned} \boldsymbol{\Sigma}_{kk}(\boldsymbol{\theta}_{N-1}) =& \text{Cov}\left(\nabla_{\boldsymbol{q}_k} R_{\mathcal{S}}^{\text{drop}}\left(\boldsymbol{\theta}_{N-1}; \boldsymbol{\delta}_N\right), \nabla_{\boldsymbol{q}_k} R_{\mathcal{S}}^{\text{drop}}\left(\boldsymbol{\theta}_{N-1}; \boldsymbol{\delta}_N\right)\right) \\ =& \left(\frac{1}{p} - 1\right)\left(\frac{1}{n}\sum_{i=1}^n \left(e_{i,\backslash k} + \frac{1}{p}a_k\sigma(\boldsymbol{w}_k^{\mathsf{T}}\boldsymbol{x}_i)\right)\nabla_{\boldsymbol{q}_k}\left(a_k\sigma(\boldsymbol{w}_k^{\mathsf{T}}\boldsymbol{x}_i)\right)\right) \\ &\otimes \left(\frac{1}{n}\sum_{i=1}^n \left(e_{i,\backslash k} + \frac{1}{p}a_k\sigma(\boldsymbol{w}_k^{\mathsf{T}}\boldsymbol{x}_i)\right)\nabla_{\boldsymbol{q}_k}\left(a_k\sigma(\boldsymbol{w}_k^{\mathsf{T}}\boldsymbol{x}_i)\right)\right) \\ &+ \left(\frac{1}{p^2} - \frac{1}{p}\right)\sum_{k'=1,k'\neq k}^m \left(\frac{1}{n}\sum_{i=1}^n a_{k'}\sigma(\boldsymbol{w}_{k'}^{\mathsf{T}}\boldsymbol{x}_i)\nabla_{\boldsymbol{q}_k}\left(a_k\sigma(\boldsymbol{w}_k^{\mathsf{T}}\boldsymbol{x}_i)\right)\right) \\ &\otimes \left(\frac{1}{n}\sum_{i=1}^n a_{k'}\sigma(\boldsymbol{w}_{k'}^{\mathsf{T}}\boldsymbol{x}_i)\nabla_{\boldsymbol{q}_k}\left(a_k\sigma(\boldsymbol{w}_k^{\mathsf{T}}\boldsymbol{x}_i)\right)\right), \end{aligned}$$

where $e_{i,\backslash k} := e_{i,\backslash k}(\boldsymbol{\theta}) := \sum_{l=1,l\neq k}^m a_l\sigma(\boldsymbol{w}_l^{\mathsf{T}}\boldsymbol{x}_i) - y_i$, and for each $k, r \in [m]$ with $k \neq r$,

$$\begin{aligned} \boldsymbol{\Sigma}_{kr}(\boldsymbol{\theta}_{N-1}) =& \text{Cov}\left(\nabla_{\boldsymbol{q}_k} R_{\mathcal{S}}^{\text{drop}}\left(\boldsymbol{\theta}_{N-1}; \boldsymbol{\delta}_N\right), \nabla_{\boldsymbol{q}_r} R_{\mathcal{S}}^{\text{drop}}\left(\boldsymbol{\theta}_{N-1}; \boldsymbol{\delta}_N\right)\right) \\ =& \left(\frac{1}{p} - 1\right)\sum_{k'=1,k'\neq k,k'\neq r}^m \left(\frac{1}{n}\sum_{i=1}^n a_{k'}\sigma(\boldsymbol{w}_{k'}^{\mathsf{T}}\boldsymbol{x}_i)\nabla_{\boldsymbol{q}_k}\left(a_k\sigma(\boldsymbol{w}_k^{\mathsf{T}}\boldsymbol{x}_i)\right)\right) \\ &\otimes \left(\frac{1}{n}\sum_{i=1}^n a_{k'}\sigma(\boldsymbol{w}_{k'}^{\mathsf{T}}\boldsymbol{x}_i)\nabla_{\boldsymbol{q}_r}\left(a_r\sigma(\boldsymbol{w}_r^{\mathsf{T}}\boldsymbol{x}_i)\right)\right) \\ &+ \left(\frac{1}{p} - 1\right)\left(\frac{1}{n}\sum_{i=1}^n \left(e_{i,\backslash k,\backslash r} + \frac{1}{p}a_k\sigma(\boldsymbol{w}_k^{\mathsf{T}}\boldsymbol{x}_i) + \frac{1}{p}a_r\sigma(\boldsymbol{w}_r^{\mathsf{T}}\boldsymbol{x}_i)\right)\nabla_{\boldsymbol{q}_k}\left(a_k\sigma(\boldsymbol{w}_k^{\mathsf{T}}\boldsymbol{x}_i)\right)\right) \\ &\otimes \left(\frac{1}{n}\sum_{i=1}^n a_k\sigma(\boldsymbol{w}_k^{\mathsf{T}}\boldsymbol{x}_i)\nabla_{\boldsymbol{q}_r}\left(a_r\sigma(\boldsymbol{w}_r^{\mathsf{T}}\boldsymbol{x}_i)\right)\right) \\ &+ \left(\frac{1}{p} - 1\right)\left(\frac{1}{n}\sum_{i=1}^n a_r\sigma(\boldsymbol{w}_r^{\mathsf{T}}\boldsymbol{x}_i)\nabla_{\boldsymbol{q}_k}\left(a_k\sigma(\boldsymbol{w}_k^{\mathsf{T}}\boldsymbol{x}_i)\right)\right) \\ &\otimes \left(\frac{1}{n}\sum_{i=1}^n \left(e_{i,\backslash k,\backslash r} + a_k\sigma(\boldsymbol{w}_k^{\mathsf{T}}\boldsymbol{x}_i) + \frac{1}{p}a_r\sigma(\boldsymbol{w}_r^{\mathsf{T}}\boldsymbol{x}_i)\right)\nabla_{\boldsymbol{q}_r}\left(a_r\sigma(\boldsymbol{w}_r^{\mathsf{T}}\boldsymbol{x}_i)\right)\right), \end{aligned}$$

where $e_{i,\backslash k,\backslash r} := e_{i,\backslash k,\backslash r}(\boldsymbol{\theta}) := \sum_{l=1,l\neq k,l\neq r}^m a_l\sigma(\boldsymbol{w}_l^{\mathsf{T}}\boldsymbol{x}_i) - y_i$. We remark that such expression is consistent in that for the extreme case where $p = 1$, dropout 'degenerates' to GD, hence the covariance matrix degenerates to a zero matrix, i.e., $\boldsymbol{\Sigma} = \boldsymbol{0}_{D\times D}$. The following part is the specific derivation process of the covariance matrix.

### J.1 ELEMENTS ON THE DIAGONAL

In this part, we compute $\boldsymbol{\Sigma}_{kk}$ for all $k \in [m]$.

$$\boldsymbol{\Sigma}_{kk}(\boldsymbol{\theta}_{N-1}) = \mathrm{Cov}\left(\nabla_{\boldsymbol{q}_k} R_{\mathcal{S}}^{\mathrm{drop}}(\boldsymbol{\theta}_{N-1}; \boldsymbol{\delta}_N), \nabla_{\boldsymbol{q}_k} R_{\mathcal{S}}^{\mathrm{drop}}(\boldsymbol{\theta}_{N-1}; \boldsymbol{\delta}_N)\right)$$

$$= \frac{1}{n^2} \sum_{i,j=1}^{n} \mathrm{Cov}\left(e_i^N(\boldsymbol{\delta}_N)_k, e_j^N(\boldsymbol{\delta}_N)_k\right) \nabla_{\boldsymbol{q}_k}\left(a_k \sigma(\boldsymbol{w}_k^{\mathsf{T}} \boldsymbol{x}_i)\right) \otimes \nabla_{\boldsymbol{q}_k}\left(a_k \sigma(\boldsymbol{w}_k^{\mathsf{T}} \boldsymbol{x}_j)\right),$$

in order to compute $\mathrm{Cov}\left(e_i^N(\boldsymbol{\delta}_N)_k, e_j^N(\boldsymbol{\delta}_N)_k\right)$, we need to compute firstly $\mathbb{E}\left[e_i^N e_j^N(\boldsymbol{\delta}_N)_k^2\right]$, and since $\mathbb{E}\left[e_i^N e_j^N(\boldsymbol{\delta}_N)_k^2\right]$ consists of four parts, one of which is

$$\mathbb{E}\left[\left(\sum_{k'=1,k'\neq k}^{m} (\boldsymbol{\delta}_N)_{k'} a_{k'} \sigma(\boldsymbol{w}_{k'}^{\mathsf{T}} \boldsymbol{x}_i) - y_i\right)\left(\sum_{l=1,l\neq k}^{m} (\boldsymbol{\delta}_N)_l a_l \sigma(\boldsymbol{w}_l^{\mathsf{T}} \boldsymbol{x}_j) - y_j\right)(\boldsymbol{\delta}_N)_k^2\right]$$

$$= \mathbb{E}\left[\left(\sum_{k'=1,k'\neq k}^{m} (\boldsymbol{\delta}_N)_{k'} a_{k'} \sigma(\boldsymbol{w}_{k'}^{\mathsf{T}} \boldsymbol{x}_i) - y_i\right)\left(\sum_{l=1,l\neq k}^{m} (\boldsymbol{\delta}_N)_l a_l \sigma(\boldsymbol{w}_l^{\mathsf{T}} \boldsymbol{x}_j) - y_j\right)\right] \mathbb{E}\left[(\boldsymbol{\delta}_N)_k^2\right]$$

$$= \frac{1}{p}\left(\mathbb{E}\left[\sum_{k'=1,k'\neq k}^{m} (\boldsymbol{\delta}_N)_{k'}^2 a_{k'}^2 \sigma(\boldsymbol{w}_{k'}^{\mathsf{T}} \boldsymbol{x}_i)\sigma(\boldsymbol{w}_{k'}^{\mathsf{T}} \boldsymbol{x}_j)\right] + \mathbb{E}\left[\sum_{k'\neq l,\ k',l\neq k} (\boldsymbol{\delta}_N)_{k'}(\boldsymbol{\delta}_N)_l a_{k'} a_l \sigma(\boldsymbol{w}_{k'}^{\mathsf{T}} \boldsymbol{x}_i)\sigma(\boldsymbol{w}_l^{\mathsf{T}} \boldsymbol{x}_j)\right]\right.$$

$$\left. - y_i \mathbb{E}\left[\sum_{k'=1,k'\neq k}^{m} (\boldsymbol{\delta}_N)_{k'} a_{k'} \sigma(\boldsymbol{w}_{k'}^{\mathsf{T}} \boldsymbol{x}_j)\right] - y_j \mathbb{E}\left[\sum_{k'=1,k'\neq k}^{m} (\boldsymbol{\delta}_N)_{k'} a_{k'} \sigma(\boldsymbol{w}_{k'}^{\mathsf{T}} \boldsymbol{x}_i)\right] + y_i y_j\right)$$

$$= \frac{1}{p^2} \sum_{k'=1,k'\neq k}^{m} a_{k'}^2 \sigma(\boldsymbol{w}_{k'}^{\mathsf{T}} \boldsymbol{x}_i)\sigma(\boldsymbol{w}_{k'}^{\mathsf{T}} \boldsymbol{x}_j) + \frac{1}{p} \sum_{k'\neq l,\ k',l\neq k} a_{k'} a_l \sigma(\boldsymbol{w}_{k'}^{\mathsf{T}} \boldsymbol{x}_i)\sigma(\boldsymbol{w}_l^{\mathsf{T}} \boldsymbol{x}_j)$$

$$- \frac{y_i}{p} \sum_{k'=1,k'\neq k}^{m} a_{k'} \sigma(\boldsymbol{w}_{k'}^{\mathsf{T}} \boldsymbol{x}_j) - \frac{y_j}{p} \sum_{k'=1,k'\neq k}^{m} a_{k'} \sigma(\boldsymbol{w}_{k'}^{\mathsf{T}} \boldsymbol{x}_i) + \frac{y_i y_j}{p}$$

$$= \frac{1}{p}\left[\sum_{k'=1,k'\neq k}^{m} a_{k'} \sigma(\boldsymbol{w}_{k'}^{\mathsf{T}} \boldsymbol{x}_i) - y_i\right]\left[\sum_{k'=1,k'\neq k}^{m} a_{k'} \sigma(\boldsymbol{w}_{k'}^{\mathsf{T}} \boldsymbol{x}_j) - y_j\right]$$

$$+ \left(\frac{1}{p^2} - \frac{1}{p}\right)\left(\sum_{k'=1,k'\neq k}^{m} a_{k'}^2 \sigma(\boldsymbol{w}_{k'}^{\mathsf{T}} \boldsymbol{x}_i)\sigma(\boldsymbol{w}_{k'}^{\mathsf{T}} \boldsymbol{x}_j)\right),$$

and the second part reads

$$\mathbb{E}\left[(\boldsymbol{\delta}_N)_k a_k \sigma(\boldsymbol{w}_k^{\mathsf{T}} \boldsymbol{x}_i)\left(\sum_{l=1,l\neq k}^{m} (\boldsymbol{\delta}_N)_l a_l \sigma(\boldsymbol{w}_l^{\mathsf{T}} \boldsymbol{x}_j) - y_j\right)(\boldsymbol{\delta}_N)_k^2\right]$$

$$= \frac{a_k \sigma(\boldsymbol{w}_k^{\mathsf{T}} \boldsymbol{x}_i)}{p^2}\left(\sum_{k'=1,k'\neq k}^{m} a_{k'} \sigma(\boldsymbol{w}_{k'}^{\mathsf{T}} \boldsymbol{x}_j) - y_j\right),$$

and by symmetry, the third part reads

$$\mathbb{E}\left[(\boldsymbol{\delta}_N)_k a_k \sigma(\boldsymbol{w}_k^{\mathsf{T}} \boldsymbol{x}_j)\left(\sum_{l=1,l\neq k}^{m} (\boldsymbol{\delta}_N)_l a_l \sigma(\boldsymbol{w}_l^{\mathsf{T}} \boldsymbol{x}_i) - y_i\right)(\boldsymbol{\delta}_N)_k^2\right]$$

$$= \frac{a_k \sigma(\boldsymbol{w}_k^{\mathsf{T}} \boldsymbol{x}_j)}{p^2}\left(\sum_{k'=1,k'\neq k}^{m} a_{k'} \sigma(\boldsymbol{w}_{k'}^{\mathsf{T}} \boldsymbol{x}_i) - y_i\right),$$

and finally, the fourth part reads

$$\mathbb{E}\left[(\boldsymbol{\delta}_N)_k a_k \sigma(\boldsymbol{w}_k^\mathsf{T}\boldsymbol{x}_i)(\boldsymbol{\delta}_N)_k a_k \sigma(\boldsymbol{w}_k^\mathsf{T}\boldsymbol{x}_j)(\boldsymbol{\delta}_N)_k^2\right] = \frac{1}{p^3}a_k^2\sigma(\boldsymbol{w}_k^\mathsf{T}\boldsymbol{x}_i)\sigma(\boldsymbol{w}_k^\mathsf{T}\boldsymbol{x}_j).$$

To sum up,

$$\mathbb{E}\left[e_i^N e_j^N (\boldsymbol{\delta}_N)_k^2\right] = \left(\frac{1}{p^2} - \frac{1}{p}\right)\left(\sum_{k'=1,k'\neq k}^m a_{k'}^2 \sigma(\boldsymbol{w}_{k'}^\mathsf{T}\boldsymbol{x}_i)\sigma(\boldsymbol{w}_{k'}^\mathsf{T}\boldsymbol{x}_j)\right)$$
$$+ \frac{1}{p}e_{i,\backslash k}e_{j,\backslash k} + \frac{a_k\sigma(\boldsymbol{w}_k^\mathsf{T}\boldsymbol{x}_j)}{p^2}e_{i,\backslash k} + \frac{a_k\sigma(\boldsymbol{w}_k^\mathsf{T}\boldsymbol{x}_i)}{p^2}e_{j,\backslash k}$$
$$+ \frac{1}{p^3}a_k^2\sigma(\boldsymbol{w}_k^\mathsf{T}\boldsymbol{x}_i)\sigma(\boldsymbol{w}_k^\mathsf{T}\boldsymbol{x}_j),$$

and

$$\mathbb{E}\left[e_i^N(\boldsymbol{\delta}_N)_k\right]\mathbb{E}\left[e_j^N(\boldsymbol{\delta}_N)_k\right]$$
$$= \left(e_{i,\backslash k} + \frac{1}{p}a_k\sigma(\boldsymbol{w}_k^\mathsf{T}\boldsymbol{x}_i)\right)\left(e_{j,\backslash k} + \frac{1}{p}a_k\sigma(\boldsymbol{w}_k^\mathsf{T}\boldsymbol{x}_j)\right)$$
$$= e_{i,\backslash k}e_{j,\backslash k} + \frac{a_k\sigma(\boldsymbol{w}_k^\mathsf{T}\boldsymbol{x}_j)}{p}e_{i,\backslash k} + \frac{a_k\sigma(\boldsymbol{w}_k^\mathsf{T}\boldsymbol{x}_i)}{p}e_{j,\backslash k} + \frac{1}{p^2}a_k^2\sigma(\boldsymbol{w}_k^\mathsf{T}\boldsymbol{x}_i)\sigma(\boldsymbol{w}_k^\mathsf{T}\boldsymbol{x}_j),$$

hence

$$\mathrm{Cov}\left(e_i^N(\boldsymbol{\delta}_N)_k, e_j^N(\boldsymbol{\delta}_N)_k\right)$$
$$= \mathbb{E}\left[e_i^N e_j^N(\boldsymbol{\delta}_N)_k^2\right] - \mathbb{E}\left[e_i^N(\boldsymbol{\delta}_N)_k\right]\mathbb{E}\left[e_i^N(\boldsymbol{\delta}_N)_k\right]$$
$$= \left(\frac{1}{p^2} - \frac{1}{p}\right)\left(\sum_{k'=1,k'\neq k}^m a_{k'}^2\sigma(\boldsymbol{w}_{k'}^\mathsf{T}\boldsymbol{x}_i)\sigma(\boldsymbol{w}_{k'}^\mathsf{T}\boldsymbol{x}_j)\right)$$
$$+ \left(\frac{1}{p} - 1\right)e_{i,\backslash k}e_{j,\backslash k} + \left(\frac{1}{p^2} - \frac{1}{p}\right)a_k\sigma(\boldsymbol{w}_k^\mathsf{T}\boldsymbol{x}_i)e_{j,\backslash k}$$
$$+ \left(\frac{1}{p^2} - \frac{1}{p}\right)a_k\sigma(\boldsymbol{w}_k^\mathsf{T}\boldsymbol{x}_j)e_{i,\backslash k} + \left(\frac{1}{p^3} - \frac{1}{p^2}\right)a_k^2\sigma(\boldsymbol{w}_k^\mathsf{T}\boldsymbol{x}_i)\sigma(\boldsymbol{w}_k^\mathsf{T}\boldsymbol{x}_j)$$
$$= \left(\frac{1}{p} - 1\right)\mathbb{E}\left(e_i^N(\boldsymbol{\delta}_N)_k\right)\mathbb{E}\left(e_j^N(\boldsymbol{\delta}_N)_k\right) + \left(\frac{1}{p^2} - \frac{1}{p}\right)\left(\sum_{k'=1,k'\neq k}^m a_{k'}^2\sigma(\boldsymbol{w}_{k'}^\mathsf{T}\boldsymbol{x}_i)\sigma(\boldsymbol{w}_{k'}^\mathsf{T}\boldsymbol{x}_j)\right),$$

by summation over the indices $i$ and $j$, for each $k \in [m]$, the covariance matrix reads:

$$\boldsymbol{\Sigma}_{kk}(\boldsymbol{\theta}_{N-1}) = \mathrm{Cov}\left(\nabla_{\boldsymbol{q}_k}R_\mathcal{S}^{\mathrm{drop}}\left(\boldsymbol{\theta}_{N-1};\boldsymbol{\delta}_N\right), \nabla_{\boldsymbol{q}_k}R_\mathcal{S}^{\mathrm{drop}}\left(\boldsymbol{\theta}_{N-1};\boldsymbol{\delta}_N\right)\right)$$
$$= \left(\frac{1}{p} - 1\right)\left(\frac{1}{n}\sum_{i=1}^n\left(e_{i,\backslash k} + \frac{1}{p}a_k\sigma(\boldsymbol{w}_k^\mathsf{T}\boldsymbol{x}_i)\right)\nabla_{\boldsymbol{q}_k}\left(a_k\sigma(\boldsymbol{w}_k^\mathsf{T}\boldsymbol{x}_i)\right)\right)$$
$$\otimes\left(\frac{1}{n}\sum_{i=1}^n\left(e_{i,\backslash k} + \frac{1}{p}a_k\sigma(\boldsymbol{w}_k^\mathsf{T}\boldsymbol{x}_i)\right)\nabla_{\boldsymbol{q}_k}\left(a_k\sigma(\boldsymbol{w}_k^\mathsf{T}\boldsymbol{x}_i)\right)\right)$$
$$+ \left(\frac{1}{p^2} - \frac{1}{p}\right)\sum_{l=1,l\neq k}^m\left(\frac{1}{n}\sum_{i=1}^n a_l\sigma(\boldsymbol{w}_l^\mathsf{T}\boldsymbol{x}_i)\nabla_{\boldsymbol{q}_k}\left(a_k\sigma(\boldsymbol{w}_k^\mathsf{T}\boldsymbol{x}_i)\right)\right)$$
$$\otimes\left(\frac{1}{n}\sum_{i=1}^n a_l\sigma(\boldsymbol{w}_l^\mathsf{T}\boldsymbol{x}_i)\nabla_{\boldsymbol{q}_k}\left(a_k\sigma(\boldsymbol{w}_k^\mathsf{T}\boldsymbol{x}_i)\right)\right).$$

## J.2 ELEMENTS OFF THE DIAGONAL

In this part, we compute $\boldsymbol{\Sigma}_{kr}$ for all $k, r \in [m]$, where $k \neq r$.

$$\boldsymbol{\Sigma}_{kr}(\boldsymbol{\theta}_{N-1}) = \mathrm{Cov}\left(\nabla_{\boldsymbol{q}_k} R_{\mathcal{S}}^{\mathrm{drop}}(\boldsymbol{\theta}_{N-1}; \boldsymbol{\delta}_N), \nabla_{\boldsymbol{q}_r} R_{\mathcal{S}}^{\mathrm{drop}}(\boldsymbol{\theta}_{N-1}; \boldsymbol{\delta}_N)\right)$$

$$= \frac{1}{n^2} \sum_{i,j=1}^{n} \mathrm{Cov}\left(e_i^N(\boldsymbol{\delta}_N)_k, e_j^N(\boldsymbol{\delta}_N)_r\right) \nabla_{\boldsymbol{q}_k}\left(a_k \sigma(\boldsymbol{w}_k^{\mathsf{T}} \boldsymbol{x}_i)\right) \otimes \nabla_{\boldsymbol{q}_r}\left(a_k \sigma(\boldsymbol{w}_k^{\mathsf{T}} \boldsymbol{x}_j)\right),$$

in order to compute $\mathrm{Cov}\left(e_i^N(\boldsymbol{\delta}_N)_k, e_j^N(\boldsymbol{\delta}_N)_r\right)$, we need to compute firstly $\mathbb{E}\left[e_i^N e_j^N(\boldsymbol{\delta}_N)_k(\boldsymbol{\delta}_N)_r\right]$, and since $\mathbb{E}\left[e_i^N e_j^N(\boldsymbol{\delta}_N)_k(\boldsymbol{\delta}_N)_r\right]$ consists of nine parts, one of which is

$$\mathbb{E}\left[\left(\sum_{k'=1,k'\neq k,k'\neq r}^{m} (\boldsymbol{\delta}_N)_{k'} a_{k'} \sigma(\boldsymbol{w}_{k'}^{\mathsf{T}} \boldsymbol{x}_i) - y_i\right)\left(\sum_{l=1,l\neq k,l\neq r}^{m} (\boldsymbol{\delta}_N)_l a_l \sigma(\boldsymbol{w}_l^{\mathsf{T}} \boldsymbol{x}_j) - y_j\right)(\boldsymbol{\delta}_N)_k(\boldsymbol{\delta}_N)_r\right]$$

$$= \mathbb{E}\left[\left(\sum_{k'=1,k'\neq k,k'\neq r}^{m} (\boldsymbol{\delta}_N)_{k'} a_{k'} \sigma(\boldsymbol{w}_{k'}^{\mathsf{T}} \boldsymbol{x}_i) - y_i\right)\left(\sum_{l=1,l\neq k,l\neq r}^{m} (\boldsymbol{\delta}_N)_l a_l \sigma(\boldsymbol{w}_l^{\mathsf{T}} \boldsymbol{x}_j) - y_j\right)\right] \mathbb{E}\left[(\boldsymbol{\delta}_N)_k(\boldsymbol{\delta}_N)_r\right]$$

$$= \frac{1}{p} \sum_{k'=1,k'\neq k,k'\neq r}^{m} a_{k'}^2 \sigma(\boldsymbol{w}_{k'}^{\mathsf{T}} \boldsymbol{x}_i) \sigma(\boldsymbol{w}_{k'}^{\mathsf{T}} \boldsymbol{x}_j) + \sum_{k'\neq l \text{ and } k',l\neq k,r} a_{k'} a_l \sigma(\boldsymbol{w}_{k'}^{\mathsf{T}} \boldsymbol{x}_i) \sigma(\boldsymbol{w}_l^{\mathsf{T}} \boldsymbol{x}_j)$$

$$- y_i \sum_{k'=1,k'\neq k,k'\neq r}^{m} a_{k'} \sigma(\boldsymbol{w}_{k'}^{\mathsf{T}} \boldsymbol{x}_j) - y_j \sum_{k'=1,k'\neq k,k'\neq r}^{m} a_{k'} \sigma(\boldsymbol{w}_{k'}^{\mathsf{T}} \boldsymbol{x}_i) + y_i y_j$$

$$= \left[\sum_{k'=1,k'\neq k,k'\neq r}^{m} a_{k'} \sigma(\boldsymbol{w}_{k'}^{\mathsf{T}} \boldsymbol{x}_i) - y_i\right]\left[\sum_{k'=1,k'\neq k,k'\neq r}^{m} a_{k'} \sigma(\boldsymbol{w}_{k'}^{\mathsf{T}} \boldsymbol{x}_j) - y_j\right]$$

$$+ \left(\frac{1}{p} - 1\right)\left(\sum_{k'=1,k'\neq k,k'\neq r}^{m} a_{k'}^2 \sigma(\boldsymbol{w}_{k'}^{\mathsf{T}} \boldsymbol{x}_i) \sigma(\boldsymbol{w}_{k'}^{\mathsf{T}} \boldsymbol{x}_j)\right)$$

$$= e_{i,\backslash k,\backslash r} e_{j,\backslash k,\backslash r} + \left(\frac{1}{p} - 1\right)\left(\sum_{k'=1,k'\neq k,k'\neq r}^{m} a_{k'}^2 \sigma(\boldsymbol{w}_{k'}^{\mathsf{T}} \boldsymbol{x}_i) \sigma(\boldsymbol{w}_{k'}^{\mathsf{T}} \boldsymbol{x}_j)\right),$$

and the second part reads

$$\mathbb{E}\left[\left(\sum_{k'=1,k'\neq k,k'\neq r}^{m} (\boldsymbol{\delta}_N)_{k'} a_{k'} \sigma(\boldsymbol{w}_{k'}^{\mathsf{T}} \boldsymbol{x}_i) - y_i\right)(\boldsymbol{\delta}_N)_k a_k \sigma(\boldsymbol{w}_k^{\mathsf{T}} \boldsymbol{x}_j)(\boldsymbol{\delta}_N)_k(\boldsymbol{\delta}_N)_r\right]$$

$$= \mathbb{E}\left[\sum_{k'=1,k'\neq k,k'\neq r}^{m} (\boldsymbol{\delta}_N)_{k'} a_{k'} \sigma(\boldsymbol{w}_{k'}^{\mathsf{T}} \boldsymbol{x}_i) - y_i\right] a_k \sigma(\boldsymbol{w}_k^{\mathsf{T}} \boldsymbol{x}_j) \mathbb{E}\left[(\boldsymbol{\delta}_N)_k^2(\boldsymbol{\delta}_N)_r\right] = \frac{a_k \sigma(\boldsymbol{w}_k^{\mathsf{T}} \boldsymbol{x}_j)}{p} e_{i,\backslash k,\backslash r},$$

by similar reasoning and symmetry, the third part reads

$$\mathbb{E}\left[\left(\sum_{k'=1,k'\neq k,k'\neq r}^{m} (\boldsymbol{\delta}_N)_{k'} a_{k'} \sigma(\boldsymbol{w}_{k'}^{\mathsf{T}} \boldsymbol{x}_i) - y_i\right)(\boldsymbol{\delta}_N)_r a_r \sigma(\boldsymbol{w}_r^{\mathsf{T}} \boldsymbol{x}_j)(\boldsymbol{\delta}_N)_k(\boldsymbol{\delta}_N)_r\right]$$

$$= \mathbb{E}\left[\sum_{k'=1,k'\neq k,k'\neq r}^{m} (\boldsymbol{\delta}_N)_{k'} a_{k'} \sigma(\boldsymbol{w}_{k'}^{\mathsf{T}} \boldsymbol{x}_i) - y_i\right] a_r \sigma(\boldsymbol{w}_r^{\mathsf{T}} \boldsymbol{x}_j) \mathbb{E}\left[(\boldsymbol{\delta}_N)_k(\boldsymbol{\delta}_N)_r^2\right] = \frac{a_r \sigma(\boldsymbol{w}_r^{\mathsf{T}} \boldsymbol{x}_j)}{p} e_{i,\backslash k,\backslash r},$$

also by similar reasoning and symmetry, the fourth part reads

$$\mathbb{E}\left[\left(\sum_{k'=1,k'\neq k,k'\neq r}^{m}(\boldsymbol{\delta}_N)_{k'}a_{k'}\sigma(\boldsymbol{w}_{k'}^\intercal\boldsymbol{x}_j)-y_j\right)(\boldsymbol{\delta}_N)_k a_k\sigma(\boldsymbol{w}_k^\intercal\boldsymbol{x}_i)(\boldsymbol{\delta}_N)_k(\boldsymbol{\delta}_N)_r\right]$$

$$=\mathbb{E}\left[\sum_{k'=1,k'\neq k,k'\neq r}^{m}(\boldsymbol{\delta}_N)_{k'}a_{k'}\sigma(\boldsymbol{w}_{k'}^\intercal\boldsymbol{x}_j)-y_j\right]a_k\sigma(\boldsymbol{w}_k^\intercal\boldsymbol{x}_i)\mathbb{E}\left[(\boldsymbol{\delta}_N)_k^2(\boldsymbol{\delta}_N)_r\right]=\frac{a_k\sigma(\boldsymbol{w}_k^\intercal\boldsymbol{x}_i)}{p}e_{j,\backslash k,\backslash r},$$

and the fifth part reads

$$\mathbb{E}\left[(\boldsymbol{\delta}_N)_k a_k\sigma(\boldsymbol{w}_k^\intercal\boldsymbol{x}_i)(\boldsymbol{\delta}_N)_k a_k\sigma(\boldsymbol{w}_k^\intercal\boldsymbol{x}_j)(\boldsymbol{\delta}_N)_k(\boldsymbol{\delta}_N)_r\right]=\mathbb{E}\left[(\boldsymbol{\delta}_N)_k^3(\boldsymbol{\delta}_N)_r a_k^2\sigma(\boldsymbol{w}_k^\intercal\boldsymbol{x}_i)\sigma(\boldsymbol{w}_k^\intercal\boldsymbol{x}_j)\right]$$

$$=\frac{1}{p^2}a_k^2\sigma(\boldsymbol{w}_k^\intercal\boldsymbol{x}_i)\sigma(\boldsymbol{w}_k^\intercal\boldsymbol{x}_j),$$

and the sixth part reads

$$\mathbb{E}\left[(\boldsymbol{\delta}_N)_k a_k\sigma(\boldsymbol{w}_k^\intercal\boldsymbol{x}_i)(\boldsymbol{\delta}_N)_r a_r\sigma(\boldsymbol{w}_r^\intercal\boldsymbol{x}_j)(\boldsymbol{\delta}_N)_k(\boldsymbol{\delta}_N)_r\right]$$

$$=\mathbb{E}\left[(\boldsymbol{\delta}_N)_k^2(\boldsymbol{\delta}_N)_r^2 a_k a_r\sigma(\boldsymbol{w}_k^\intercal\boldsymbol{x}_i)\sigma(\boldsymbol{w}_r^\intercal\boldsymbol{x}_j)\right]=\frac{1}{p^2}a_k a_r\sigma(\boldsymbol{w}_k^\intercal\boldsymbol{x}_i)\sigma(\boldsymbol{w}_r^\intercal\boldsymbol{x}_j),$$

also by similar reasoning and symmetry, the seventh part reads

$$\mathbb{E}\left[\left(\sum_{k'=1,k'\neq k,k'\neq r}^{m}(\boldsymbol{\delta}_N)_{k'}a_{k'}\sigma(\boldsymbol{w}_{k'}^\intercal\boldsymbol{x}_j)-y_j\right)(\boldsymbol{\delta}_N)_r a_r\sigma(\boldsymbol{w}_r^\intercal\boldsymbol{x}_i)(\boldsymbol{\delta}_N)_k(\boldsymbol{\delta}_N)_r\right]$$

$$=\mathbb{E}\left[\sum_{k'=1,k'\neq k,k'\neq r}^{m}(\boldsymbol{\delta}_N)_{k'}a_{k'}\sigma(\boldsymbol{w}_{k'}^\intercal\boldsymbol{x}_j)-y_j\right]a_r\sigma(\boldsymbol{w}_r^\intercal\boldsymbol{x}_i)\mathbb{E}\left[(\boldsymbol{\delta}_N)_k(\boldsymbol{\delta}_N)_r^2\right]=\frac{a_r\sigma(\boldsymbol{w}_r^\intercal\boldsymbol{x}_i)}{p}e_{j,\backslash k,\backslash r},$$

and the eighth part reads

$$\mathbb{E}\left[(\boldsymbol{\delta}_N)_r a_r\sigma(\boldsymbol{w}_r^\intercal\boldsymbol{x}_i)(\boldsymbol{\delta}_N)_k a_k\sigma(\boldsymbol{w}_k^\intercal\boldsymbol{x}_j)(\boldsymbol{\delta}_N)_k(\boldsymbol{\delta}_N)_r\right]=\mathbb{E}\left[(\boldsymbol{\delta}_N)_k^2(\boldsymbol{\delta}_N)_r^2 a_k a_r\sigma(\boldsymbol{w}_k^\intercal\boldsymbol{x}_i)\sigma(\boldsymbol{w}_r^\intercal\boldsymbol{x}_j)\right]$$

$$=\frac{1}{p^2}a_k a_r\sigma(\boldsymbol{w}_k^\intercal\boldsymbol{x}_i)\sigma(\boldsymbol{w}_r^\intercal\boldsymbol{x}_j),$$

and the ninth part reads

$$\mathbb{E}\left[(\boldsymbol{\delta}_N)_r a_r\sigma(\boldsymbol{w}_r^\intercal\boldsymbol{x}_i)(\boldsymbol{\delta}_N)_r a_r\sigma(\boldsymbol{w}_r^\intercal\boldsymbol{x}_j)(\boldsymbol{\delta}_N)_k(\boldsymbol{\delta}_N)_r\right]$$

$$=\mathbb{E}\left[(\boldsymbol{\delta}_N)_k(\boldsymbol{\delta}_N)_r^3 a_r^2\sigma(\boldsymbol{w}_r^\intercal\boldsymbol{x}_i)\sigma(\boldsymbol{w}_r^\intercal\boldsymbol{x}_j)\right]=\frac{1}{p^2}a_r^2\sigma(\boldsymbol{w}_r^\intercal\boldsymbol{x}_i)\sigma(\boldsymbol{w}_r^\intercal\boldsymbol{x}_j).$$

To sum up,

$$\mathbb{E}\left[e_i^N e_j^N(\boldsymbol{\delta}_N)_k(\boldsymbol{\delta}_N)_r\right]$$

$$=e_{i,\backslash k,\backslash r}e_{j,\backslash k,\backslash r}+\left(\frac{1}{p}-1\right)\left(\sum_{k'=1,k'\neq k,k'\neq r}^{m}a_{k'}^2\sigma(\boldsymbol{w}_{k'}^\intercal\boldsymbol{x}_i)\sigma(\boldsymbol{w}_{k'}^\intercal\boldsymbol{x}_j)\right)+\frac{a_k\sigma(\boldsymbol{w}_k^\intercal\boldsymbol{x}_j)}{p}e_{i,\backslash k,\backslash r}$$

$$+\frac{a_r\sigma(\boldsymbol{w}_r^\intercal\boldsymbol{x}_j)}{p}e_{i,\backslash k,\backslash r}+\frac{a_k\sigma(\boldsymbol{w}_k^\intercal\boldsymbol{x}_i)}{p}e_{j,\backslash k,\backslash r}+\frac{1}{p^2}a_k^2\sigma(\boldsymbol{w}_k^\intercal\boldsymbol{x}_i)\sigma(\boldsymbol{w}_k^\intercal\boldsymbol{x}_j)$$

$$+\frac{1}{p^2}a_k a_r\sigma(\boldsymbol{w}_k^\intercal\boldsymbol{x}_i)\sigma(\boldsymbol{w}_r^\intercal\boldsymbol{x}_j)+\frac{a_r\sigma(\boldsymbol{w}_r^\intercal\boldsymbol{x}_i)}{p}e_{j,\backslash k,\backslash r}$$

$$+\frac{1}{p^2}a_k a_r\sigma(\boldsymbol{w}_k^\intercal\boldsymbol{x}_i)\sigma(\boldsymbol{w}_r^\intercal\boldsymbol{x}_j)+\frac{1}{p^2}a_r^2\sigma(\boldsymbol{w}_r^\intercal\boldsymbol{x}_i)\sigma(\boldsymbol{w}_r^\intercal\boldsymbol{x}_j),$$

and

$$
\mathbb{E}\left[e_i^N(\boldsymbol{\delta}_N)_k\right]\mathbb{E}\left[e_j^N(\boldsymbol{\delta}_N)_r\right]
$$

$$
=\left(e_{i,\backslash k,\backslash r}+a_r\sigma(\boldsymbol{w}_r^\intercal\boldsymbol{x}_i)+\frac{1}{p}a_k\sigma(\boldsymbol{w}_k^\intercal\boldsymbol{x}_i)\right)\left(e_{j,\backslash k,\backslash r}+a_k\sigma(\boldsymbol{w}_k^\intercal\boldsymbol{x}_j)+\frac{1}{p}a_r\sigma(\boldsymbol{w}_r^\intercal\boldsymbol{x}_j)\right)
$$

$$
=e_{i,\backslash k,\backslash r}e_{j,\backslash k,\backslash r}+e_{i,\backslash k,\backslash r}a_k\sigma(\boldsymbol{w}_k^\intercal\boldsymbol{x}_j)+\frac{1}{p}e_{i,\backslash k,\backslash r}a_r\sigma(\boldsymbol{w}_r^\intercal\boldsymbol{x}_j)+a_r\sigma(\boldsymbol{w}_r^\intercal\boldsymbol{x}_i)e_{j,\backslash k,\backslash r}
$$

$$
+a_ra_k\sigma(\boldsymbol{w}_r^\intercal\boldsymbol{x}_i)\sigma(\boldsymbol{w}_k^\intercal\boldsymbol{x}_j)+\frac{1}{p}a_r^2\sigma(\boldsymbol{w}_r^\intercal\boldsymbol{x}_i)\sigma(\boldsymbol{w}_r^\intercal\boldsymbol{x}_j)+\frac{1}{p}a_k\sigma(\boldsymbol{w}_k^\intercal\boldsymbol{x}_i)e_{j,\backslash k,\backslash r}
$$

$$
+\frac{1}{p}a_k^2\sigma(\boldsymbol{w}_k^\intercal\boldsymbol{x}_i)\sigma(\boldsymbol{w}_k^\intercal\boldsymbol{x}_j)+\frac{1}{p^2}a_ra_k\sigma(\boldsymbol{w}_k^\intercal\boldsymbol{x}_i)\sigma(\boldsymbol{w}_r^\intercal\boldsymbol{x}_j),
$$

hence

$$
\mathrm{Cov}\left(e_i^N(\boldsymbol{\delta}_N)_k,e_j^N(\boldsymbol{\delta}_N)_r\right)
$$

$$
=\mathbb{E}\left[e_i^Ne_j^N(\boldsymbol{\delta}_N)_k(\boldsymbol{\delta}_N)_r\right]-\mathbb{E}\left[e_i^N(\boldsymbol{\delta}_N)_k\right]\mathbb{E}\left[e_i^N(\boldsymbol{\delta}_N)_r\right]
$$

$$
=\left(\frac{1}{p}-1\right)\left(\sum_{k'=1,k'\neq k,k'\neq r}^m a_{k'}^2\sigma(\boldsymbol{w}_{k'}^\intercal\boldsymbol{x}_i)\sigma(\boldsymbol{w}_{k'}^\intercal\boldsymbol{x}_j)\right)+\left(\frac{1}{p}-1\right)a_k\sigma(\boldsymbol{w}_k^\intercal\boldsymbol{x}_j)e_{i,\backslash k,\backslash r}
$$

$$
+\left(\frac{1}{p}-1\right)a_r\sigma(\boldsymbol{w}_r^\intercal\boldsymbol{x}_i)e_{j,\backslash k,\backslash r}+\left(\frac{1}{p^2}-\frac{1}{p}\right)a_r^2\sigma(\boldsymbol{w}_r^\intercal\boldsymbol{x}_i)\sigma(\boldsymbol{w}_r^\intercal\boldsymbol{x}_j)
$$

$$
+\left(\frac{1}{p^2}-\frac{1}{p}\right)a_k^2\sigma(\boldsymbol{w}_k^\intercal\boldsymbol{x}_i)\sigma(\boldsymbol{w}_k^\intercal\boldsymbol{x}_j)+\left(\frac{1}{p^2}-1\right)a_ra_k\sigma(\boldsymbol{w}_r^\intercal\boldsymbol{x}_i)\sigma(\boldsymbol{w}_k^\intercal\boldsymbol{x}_j),
$$

by summation over the indices $i$ and $j$, the covariance matrix reads

$$
\boldsymbol{\Sigma}_{kr}(\boldsymbol{\theta}_{N-1})=\mathrm{Cov}\left(\nabla_{\boldsymbol{q}_k}R_{\mathcal{S}}^{\mathrm{drop}}\left(\boldsymbol{\theta}_{N-1};\boldsymbol{\delta}_N\right),\nabla_{\boldsymbol{q}_r}R_{\mathcal{S}}^{\mathrm{drop}}\left(\boldsymbol{\theta}_{N-1};\boldsymbol{\delta}_N\right)\right)
$$

$$
=\left(\frac{1}{p}-1\right)\left(\frac{1}{n}\sum_{i=1}^n\left(e_{i,\backslash k,\backslash r}+\frac{1}{p}a_k\sigma(\boldsymbol{w}_k^\intercal\boldsymbol{x}_i)+\frac{1}{p}a_r\sigma(\boldsymbol{w}_r^\intercal\boldsymbol{x}_i)\right)\nabla_{\boldsymbol{q}_k}\left(a_k\sigma(\boldsymbol{w}_k^\intercal\boldsymbol{x}_i)\right)\right)
$$

$$
\otimes\left(\frac{1}{n}\sum_{i=1}^n a_k\sigma(\boldsymbol{w}_k^\intercal\boldsymbol{x}_i)\nabla_{\boldsymbol{q}_r}\left(a_r\sigma(\boldsymbol{w}_r^\intercal\boldsymbol{x}_i)\right)\right)
$$

$$
+\left(\frac{1}{p}-1\right)\left(\frac{1}{n}\sum_{i=1}^n a_r\sigma(\boldsymbol{w}_r^\intercal\boldsymbol{x}_i)\nabla_{\boldsymbol{q}_k}\left(a_k\sigma(\boldsymbol{w}_k^\intercal\boldsymbol{x}_i)\right)\right)
$$

$$
\otimes\left(\frac{1}{n}\sum_{i=1}^n\left(e_{i,\backslash k,\backslash r}+a_k\sigma(\boldsymbol{w}_k^\intercal\boldsymbol{x}_i)+\frac{1}{p}a_r\sigma(\boldsymbol{w}_r^\intercal\boldsymbol{x}_i)\right)\nabla_{\boldsymbol{q}_r}\left(a_r\sigma(\boldsymbol{w}_r^\intercal\boldsymbol{x}_i)\right)\right),
$$

## K   THE STRUCTURAL SIMILARITY BETWEEN HESSIAN AND COVARIANCE

We can derive the Hessian of the loss function in the expectation sense with respect to the dropout noise $\boldsymbol{\delta}$ and the covariance matrix of dropout noise under intuitive approximations. We first show our assumptions as follows:

**Assumption 1.** *The NN piece-wise linear activation.*

**Assumption 2.** *The parameters of NN's output layer are fixed during training.*

**Assumption 3.** *We study the loss landscape after training reaches a stable stage, i.e., the loss function in the sense of expectation is small enough,*

$$\mathbb{E}_{\boldsymbol{\delta}} \nabla_{\boldsymbol{\theta}} R_{\mathcal{S}}^{\mathrm{drop}}(\boldsymbol{\theta}; \boldsymbol{\delta}) \approx \mathbf{0}.$$

**Hessian matrix with dropout regularization** Based on the Assumption 1, 2, the Hessian matrix of the loss function with respect to $f_{\boldsymbol{\theta},\boldsymbol{\delta}}^{\mathrm{drop}}(\boldsymbol{x})$ can be written in the mean sense as:

$$\boldsymbol{H}(\boldsymbol{\theta}) \approx \frac{1}{n} \sum_{i=1}^{n} \left[ \nabla_{\boldsymbol{\theta}} f_{\boldsymbol{\theta}}(\boldsymbol{x}_i) \otimes \nabla_{\boldsymbol{\theta}} f_{\boldsymbol{\theta}}(\boldsymbol{x}_i) + \frac{1-p}{p} \sum_{r=1}^{m} \nabla_{\boldsymbol{q}_r} (a_r \sigma(\boldsymbol{w}_r^{\mathsf{T}} \boldsymbol{x}_i)) \otimes \nabla_{\boldsymbol{q}_r} (a_r \sigma(\boldsymbol{w}_r^{\mathsf{T}} \boldsymbol{x}_i)) \right],$$

where $\boldsymbol{H}(\boldsymbol{\theta}) := \nabla_{\boldsymbol{\theta}}^2 L_{\mathcal{S}}(\boldsymbol{\theta})$.

*Proof.* We first compute the Hessian matrix after taking expectations with respect to the dropout variable,

$$\nabla_{\boldsymbol{\theta}}^2 L_{\mathcal{S}}(\boldsymbol{\theta}) = \nabla_{\boldsymbol{\theta}}^2 R_{\mathcal{S}}(\boldsymbol{\theta}) + \frac{1-p}{2np} \sum_{i=1}^{n} \sum_{r=1}^{m} \nabla_{\boldsymbol{q}_r}^2 (a_r \sigma(\boldsymbol{w}_r^{\mathsf{T}} \boldsymbol{x}_i))^2. \tag{53}$$

The first and second terms on the RHS of the Eq. (53) are as follows,

$$\nabla_{\boldsymbol{\theta}}^2 R_{\mathcal{S}}(\boldsymbol{\theta}) = \frac{1}{n} \sum_{i=1}^{n} \left( \nabla_{\boldsymbol{\theta}} f_{\boldsymbol{\theta}}(\boldsymbol{x}_i) \otimes \nabla_{\boldsymbol{\theta}} f_{\boldsymbol{\theta}}(\boldsymbol{x}_i) + (f_{\boldsymbol{\theta}}(\boldsymbol{x}_i) - y_i) \cdot \nabla_{\boldsymbol{\theta}}^2 f_{\boldsymbol{\theta}}(\boldsymbol{x}_i) \right)$$

$$\frac{1-p}{2np} \sum_{i=1}^{n} \sum_{r=1}^{m} \nabla_{\boldsymbol{q}_r}^2 (a_r \sigma(\boldsymbol{w}_r^{\mathsf{T}} \boldsymbol{x}_i))^2$$

$$= \frac{1-p}{np} \sum_{i=1}^{n} \sum_{r=1}^{m} \left( \nabla_{\boldsymbol{q}_r} (a_r \sigma(\boldsymbol{w}_r^{\mathsf{T}} \boldsymbol{x}_i)) \otimes \nabla_{\boldsymbol{q}_r} (a_r \sigma(\boldsymbol{w}_r^{\mathsf{T}} \boldsymbol{x}_i)) + (a_r \sigma(\boldsymbol{w}_r^{\mathsf{T}} \boldsymbol{x}_i)) \cdot \nabla_{\boldsymbol{q}_r}^2 (a_r \sigma(\boldsymbol{w}_r^{\mathsf{T}} \boldsymbol{x}_i))^2 \right).$$

Note that for linear activate function, $\nabla_{\boldsymbol{\theta}}^2 f_{\boldsymbol{\theta}}(\boldsymbol{x}_i) = \nabla_{\boldsymbol{q}_r}^2 (a_r \sigma(\boldsymbol{w}_r^{\mathsf{T}} \boldsymbol{x}_i))^2 = \mathbf{0}$, $a.e. \ \forall i \in [n], \forall r \in [m]$, we have

$$\nabla_{\boldsymbol{\theta}}^2 R_{\mathcal{S}}(\boldsymbol{\theta}) = \frac{1}{n} \sum_{i=1}^{n} \nabla_{\boldsymbol{\theta}} f_{\boldsymbol{\theta}}(\boldsymbol{x}_i) \otimes \nabla_{\boldsymbol{\theta}} f_{\boldsymbol{\theta}}(\boldsymbol{x}_i)$$

$$\frac{1-p}{2np} \sum_{i=1}^{n} \sum_{r=1}^{m} \nabla_{\boldsymbol{q}_r}^2 (a_r \sigma(\boldsymbol{w}_r^{\mathsf{T}} \boldsymbol{x}_i))^2 = \frac{1-p}{np} \sum_{i=1}^{n} \sum_{r=1}^{m} \nabla_{\boldsymbol{q}_r} (a_r \sigma(\boldsymbol{w}_r^{\mathsf{T}} \boldsymbol{x}_i)) \otimes \nabla_{\boldsymbol{q}_r} (a_r \sigma(\boldsymbol{w}_r^{\mathsf{T}} \boldsymbol{x}_i)).$$

Thus the Eq. (53) can be rewritten as

$$\boldsymbol{H}(\boldsymbol{\theta}) = \frac{1}{n} \sum_{i=1}^{n} \left( \nabla_{\boldsymbol{\theta}} f_{\boldsymbol{\theta}}(\boldsymbol{x}_i) \otimes \nabla_{\boldsymbol{\theta}} f_{\boldsymbol{\theta}}(\boldsymbol{x}_i) + \frac{1-p}{p} \sum_{r=1}^{m} \nabla_{\boldsymbol{q}_r} (a_r \sigma(\boldsymbol{w}_r^{\mathsf{T}} \boldsymbol{x}_i)) \otimes \nabla_{\boldsymbol{q}_r} (a_r \sigma(\boldsymbol{w}_r^{\mathsf{T}} \boldsymbol{x}_i)) \right).$$

$\square$

**Covariance matrix with dropout regularization** Based on the Assumption 3, the covariance matrix of the loss function under the randomness of dropout variable $\boldsymbol{\delta}$ and data $\boldsymbol{x}$ can be written as:

$$\boldsymbol{\Sigma}(\boldsymbol{\theta}) \approx \frac{1}{n} \sum_{i=1}^{n} \left[ l_{i,1} \nabla_{\boldsymbol{\theta}} f_{\boldsymbol{\theta}}(\boldsymbol{x}_i) \otimes \nabla_{\boldsymbol{\theta}} f_{\boldsymbol{\theta}}(\boldsymbol{x}_i) + l_{i,2} \frac{1-p}{p} \sum_{r=1}^{m} \nabla_{\boldsymbol{q}_r} (a_r \sigma(\boldsymbol{w}_r^{\mathsf{T}} \boldsymbol{x}_i)) \otimes \nabla_{\boldsymbol{q}_r} (a_r \sigma(\boldsymbol{w}_r^{\mathsf{T}} \boldsymbol{x}_i)) \right],$$

where $l_{i,1} := (e_i)^2 + \frac{1-p}{p} \sum_{r=1}^{m} a_r^2 \sigma(\boldsymbol{w}_r^{\mathsf{T}} \boldsymbol{x}_i)^2$, $l_{i,2} := (e_i)^2$.

*Proof.* For simplicity, we approximate the loss function through Taylor expansion, which is also used in Wei et al. (2020),

$$\ell(f_{\boldsymbol{\theta}}(\boldsymbol{x}_i; \boldsymbol{\delta}), y_i) \approx \ell(f_{\boldsymbol{\theta}}(\boldsymbol{x}_i), y_i) + (f_{\boldsymbol{\theta}}(\boldsymbol{x}_i) - y_i) \sum_{r=1}^{m} a_r (\boldsymbol{\delta} - \mathbf{1})_r \sigma(\boldsymbol{w}_r^{\mathsf{T}} \boldsymbol{x}_i),$$

where $\ell(f_{\boldsymbol{\theta}}(\boldsymbol{x}_i; \boldsymbol{\delta}), y_i) = \frac{1}{2}(f_{\boldsymbol{\theta}}(\boldsymbol{x}_i; \boldsymbol{\delta}) - y_i)^2$, $\ell(f_{\boldsymbol{\theta}}(\boldsymbol{x}_i), y_i) = \frac{1}{2}(f_{\boldsymbol{\theta}}(\boldsymbol{x}_i) - y_i)^2$. The covariance matrix under dropout regularization is

$$\boldsymbol{\Sigma}(\boldsymbol{\theta}) \approx \frac{1}{n} \sum_{i=1}^{n} \mathbb{E}_{\boldsymbol{\delta}} \left( \nabla_{\boldsymbol{\theta}} \ell(f_{\boldsymbol{\theta}}(\boldsymbol{x}_i; \boldsymbol{\delta}), y_i) \otimes \nabla_{\boldsymbol{\theta}} \ell(f_{\boldsymbol{\theta}}(\boldsymbol{x}_i; \boldsymbol{\delta}), y_i) \right) - \nabla_{\boldsymbol{\theta}} \mathbb{E}_{\boldsymbol{\delta}} R_{\mathcal{S}}^{\mathrm{drop}}(\boldsymbol{\theta}; \boldsymbol{\delta}) \otimes \nabla_{\boldsymbol{\theta}} \mathbb{E}_{\boldsymbol{\delta}} R_{\mathcal{S}}^{\mathrm{drop}}(\boldsymbol{\theta}; \boldsymbol{\delta})$$

$$\approx \frac{1}{n} \sum_{i=1}^{n} \mathbb{E}_{\boldsymbol{\delta}} \left( \nabla_{\boldsymbol{\theta}} \ell(f_{\boldsymbol{\theta}}(\boldsymbol{x}_i; \boldsymbol{\delta}), y_i) \otimes \nabla_{\boldsymbol{\theta}} \ell(f_{\boldsymbol{\theta}}(\boldsymbol{x}_i; \boldsymbol{\delta}), y_i) \right).$$

Combining the properties of the dropout variable $\boldsymbol{\delta}$, we have,

$$\boldsymbol{\Sigma}(\boldsymbol{\theta}) \approx \frac{1}{n} \sum_{i=1}^{n} \nabla_{\boldsymbol{\theta}} \ell(f_{\boldsymbol{\theta}}(\boldsymbol{x}_i), y_i) \otimes \nabla_{\boldsymbol{\theta}} \ell(f_{\boldsymbol{\theta}}(\boldsymbol{x}_i), y_i)$$

$$+ \frac{1}{n} \sum_{i=1}^{n} \mathbb{E}_{\boldsymbol{\delta}} \left( \sum_{r=1}^{m} (\boldsymbol{\delta} - \mathbf{1})_r \nabla_{\boldsymbol{q}_r} (a_r \sigma(\boldsymbol{w}_r^{\mathsf{T}} \boldsymbol{x}_i) e_i) \otimes \sum_{r=1}^{m} (\boldsymbol{\delta} - \mathbf{1})_r \nabla_{\boldsymbol{q}_r} (a_r \sigma(\boldsymbol{w}_r^{\mathsf{T}} \boldsymbol{x}_i) e_i) \right)$$

$$= \frac{1}{n} \sum_{i=1}^{n} \left( \nabla_{\boldsymbol{\theta}} \ell(f_{\boldsymbol{\theta}}(\boldsymbol{x}_i), y_i) \otimes \nabla_{\boldsymbol{\theta}} \ell(f_{\boldsymbol{\theta}}(\boldsymbol{x}_i), y_i) + \frac{1-p}{p} \sum_{r=1}^{m} \nabla_{\boldsymbol{q}_r} (a_r \sigma(\boldsymbol{w}_r^{\mathsf{T}} \boldsymbol{x}_i) e_i) \otimes \nabla_{\boldsymbol{q}_r} (a_r \sigma(\boldsymbol{w}_r^{\mathsf{T}} \boldsymbol{x}_i) e_i) \right)$$

$$:= \frac{1}{n} \sum_{i=1}^{n} \left( \boldsymbol{\Sigma}_1(\boldsymbol{x}_i, y_i) + \frac{1-p}{p} \boldsymbol{\Sigma}_2(\boldsymbol{x}_i, y_i) \right).$$

$$(54)$$

We calculate the two terms on the RHS of the Eq. (54) separately:

$$\boldsymbol{\Sigma}_1(\boldsymbol{x}_i, y_i) = (e_i)^2 \cdot \nabla_{\boldsymbol{\theta}} f_{\boldsymbol{\theta}}(\boldsymbol{x}_i) \otimes \nabla_{\boldsymbol{\theta}} f_{\boldsymbol{\theta}}(\boldsymbol{x}_i),$$

$$\boldsymbol{\Sigma}_2(\boldsymbol{x}_i, y_i) = (e_i)^2 \sum_{r=1}^{m} \nabla_{\boldsymbol{q}_r} (a_r \sigma(\boldsymbol{w}_r^{\mathsf{T}} \boldsymbol{x}_i)) \otimes \nabla_{\boldsymbol{q}_r} (a_r \sigma(\boldsymbol{w}_r^{\mathsf{T}} \boldsymbol{x}_i)) + \nabla_{\boldsymbol{\theta}} f_{\boldsymbol{\theta}}(\boldsymbol{x}_i) \otimes \nabla_{\boldsymbol{\theta}} f_{\boldsymbol{\theta}}(\boldsymbol{x}_i) \sum_{r=1}^{m} (a_r \sigma(\boldsymbol{w}_r^{\mathsf{T}} \boldsymbol{x}_i))^2$$

$$+ 2 \sum_{r=1}^{m} e_i a_r \sigma(\boldsymbol{w}_r^{\mathsf{T}} \boldsymbol{x}_i) \cdot \nabla_{\boldsymbol{\theta}} e_i \otimes \nabla_{\boldsymbol{q}_r} (a_r \sigma(\boldsymbol{w}_r^{\mathsf{T}} \boldsymbol{x}_i))$$

$$= (e_i)^2 \sum_{r=1}^{m} \nabla_{\boldsymbol{q}_r} (a_r \sigma(\boldsymbol{w}_r^{\mathsf{T}} \boldsymbol{x}_i)) \otimes \nabla_{\boldsymbol{q}_r} (a_r \sigma(\boldsymbol{w}_r^{\mathsf{T}} \boldsymbol{x}_i)) + \nabla_{\boldsymbol{\theta}} f_{\boldsymbol{\theta}}(\boldsymbol{x}_i) \otimes \nabla_{\boldsymbol{\theta}} f_{\boldsymbol{\theta}}(\boldsymbol{x}_i) \sum_{r=1}^{m} (a_r \sigma(\boldsymbol{w}_r^{\mathsf{T}} \boldsymbol{x}_i))^2$$

$$+ \frac{1}{2} \sum_{r=1}^{m} \nabla_{\boldsymbol{\theta}} (e_i)^2 \otimes \nabla_{\boldsymbol{q}_r} (a_r \sigma(\boldsymbol{w}_r^{\mathsf{T}} \boldsymbol{x}_i))^2.$$

Under the assumption that $\nabla_{\boldsymbol{\theta}} (e_i)^2 = 2 \cdot \nabla_{\boldsymbol{\theta}} \ell(f_{\boldsymbol{\theta}}(\boldsymbol{x}_i), y_i) = \mathbf{0}$, $\forall i \in [n]$, we have

$$\boldsymbol{\Sigma}_2(\boldsymbol{x}_i, y_i) = (e_i)^2 \sum_{r=1}^{m} \nabla_{\boldsymbol{q}_r} (a_r \sigma(\boldsymbol{w}_r^{\mathsf{T}} \boldsymbol{x}_i)) \otimes \nabla_{\boldsymbol{q}_r} (a_r \sigma(\boldsymbol{w}_r^{\mathsf{T}} \boldsymbol{x}_i)) + \nabla_{\boldsymbol{\theta}} f_{\boldsymbol{\theta}}(\boldsymbol{x}_i) \otimes \nabla_{\boldsymbol{\theta}} f_{\boldsymbol{\theta}}(\boldsymbol{x}_i) \sum_{r=1}^{m} (a_r \sigma(\boldsymbol{w}_r^{\mathsf{T}} \boldsymbol{x}_i))^2.$$

Thus the Eq. (54) can be rewritten as

$$\boldsymbol{\Sigma}(\boldsymbol{\theta}) = \frac{1}{n} \sum_{i=1}^{n} \nabla_{\boldsymbol{\theta}} f_{\boldsymbol{\theta}}(\boldsymbol{x}_i) \otimes \nabla_{\boldsymbol{\theta}} f_{\boldsymbol{\theta}}(\boldsymbol{x}_i) \left( (e_i)^2 + \frac{1-p}{p} \sum_{r=1}^{m} (a_r \sigma(\boldsymbol{w}_r^{\mathsf{T}} \boldsymbol{x}_i))^2 \right)$$

$$+ \frac{1-p}{np} \sum_{i=1}^{n} \sum_{r=1}^{m} (e_i)^2 \cdot \nabla_{\boldsymbol{q}_r} (a_r \sigma(\boldsymbol{w}_r^{\mathsf{T}} \boldsymbol{x}_i)) \otimes \nabla_{\boldsymbol{q}_r} (a_r \sigma(\boldsymbol{w}_r^{\mathsf{T}} \boldsymbol{x}_i)).$$

Note that

$$(e_i)^2 + \frac{1-p}{p} \sum_{r=1}^m (a_r \sigma(\boldsymbol{w}_r^\mathsf{T} \boldsymbol{x}_i))^2 = \mathbb{E}_{\boldsymbol{\delta}} 2\ell(f_{\boldsymbol{\theta}}(\boldsymbol{x}_i; \boldsymbol{\delta}), y_i),$$

we have

$$\boldsymbol{\Sigma}(\boldsymbol{\theta}) = \frac{2}{n} \sum_{i=1}^n \mathbb{E}_{\boldsymbol{\delta}} \ell(f_{\boldsymbol{\theta}}(\boldsymbol{x}_i; \boldsymbol{\delta}), y_i) \cdot \nabla_{\boldsymbol{\theta}} f_{\boldsymbol{\theta}}(\boldsymbol{x}_i) \otimes \nabla_{\boldsymbol{\theta}} f_{\boldsymbol{\theta}}(\boldsymbol{x}_i)$$

$$+ \frac{2(1-p)}{np} \sum_{i=1}^n \sum_{r=1}^m (\ell(f_{\boldsymbol{\theta}}(\boldsymbol{x}_i), y_i)) \cdot \nabla_{\boldsymbol{q}_r}(a_r \sigma(\boldsymbol{w}_r^\mathsf{T} \boldsymbol{x}_i)) \otimes \nabla_{\boldsymbol{q}_r}(a_r \sigma(\boldsymbol{w}_r^\mathsf{T} \boldsymbol{x}_i)).$$

$\square$

