# OpenReview forum: "Stochastic Modified Equations and Dynamics of Dropout Algorithm"
_ICLR.cc/2024/Conference — ICLR 2024 poster_

### Official Review · Reviewer_hhFg · 2023-10-26

**Soundness:** 3 good
**Presentation:** 3 good
**Contribution:** 3 good
**Rating:** 6
**Confidence:** 4

**Summary:**

This work analyzes the generalization capabilities of dropout through the lens of stochastic modified equations. The authors derive a weak-sense stochastic continuous time limit approximation of the discrete dropout algorithm and use this modified ODE to explain the phenomenon of dropout. This SDE is driven by a deterministic modified loss (due to dropout) and a stochastic Brownian motion with a covariance vector that also depends on the dropout parameter. The authors derive their conclusion from the similarity between the strcuture of the Hessian and the stochastic noise covariance matrix which could aid in getting flatter minimas as sharper directions would have more stochastic noise (due to similarity between Hessian structure and covariance matrix).

**Strengths:**

1) The paper is moderately well written.
2) Analyzing the effect of droput for generalization is an important problem in ML.

**Weaknesses:**

Although the paper seems to be easy to read at first glance, there are some portions where the authors do not clearly mention how they obtain certain equations, for example:

1) In section-4.1, how the modified loss was derived is not very clear. It seems clear from the appendix. Better to refer it to the appendix or provide a statement on how it was derived.

2) It's not evident how the authors get equation 8 from section 4.1. I missed the part where the stochasticity V comes into the picture from section 4.1. From Theorem-1, it is not clear, what effect dropout is having as the author's do not clearly mention the source of stochasticity.
It is well known a stochastic discrete algorithm can be well approximated by Euler-Maruyama discretization of an SDE, but in the manuscript the connection/derivation is not clear.

Now here are some technical concerns I had:

3) The role of the modified loss $L_{S}$ from the SDE is not clear in terms of generalization. Authors only use the flatness argument from the Hessian-covaraince alignment, but the deterministic part driving the SDE seems to not play any role ?

5) It's not quite clear how the Hessian alignment with the covariance matrix is aiding to flatter minimas. Does the eigenvector directions of the covariance matrix also correspond exactly as that of the Hessian? Given, the equations in 5.2, it is not clear how the structure of the Hessian and the Cov matrix are the same. The coefficients (involving $e_{i}$) seem to suggest that the alignment may not be same. Infact near convegence, $l_{i,2}$ tends to 0, making the second term in the covariance term vanish.

6) How the effect of dropout is in aiding to flat minima correspond to that of SGD and parameter noise injection. In some sense, dropout has some correspondace to both. Parametrer noise injection is well known as the explicit hessian regularizer (more explicitly on the trace of the hessian https://arxiv.org/abs/2206.04613). In my opinion, a comparison between SGD, parameter noise-injection and dropout are critical.

minor:

4) Which SME is used in the experiments?  The order-1 approximation or the order-2 approximation .

**Questions:**

Answers to questions posed in 3 to 7 are critical in my opinion as they form the basic claims and crux of the paper.
Some sections in the manuscript need to be modified to make reading easier, see point 1 and 2.

---

> ### Author Response · Authors · 2023-11-18
> **Official Comment by Authors (Part I)**
>
> $\textbf{Point 1}$
>
> In section-4.1, how the modified loss was derived is not very clear. It seems clear from the appendix. Better to refer it to the appendix or provide a statement on how it was derived.
>
> $\textbf{Reply}$
>
> We thank the reviewer for pointing this out, and we have added some brief derivation and discussion about $L_{S}$ in Section 4.1 of the revised manuscript to help readers understand $L_{S}$. Meanwhile, we have added the following content at the end of 4.1 to help readers refer to the detailed derivation part more conveniently: "Please refer to Appendix E.1 for the detailed derivation of $L_{S}$"
>
> $\textbf{Point 2}$
>
> It's not evident how the authors get equation 8 from section 4.1. I missed the part where the stochasticity V comes into the picture from section 4.1. From Theorem-1, it is not clear, what effect dropout is having as the author's do not clearly mention the source of stochasticity. It is well known a stochastic discrete algorithm can be well approximated by Euler-Maruyama discretization of an SDE, but in the manuscript the connection/derivation is not clear.
>
> $\textbf{Reply}$
>
> $V$ stands for the fluctuation term of dropout,   and we point out in the very front of Section 4.1 that the stochasticity originates from the random vector $\eta=(\eta_1, \eta_2, \cdots, \eta_m)^T$ that randomly drops the parameters $\theta$ from time to time at each step. To avoid such confusion,  we add some comments at the very start of Section 4.2 "In  pursuit of a more comprehensive understanding of the dynamics of dropout, we  integrate the fluctuation term of dropout into our analysis."  We remark that the modified equation approach relies heavily on the Euler-Maruyama discretization of   SDE. As is shown already in   Equation (9) and (10), in order to align the dropout iteration with the   Euler-Maruyama discretization of the SDE, we thereby choose the coefficients $b$ and $\sigma$ to match the coefficients in the discrete iteration process of dropout. This meticulous alignment ensures that the trajectories of the dropout iteration and the Euler-Maruyama discretization remain proximate up to the first order.  It is crucial to highlight that Theorem 1 serves as an approximation result, with the emphasis placed on providing an effective approximation rather than expounding on the origin of stochasticity.
>
>
> $\textbf{Point 3}$
>
> The role of the modified loss $L_S$ from the SDE is not clear in terms of generalization. Authors only use the flatness argument from the Hessian-covariance alignment, but the deterministic part driving the SDE seems to not play any role ?
>
> $\textbf{Reply}$
>
> Zhang and Xu study the influence of the drift term, i.e., $L_S$, on training, and verify experimentally and theoretically that the drift term can significantly improve the flatness of the model loss landscape compared with GD without dropout, and promote the condensation. Meanwhile, they experimentally verify that $L_S$ has better performance than GD without dropout at a small learning rate. Meanwhile, in this work, we demonstrate the promotion effect of dropout noise on generalization ability, which is supported by Fig. 2(b) in the main text, i.e., the comparison between the model trained with $L_S$ and the model trained with dropout. Based on the above two points, we believe that both the drift term and the dropout noise contribute to generalization.
>
> Zhang and Xu. Implicit regularization of dropout. arxiv 2207.05952

---

> ### Author Response · Authors · 2023-11-18
> **Official Comment by Authors (Part II)**
>
> $\textbf{Point 4}$
>
> It's not quite clear how the Hessian alignment with the covariance matrix is aiding to flatter minima. Does the eigenvector directions of the covariance matrix also correspond exactly as that of the Hessian? Given, the equations in 5.2, it is not clear how the structure of the Hessian and the Cov matrix are the same. The coefficients (involving $e_i$) seem to suggest that the alignment may not be same. Infact near convegence, $l_{i, 2}$ tends to 0, making the second term in the covariance term vanish.
>
> $\textbf{Reply}$
>
> The Inverse Variance-Flatness relation, extensively discussed in Appendix B, delves into a more intricate exploration of the interplay between the Hessian matrix and the covariance matrix eigenspace. Additionally, we have conducted numerical experiments in Appendix B.3 to investigate the alignment of eigenvectors with a significant energy proportion between the Hessian matrix and the covariance matrix. The experimental findings underscore a pronounced alignment property in the eigenspace of both matrices.
>
> Regarding the equation in Section 5.2, we acknowledge your point. This equation does not offer a detailed portrayal of the similarity between the Hessian matrix and the covariance matrix; rather, it serves as a heuristic tool. Our primary objective is to empirically examine the alignment and structural similarity of their eigenspaces. In this equation, $\nabla_{\theta} f_{\theta}\left(x_{i}\right)$ and $\nabla_{q_r}\left(a_r\sigma(w_r^{T}x_i)\right)$ can be analogized to eigenvectors, inspiring us to explore the similarity in their eigenspace structures.
>
> Moreover, based on the experimental results in Appendix B.3, we observe that towards the end of training, both the Hessian matrix and the covariance matrix typically possess only a few significant eigenvalues (nine in our experiment). During this phase, the eigenvectors corresponding to these significant eigenvalues in both matrices dictate the level of similarity between them. Consequently, the impact of $\nabla_{q_r}\left(a_r\sigma(w_r^{T}x_i)\right)$ on the model structure may become negligible at the end of training. This observation elucidates why, even in cases where $l_{i,2} \approx 0$, the two matrices still exhibit high similarity.
>
> $\textbf{Point 5}$
>
> How the effect of dropout is in aiding to flat minima correspond to that of SGD and parameter noise injection. In some sense, dropout has some correspondace to both. Parametrer noise injection is well known as the explicit hessian regularizer (more explicitly on the trace of the hessian https://arxiv.org/abs/2206.04613). In my opinion, a comparison between SGD, parameter noise-injection and dropout are critical.
>
> $\textbf{Reply}$
>
> The above three methods are all common training techniques. They improve the generalization ability of the model by adding different forms of noise during the training process. Among them, the noise added by SGD is an unbiased noise, while the noise added by parameter noise injection and dropout is biased noise. The bias term and the variance term in the noise jointly affect the training process. Meanwhile, the variance terms of the three, even if they have different structures, appear to be able to improve the flatness of the model, thus affecting the generalization ability of the model.
>
> We experimentally compared the differences in output, flatness, and generalization capabilities of the models obtained by the three training strategies. The parameter noise-injection method follows the mentioned article (https://arxiv.org/abs/2206.04613) and the corresponding code in github. Under the settings we are concerned about, i.e., the one-dimensional fitting task and the MNIST classification task, dropout has achieved better performance in multiple parallel experiments. For specific settings and results, please refer to Appendix D in the revised manuscript.
>
> $\textbf{Point 6}$
>
> Which SME is used in the experiments? The order-1 approximation or the order-2 approximation.
>
> $\textbf{Reply}$
>
> We use the order-1 approximation in our numerical validation. We add the clarification of the simulation setup in the first paragraph of Section 4.3: "For the numerical simulation of the SME, unless otherwise specified, we employ the Euler-Maruyama method to approximate its dynamic evolution by the order-1 approximation." In addition, we add validation experiments for second-order SMEs.

---

> > ### Comment · Reviewer_hhFg · 2023-11-21
> > **Reviewer response**
> >
> > I thank the authors for the detailed response and the changes made in the manuscript reflecting it. I am satisfied with the responses and especially like the detailed analysis on the spectrum of the hessian and the covariance matrix in the appendix.
> >
> >  I had one final concern: the SDE analysis of first order continuous trajectory has the constant C(T), which to my knowledge is exponential in time T, usually for first order approximations. The authors further justify this "We remark that local existence of the solution to SDE and estimates of all 2l-moments of the solution to SDE can be guaranteed for smooth coefficients and sufficiently small time T". Does this hinder the conclusion of the analysis only to early time results? Is there any guarantee that the SDE is still a strong version of the dropout algorithm near convergence, where T is significantly large (where the effect of dropout is known to be more prominenent)?

---

> > > ### Author Response · Authors · 2023-11-21
> > >
> > > $\textbf{Notation Adjustment}$
> > >
> > > In response to the suggestion from Point 4 in the further review of Reviewer h4F1, we have adjusted the notation for the learning rate and the dropout variable to $\eta$ and $\delta$ respectively. Throughout the subsequent response, we adhere to this updated notation.
> > >
> > > $\textbf{Point 1}$
> > >
> > > I had one final concern: the SDE analysis of first order continuous trajectory has the constant C(T), which to my knowledge is exponential in time T, usually for first order approximations. The authors further justify this "We remark that local existence of the solution to SDE and estimates of all 2l-moments of the solution to SDE can be guaranteed for smooth coefficients and sufficiently small time T". Does this hinder the conclusion of the analysis only to early time results? Is there any guarantee that the SDE is still a strong version of the dropout algorithm near convergence, where T is significantly large (where the effect of dropout is known to be more prominent)?
> > >
> > > $\textbf{Reply}$
> > >
> > > We concur with the reviewer's observation that $C(T)$ exhibits exponential growth over time. In a straightforward scenario, as we employ the Euler scheme for a linear differential equation $\frac{d \theta}{d t}=\mu \theta$, it reveals that the numerical error at the n-th step with step size $\eta$ satisfies the relation $E_{n+1} = E_n(1+\mu \eta)$. For sufficiently large $T$, this leads to   $E_n=(1+\mu \eta)^nE_0\leq \exp(\mu T) E_0$. This example underscores the inherent exponential nature of $C(T)$ over time via the discrete step approximation.
> > >
> > > Moreover, our analysis lacks assurance that this approximation holds for sufficiently large $T$ due to the inability to impose the uniform Lipschitz condition on the coefficients  $\nabla L_S$ and $\Sigma$. This limitation is highlighted in Appendix I.3 through the example of the ODE:
> > > $$
> > > \frac{d \theta}{ d t}=1+\theta^2,~~\theta(0)=0,
> > > $$
> > > where the uniform Lipschitz condition breaks down, leading to a solution, $\theta = \tan(t)$, that diverges to infinity within a finite time $T^*$.
> > >
> > > Despite these challenges, empirical evidence from our numerical experiments indicates that even with a learning rate $\eta = 1$, the SME still demonstrates the desired approximation ability for dropout.   One plausible explanation for this is provided in  Remark 11 of Li et al.[1], where they acknowledge that  ``Finally, for applications, typical loss functions have inward pointing gradients for all sufficiently large x, meaning that the iterates will be uniformly bounded almost surely.  Thus, we may simply modify the loss functions for large x (without affecting the iterates) to satisfy the conditions above".  It's important to note that the "$x$" referred to in the above remark corresponds to our parameter $\theta$.
> > >
> > >
> > > [1] Li Q, Tai C, Weinan E. Stochastic modified equations and dynamics of stochastic gradient algorithms i: Mathematical foundations[J].

---

> > > > ### Comment · Reviewer_hhFg · 2023-11-21
> > > > **Reviewer response-2**
> > > >
> > > > Thank you for your response and candid reply regarding this potential issue. Although theoretically this may remain a concern, but the empirical evidence indeed demonstrates the desired approximation ability. I would like the authors to state this issue in a statement or two and write that empirical findings suggest otherwise. **Conditioned to this change**, I am happy to recommend borderline accept. I will change my score to 6.

---

> ### Comment · Area_Chair_B2bX · 2023-11-20
> **Respond to authors' rebuttal**
>
> Please, confirm that you have read the author's response and the other reviewers' comments and indicate if you are willing to revise your rating.

---

> ### Author Response · Authors · 2023-11-22
>
> We appreciate the reviewer's suggestion to concisely address the issue. In our revised manuscript, we've added some  brief remarks following Assumption 1, acknowledging the limitations and empirical findings:  ``We remark that local existence of the solution to SDE  and estimates of all $2l$-moments  of the solution to SDE can be guaranteed for  smooth coefficients and  sufficiently small time $T^*>0$. Moreover, as  the constants $C(T^*,\Theta_0)$ and $C(T^*,\theta_0,\eta_0)$ are exponential in time, the $2l$-moments  of the solution might blow up for large enough $T^*$, which is unavoidable since we are unable to impose  the uniform Lipschitz condition on $ \nabla L_{S}$ and $\Sigma$. However, our empirical findings suggest that the SME still possess the desired approximation ability to dropout even for a  large learning rate, as shown in Figure 1 (a).''
>
> Furthermore, we are grateful for your willingness to consider raising the score for our work.

---

### Official Review · Reviewer_h4F1 · 2023-10-30

**Soundness:** 4 excellent
**Presentation:** 3 good
**Contribution:** 2 fair
**Rating:** 6
**Confidence:** 4

**Summary:**

The authors study dropout in a one hidden layer neural network. They first compute the gradient after one step averaged over the dropout randomness, and show that this is the gradient of a modified loss which penalizes the mean squared value of each neuron's contributions to the output. The authors then write down an SDE which attempts to establish an approximation of dropout dynamics which is locally consistent with the real dynamics to second order. They conclude with theory and experiments that suggest the dropout noise covariance is aligned with the Hessian.

**Strengths:**

The setup and analysis of dropout is well presented. The work uses relevant techniques to paint a picture of the inductive biases and some of the dynamical effects of the dropout procedure. The modified loss is easy to interpret, and it seems that at least at low learning rate the SDE performs quite similarly to the actual dynamics.

**Weaknesses:**

Overall, there is a question of the impact of the contribution. Everything within the paper is well executed (up to some minor comments addressed in questions), but the main result seems to be writing down the modified loss and SDE. The results about the alignment of the Hessian and dropout noise seem somewhat incomplete; I have given suggestions for improving those analyses as well.

In particular I wonder if the conclusions will generalize to the case of deeper networks, or if the qualitative picture changes, both in terms of the modified version of the loss, and the Hessian alignment. A related question is whether or not the results hold for the larger learning rates which are common in practical ML settings, and in the SGD+dropout setting. Even additional numerical evidence would be sufficient here in my opinion. Perhaps focusing experiments more on sweeps over $\epsilon$ and $p$ can improve this.

Some of the figures can also use improvement; see "Questions" for more detailed comments.

I'm looking forward to comments from the authors on these points and am open to changing my review score if they are sufficiently addressed.

Update: after author responses, many of the weaknesses were addressed and I have updated my review score.

**Questions:**

How do the results change for deeper networks? Is there evidence that the same structure should remain?

The theorems in the paper rely on small learning rate; however, typically neural networks are trained with large stepsizes (which are far from gradient flow for example). Which aspects of the theory will hold at larger learning rates?

Figure one and its related experiments could use improvement. For one, it would be better to plot e.g. panel 2 as a 2-D heatmap versus $\epsilon$ and $p$, so one can understand the effects of varying each (and any interaction that occurs). Additionally, I think its important to probe some smaller $p$ values (1e-2, 1e-1 perhaps) as the effects of dropout get more interesting with sparser activations.

For Figure 2, it is better to label each panel by the parameters directly rather than (setting 1, setting 2).

For the discussion of matrix alignment, it is worth pointing out that random PSD matrices have non-trivial correlation even in high dimensions. In particular, for random PSD matrices the alignment is related to the average eigenvalue. Therefore the statement

```
It is noteworthy that the cosine similarity between two random matrices of the same dimensions is highly improbable to exceed the threshold of 0.001.
```

is not the right one. It would be helpful to add a comparison to the plots of the covariance of two matrices with random eigenvectors, but the same eigenvalues as each of the relevant matrices. This will provide a better null model.

I would suggest finding a different notation for the stepsize (currently $\epsilon$). This will bring the notation more in line with what practitioners use, which I think will help the paper find a broader audience.

As a minor point: this line in the abstract:

```
These dual facets of our research, encompassing both theoretical derivations and empirical observations, collectively constitute a substantial contribution towards a deeper understanding of the inherent tendency of dropout to locate flatter minima.
```

doesn't really add anything for the reader - the content of the paper is what will convince me if the contribution is substantial or not!

---

> ### Author Response · Authors · 2023-11-18
> **Official Comment by Authors (Part I)**
>
> $\textbf{Point 1}$
>
> Overall, there is a question of the impact of the contribution. Everything within the paper is well executed (up to some minor comments addressed in questions), but the main result seems to be writing down the modified loss and SDE. The results about the alignment of the Hessian and dropout noise seem somewhat incomplete; I have given suggestions for improving those analyses as well.
>
> $\textbf{Reply}$
>
> Dropout has been proven to be a useful and widely adopted technique for training neural networks. The SME framework serves as a rigorous tool for comprehending the noise structure associated with dropout and establishing connections between this structure, generalization, and parameter condensation phenomena (Zhang and Xu). In the revised manuscript, we have provided more elaborate experiments and analyses with the goal of expanding the scope of our SME framework and alignment structure. We hope these changes will meet your expectations. For specific discussions, please refer to the responses addressing the detailed questions below.
>
> Zhang and Xu. Implicit regularization of dropout. arxiv 2207.05952
>
> $\textbf{Point 2}$
>
> A related question is whether or not the results hold in the SGD+dropout setting. Even additional numerical evidence would be sufficient here in my opinion.
>
> $\textbf{Reply}$
>
> We concur with the reviewer's point that theoretical analysis using GD may not comprehensively capture the behavior of dropout regularization when implemented with SGD. Nevertheless, the dropout algorithm implemented by SGD remains fundamentally a time-homogeneous Markov chain. Consequently, the first-order and second-order approximations of the SDE do not pose intrinsic challenges and the SME framework employed in this study is also well-suited for analyzing dropout algorithms implemented with SGD.
>
> Given the nature of SGD as an unbiased estimator with respect to the full sample,  we conjecture that  an order-$1$ approximation utilizing SME for SGD combined with dropout shall be in the form:
> $$
> d \theta_t=-\nabla_{\theta}L_{S}(\theta_t)d t+\widetilde{\Sigma} \left(\theta_t\right) d W_t,
> $$
> wherein the drift term remains invariant regardless of GD or SGD, while the diffusion term $\widetilde{\Sigma} \left(\theta_t\right)$ combines noise from both dropout and SGD.
>
> Based on the above SME, we numerically simulate the loss path of dropout implemented by SGD, and the simulation results further support the feasibility of our conjecture. Meanwhile, we added experiments to verify the Hessian-variance alignment combined with SGD noise. For detailed experimental settings and results, please refer to  Appendix C in the revised manuscript.
>
> $\textbf{Point 3}$
>
> How do the results change for deeper networks? Is there evidence that the same structure should remain?
>
> $\textbf{Reply}$
>
> We have a little more discussion about the difficulty of extending our analysis to the deeper or more complex NN architecture in the section "Conclusion and Discussion". Due to space limitations, please refer to Point 1 of reviewer 25p8's reply and Appendix C in the revised manuscript for detailed discussion and numerical validation of this issue.
>
> $\textbf{Point 4}$
>
> The theorems in the paper rely on small learning rate; however, typically neural networks are trained with large stepsizes (which are far from gradient flow for example). Which aspects of the theory will hold at larger learning rates?
>
> $\textbf{Reply}$
>
> We remark that regardless of any learning rate, dropout still updates the parameters recursively in the form of a time-homogeneous Markov Chain, i.e.,  $\theta_{N}=F(\theta_{N-1},\eta_N)$,  therefore, our theory holds true for any learning rate. However, it's established in numerical analysis that algorithms with higher-order accuracies are preferred. This preference is based on the well-known principle that lower learning rates (smaller than 1) lead to a favorable relationship $\varepsilon^{k_1}\leq \varepsilon^{k_2}$ whenever $k_1\geq k_2$. Therefore, we prefer opting for a small learning rate ($\varepsilon<1$) to ensure that higher-order terms can be treated as small quantities compared to lower-order terms. Moreover, the commonly employed size of learning rates is usually less than 1.
>
> It is noteworthy that as we set the learning rate to 1, the approximation ability of  SME for dropout still remains robust, as is shown in both Figure 1(b) and Figure 2(b).

---

> ### Author Response · Authors · 2023-11-18
> **Official Comment by Authors (Part II)**
>
> $\textbf{Point 5}$
>
> Figure one and its related experiments could use improvement. For one, it would be better to plot e.g. panel 2 as a 2-D heatmap versus $\varepsilon$ and $p$, so one can understand the effects of varying each (and any interaction that occurs). Additionally, I think its important to probe some smaller values (1e-2, 1e-1 perhaps) as the effects of dropout get more interesting with sparser activations.
>
> $\textbf{Reply}$
>
> Drawing inspiration from the heatmap you mentioned, we generated point plots based on varying learning rates and dropout rates, providing a numerical verification of the accuracy of first-order and second-order SME approximation orders. The results are now included in the revised manuscript as Figure 1(c,d). In the figure, it is evident that an increase in learning rate and a decrease in $p$ both contribute to an escalation in error, attributed to heightened noise. We hope that this new presentation method aligns with your expectations.
>
> Regarding the experimental study involving small values of $p$, such as $p=0.1$, we acknowledge its significance. However, we encountered challenges in training with such values. In our ongoing efforts and the preparation of a subsequent paper expanding on this conference paper, we are actively exploring avenues to secure additional computational resources to facilitate these experiments.
>
> $\textbf{Point 6}$
>
> For Figure 2, it is better to label each panel by the parameters directly rather than (setting 1, setting 2).
>
> $\textbf{Reply}$
>
> We have removed the statement for cases 1-4 and introduced the detailed setup for both $p$ and $\varepsilon$ in the caption and labels of Fig. 2.
>
> $\textbf{Point 7}$
>
> For the discussion of matrix alignment, it is worth pointing out that random PSD matrices have non-trivial correlation even in high dimensions... This will provide a better null model.
>
> $\textbf{Reply}$
>
> We thank the reviewer for pointing this out. We have corrected the construction method of the random matrix to compare the differences between the cosine similarity between two null models and the cosine similarity between the Hessian matrix and the covariance matrix fairly. Please refer to Section 5.3 in the revised manuscript for our change.
>
> $\textbf{Point 8}$
>
> I would suggest finding a different notation for the stepsize (currently $\varepsilon$). This will bring the notation more in line with what practitioners use, which I think will help the paper find a broader audience.
>
> $\textbf{Reply}$
>
> We thank the reviewer for pointing this out and we do prefer using the usual notation $\eta$ for the learning rate, however, it shall be noted that $\eta$ has also been widely employed in the context of dropout to denote the dropout random variable, and we have no choice but use $\varepsilon$ for the learning rate to avoid confusion. In our revised manuscript, we also add the following comments to clarify our settings: " We denote the learning rate of the dropout iteration by $\varepsilon$ to  mitigate potential confusion with the scaling vector  $\eta_N$."
>
> $\textbf{Point 9}$
>
> this line in the abstract: "These dual facets of our research, encompassing both theoretical derivations and empirical observations, collectively constitute a substantial contribution towards a deeper understanding of the inherent tendency of dropout to locate flatter minima.'' doesn't really add anything for the reader - the content of the paper is what will convince me if the contribution is substantial or not!
>
> $\textbf{Reply}$
>
> Thank you for your constructive feedback. We agree that the focus of the abstract should be on the content of the paper to demonstrate the substantial contribution. In response to your suggestion, we revised the abstract to ensure clarity and conciseness and removed the content mentioned in the point. The revised abstract can be seen in the revised manuscript.

---

> > ### Comment · Reviewer_h4F1 · 2023-11-20
> > **Response to revisions**
> >
> > I thank the authors for the detailed responses and revisions. Some followup comments are below.
> >
> > I do like the new appendix C. I wonder if any of those results can be moved to the main text? At the very least, I think the basic idea of the approach should be a bit better described in the main text to give readers (who may not read the appendices in detail) an idea for what the difficulties are and how verification might occur.
> >
> > I do find Figure 1 to be much improved, but still a bit hard to parse; it requires excessive zooming to see the details. I suggest that it be expanded out a bit. Also one thing I'm not sure of: it seems that for, e.g. p = 3 and $\epsilon = 1$ the errors are $O(1)$ - suggesting a bad approximation. However the loss plots seem to show the approximation is good. Can the authors comment on this discrepancy? It's possible that I simply have not understood the error units.
> >
> > I appreciate the authors properly computing the null expectation of the cosine similarity of the two PSD matrices. I'm a bit surprised that the correlation is so low; roughly speaking, it should be something like the ratio of the average eigenvalue to the RMS eigenvalue. Is there large eigenvalue spread in this setting?
> >
> > Regarding $\epsilon$ as step size: I have also seen $\delta$ used to describe the dropout vector. I wonder if there are common alternatives to $\epsilon$ for the stepsize?

---

> > > ### Author Response · Authors · 2023-11-21
> > >
> > > $\textbf{Notation Adjustment}$
> > >
> > > In response to the suggestion from Point 4, we have adjusted the notation for the learning rate and the dropout variable to $\eta$ and $\delta$ respectively. Throughout the subsequent response, we adhere to this updated notation.
> > >
> > > $\textbf{Point 1}$
> > >
> > > I do like the new appendix C. I wonder if any of those results can be moved to the main text? At the very least, I think the basic idea of the approach should be a bit better described in the main text to give readers (who may not read the appendices in detail) an idea for what the difficulties are and how verification might occur.
> > >
> > > $\textbf{Reply}$
> > >
> > > We appreciate the positive feedback from the reviewer. Due to space constraints in the main text, we have incorporated additional discussion and explanations in Section 4.3 to broaden the application scope of the SME framework, instead of relocating the results to the main text:
> > >
> > > “We also conduct experiments to validate the applicability of the SME approximation in complex networks and SGD settings. In the former, we simulate complex networks through numerical approximation of the drift term, while in the latter, we rely on the fact that SGD noise is unbiased. For a thorough discussion and detailed numerical results, please refer to Appendix C.”
> > >
> > > $\textbf{Point 2}$
> > >
> > > I do find Figure 1 to be much improved, but still a bit hard to parse; it requires excessive zooming to see the details. I suggest that it be expanded out a bit. Also one thing I'm not sure of: it seems that for, e.g. $p = 0.3$ and $\varepsilon =1$, the errors are $O(1)$ - suggesting a bad approximation. However the loss plots seem to show the approximation is good. Can the authors comment on this discrepancy? It's possible that I simply have not understood the error units.
> > >
> > > $\textbf{Reply}$
> > >
> > > We have excluded the training loss trajectory diagram Fig. 1(a) due to its substantial resemblance to the information presented in Fig. 1(b). Following this modification, we have expanded the size of the last two images in Fig. 1 to enhance readability.
> > >
> > > Addressing the issue of errors, we want to emphasize that, particularly under the condition of a large learning rate ($\eta=1$), a deviation of $O(1)$ among different SDE simulations is anticipated. This aligns with our theoretical findings, indicating that $O(\eta)$ cannot be considered a small quantity under such circumstances. Concerning the similarity of loss values among different SDE simulations, if we zoom in on the details of the training trajectory, we can find that there is still a difference between different simulation trajectories (approximately 0.001)
> > >
> > > Furthermore, we would like to highlight that, as discussed in Appendix B.3, the Hessian matrix exhibits only a few significant eigenvalues (9 out of 2500). The sparsity of the Hessian matrix is further assured when the $p$ value is relatively small, such as in the case of $p=0.3$. Based on these observations, deviations in parameters may not be directly reflected in the loss value. The direction of deviation is highly likely to align with the eigenvector corresponding to the nearly zero eigenvalues of the Hessian matrix. This assertion is supported by our verification, where we projected the direction of parameter deviations between different simulations under the settings $p=0.3$ and $\eta=1$.
> > >
> > > $\textbf{Point 3}$
> > >
> > > I appreciate the authors properly computing the null expectation of the cosine similarity of the two PSD matrices. I'm a bit surprised that the correlation is so low; roughly speaking, it should be something like the ratio of the average eigenvalue to the RMS eigenvalue. Is there large eigenvalue spread in this setting?
> > >
> > > $\textbf{Reply}$
> > >
> > > The eigenspace details of the model, on which the null model is based, are elucidated in Appendix B.3. Figure (a) displays the eigenvalue distribution of the Hessian matrix and covariance matrix. Clearly, both matrices associated with the original model exhibit only a small number of significant eigenvalues. This implies that the alignment of the two matrices primarily relies on the correlation between the eigenvectors corresponding to the significant eigenvalues.
> > >
> > > Contrastingly, for a high-dimensional random vector (2500 dimensions) in the null model, the similarity between the two is approximately 0. Therefore, the random vectors corresponding to the significant eigenvalues of the two matrices exhibit low similarity. This leads to a low correlation between the two null models.
> > >
> > > $\textbf{Point 4}$
> > >
> > > Regarding $\epsilon$ as step size: I have also seen $\delta$ used to describe the dropout vector. I wonder if there are common alternatives to $\epsilon$ for the stepsize?
> > >
> > > $\textbf{Reply}$
> > >
> > > We acknowledge that the notation mentioned by the reviewer is more appropriate. In the revised manuscript, we have adjusted the notation for the learning rate and the dropout variable to $\eta$ and $\delta$ respectively.

---

> > > > ### Comment · Reviewer_h4F1 · 2023-11-21
> > > > **Response to authors**
> > > >
> > > > I thank the authors for their engagement during this process, and appreciate the changes made. I will update my review score accordingly.

---

> > > > > ### Author Response · Authors · 2023-11-22
> > > > >
> > > > > We would like to reiterate our sincere gratitude for the recommendations you provided. Moreover, we genuinely appreciate your willingness to raise the score for our work. Should you have any additional questions or comments, please do not hesitate to share them with us.

---

### Official Review · Reviewer_Dhk2 · 2023-10-31

**Soundness:** 3 good
**Presentation:** 3 good
**Contribution:** 3 good
**Rating:** 8
**Confidence:** 3

**Summary:**

This paper examines the dropout algorithm in deep learning. The authors provide a theoretical framework by deriving stochastic modified equations to analyze the dynamics of dropout. In addition, the paper presents experimental findings that explore the relationship between the Hessian matrix and the covariance of dropout's noise

**Strengths:**

The optimization dynamics and generalization benifit of dropout is lack of understanding. This paper offers a rigorous theoretical analysis of the Stochastic Modified Equations associated with dropout. In addition, they conduct comprehensive experiments to explore the relationship between the Hessian matrix and the covariance of dropout's noise, which can unveil the genralization benifit of dropout. In summary, this article makes a substantial contribution to the understanding of dropout.

**Weaknesses:**

Theoretical analysis focused on two-layer neural networks, but it is indeed a meaningful step towards understanding dropout.

**Questions:**

- In Wu et al (2022), they provide an upper bound for the Hessian of the solution found by SGD based on the analysis of SGD noise, using dynamic stability analysis. Is it possible to develop a similar analysis that can offer an estimate of the Hessian for the solution found by dropout?

- As shown in Figure 3, the alignment measure $\alpha(\theta_t)$ appears to strengthen after the initial training phase. What could be the potential factors contributing to this observed phenomenon?


Wu, Wang, and Su (2022). The alignment property of SGD noise and how it helps select flat minima: A stability analysis.

---

> ### Author Response · Authors · 2023-11-18
>
> $\textbf{Point 1}$
>
> Theoretical analysis focused on two-layer neural networks, but it is indeed a meaningful step towards understanding dropout.
>
> $\textbf{Reply}$
>
> Thank the reviewer for the positive comment. We have a little more discussion about the difficulty of extending our analysis to the deeper or more complex NN architecture in the section "Conclusion and Discussion". Due to space limitations, please refer to Point 1 of reviewer 25p8's reply and Appendix C in the revised manuscript for detailed discussion and numerical validation of this issue.
>
> $\textbf{Point 2}$
>
> In Wu et al (2022), they provide an upper bound for the Hessian of the solution found by SGD based on the analysis of SGD noise, using dynamic stability analysis. Is it possible to develop a similar analysis that can offer an estimate of the Hessian for the solution found by dropout?
>
> $\textbf{Reply}$
>
> The dynamic stability analysis, also known as linear stability analysis, as presented in Wu et al. (2018) and Wu et al. (2022), serves as a valuable technique for approximating the dynamical behavior of stochastic gradient descent (SGD). The foundation for the Hessian estimate of the solution found by SGD   arises from Lemma 3.2 in Wu et al. (2022), wherein for a general SGD  $\theta_{t+1}=\theta_t-\eta(\nabla L(\theta_t)+\xi_t)$ where $ \xi_t$ are any noises satisfying $\mathbb{E}[\xi_t]=0,~~\mathbb{E}[\xi_t \xi_t^T]=S(\theta_t).$ Then we have for the linearized model equipped with learning rate $\eta$,  the following holds:
> $
> \mathbb{E}[L(\theta_{t+1})]=\mathbb{E}[r(\theta_t) L(\theta_t)+\eta^2 \nu(\theta_t)],
> $
> where $\nu(\theta)=\operatorname{Tr}(H S(\theta)) / 2$ and $r(\theta) \geq 0$.
>
> In our work, we demonstrate that    the dynamics of dropout follow close to the gradient descent   trajectory of the modified loss
> $
> L_S(\theta):=L(\theta)+L_1(\theta):=\frac{1}{2n}\sum_{i=1}^ne_i^2 +\frac{1-p}{2np}\sum_{i=1}^n \sum_{r=1}^m a_{r}^2\sigma(w_{r}^{T}x_i)^2,
> $
> where $L(\theta)$ is the same loss in SGD, and the extra term $L_1(\theta)$ accounts for the explicit regularization effect introduced by dropout. Due to the presence of term $L_1(\theta)$ in the dropout dynamics,  direct application of the linear stability analysis technique is not straightforward. However, we propose a reasonable approach by considering that as the dropout dynamics follow the gradient descent trajectory of the modified loss $L_S(\theta)$,    linearizing the dropout dynamics leads to an expression of the form: $\theta_{t+1}\approx\theta_t-\eta \nabla^2 L_S(\theta^*)(\theta_t-\theta^*),$ where $\theta^*$ is now one of the global minimum of $L_S(\theta)$. Following the high-level idea from the proof of Lemma 3.2, and considering the dropout dynamics as $\theta_{t+1}=\theta_t-\eta\left(\nabla L_S\left(\theta_t\right)+\xi_t\right),$
> for the above linearized model, where $\xi_t$ are any noises satisfying $\mathbb{E}[\xi_t]=0$ and  $\mathbb{E}[\xi_t \xi_t^T]=S(\theta_t)$.
> Then we have
> $\mathbb{E}[L_S(\theta_{t+1})]=\mathbb{E}[r_S(\theta_t) L_S(\theta_t)+\eta^2 v(\theta_t)],$
> where $\nu(\theta)=\operatorname{Tr}(\nabla^2 L_S\left(\theta^*\right) S(\theta)) / 2$ and $r_S(\theta) \geq 0$. Similarly, the two terms $r_S(\theta_t) L_S(\theta_t)$ and $\eta^2 v(\theta_t)$ denote the contributions from the modified gradient $\nabla L_S(\theta_t)$ and the noise $\xi_t$, and linear stability is affected by both terms simultaneously.
>
> To sum up, the above paragraph outlines a rough idea of the extension from linear stability analysis on SGD to dropout. In SGD, as the noise arises from the stochasticity involved in the selection of training samples, the noisy gradient is an unbiased estimator for the loss $L(\theta)$. However, as the dropout algorithm introduces noise through the stochastic removal of parameters,  the parameter-removal gradient is a   biased estimator for the loss $L(\theta)$, which makes the direct extension of the linear stability analysis technique hard. But as we observe that the parameter-removal gradient is an unbiased estimator for the modified loss $L_S(\theta)$, hence with slight modifications, the extension of the linear stability analysis technique to dropout becomes plausible.
>
> $\textbf{Point 3}$
>
> As shown in Figure 3, the alignment measure $\alpha(\theta_t)$ appears to strengthen after the initial training phase. What could be the potential factors contributing to this observed phenomenon?
>
> $\textbf{Reply}$
>
> The alignment between the covariance matrix and the Hessian matrix is inspired by the equations in Section 5.2. The establishment of the similarity between these two equations requires the training process at the final stage, i.e. $L_S \approx 0$. Therefore, as training proceeds, the loss value gradually decreases, and the degree of alignment gradually increases. Meanwhile, the increase in alignment helps the model escape the sharp minimum point faster, thereby improving the flatness of the model. We add a remark at the end of Section 5.3.

---

> > ### Comment · Reviewer_Dhk2 · 2023-11-21
> > **Response to Revisions**
> >
> > I thank the authors for the detailed responses and revisions.
> > My questions have been addressed, and I have improved my score.
> > Based on the supplementary experiments and theory, I think this paper is very helpful in understanding the optimization dynamics and generalization benefits of Dropout.

---

> > > ### Author Response · Authors · 2023-11-21
> > >
> > > Thank you for your positive feedback and acknowledgment of our efforts in addressing your questions and incorporating revisions. We are pleased to hear that the supplementary experiments and theory have enhanced the paper's contribution to understanding optimization dynamics and generalization benefits of Dropout. We genuinely appreciate your willingness to raise the score for our work.

---

> ### Comment · Area_Chair_B2bX · 2023-11-20
> **Respond to authors' rebuttal**
>
> Please, confirm that you have read the author's response and the other reviewers' comments and indicate if you are willing to revise your rating.

---

### Official Review · Reviewer_25p8 · 2023-11-01

**Soundness:** 3 good
**Presentation:** 3 good
**Contribution:** 3 good
**Rating:** 6
**Confidence:** 2

**Summary:**

This paper aims to deepen the understanding of dropout regularization in neural network training. The authors firstly derive stochastic modified equations (SMEs) that approximate the iterative process of the dropout algorithm. This provides valuable insights into dropout's dynamics. The study then conducts empirical investigations to explore how dropout assists in identifying flatter minima. By employing intuitive approximations and drawing analogies between the Hessian and covariance of dropout, the authors probe the mechanisms behind dropout's effectiveness. The empirical findings consistently demonstrate the presence of the Hessian-variance alignment relation throughout dropout's training process. This alignment relation, known for aiding in locating flatter minima, highlights dropout's implicit regularization effect, enhancing the model's generalization power.

**Strengths:**

1. The authors present a rigorous theoretical derivation of the stochastic modified equations that approximate the iterative process of the dropout algorithm. This theoretical framework enhances the understanding of the underlying mechanisms behind dropout regularization.

2. The empirical findings support the idea that dropout serves as an implicit regularizer by facilitating the identification of flatter minima. This discovery contributes to a more profound comprehension of dropout's intrinsic characteristics and its ability to improve the model's generalization capabilities.

**Weaknesses:**

1. The results presented in this paper are specifically applicable to shallow neural networks. The analysis and findings may not directly extend to deeper or more complex neural network architectures.

2. The findings and conclusions derived from the theoretical analysis using GD may not fully reflect the behavior and performance of dropout regularization when applied in practice with SGD.

**Questions:**

1. Assumption 1 seems to be quite strong? Any concrete example for this?

2. Do the results hold for any activation functions? Considering the impact of different activation functions on the behavior of dropout regularization would contribute to a more comprehensive understanding

3. The main theoretical result presented in this paper is an informal theorem, which may look a little bit weird.

---

> ### Author Response · Authors · 2023-11-18
> **Official Comment by Authors (Part I)**
>
> $\textbf{Point 1}$
>
> The results presented in this paper are specifically applicable to shallow neural networks. The analysis and findings may not directly extend to deeper or more complex neural network architectures.
>
> $\textbf{Reply}$
>
> We acknowledge the importance of broadening the analysis to encompass other neural network architectures. Meanwhile, we want to inform that the analysis of complex network architectures is a notoriously difficult problem for deep learning theory. Below, we demonstrate the obstacles of complex network structures.
>
> In the case of multi-layer neural networks, if dropout is applied only to the outermost layer, we can still calculate the explicit expression for the modified loss due to the linear structure. For deep neural network $f_{\theta}(x)=\sum_{r=1}^m a_r(\eta)_{r}\sigma(w_r^{T}x^{[L]})$, where  $x^{[L]}$ is the output function of a $L$-layer neural network. The modified loss $L_S(\cdot)$ reads:
>
> $$
> L_S(\theta):=\frac{1}{2n}\sum_{i=1}^ne_i^2 +\frac{1-p}{2np}\sum_{i=1}^n \sum_{r=1}^m a_{r}^2\sigma(w_{r}^{T}x_i^{[L]})^2.
> $$
>
> However, the situation becomes significantly more challenging when dropout is applied to the inner layers of the multi-layer neural network. In this scenario, obtaining a closed-form expression for the expectation $\mathbb{E}[h(\eta)]$, where $h$ is a highly nonlinear function with respect to $\eta$, becomes nearly impossible, which is a well-known difficulty in deep learning theory. However, the SME framework in this work is still valid if the drift item can be obtained explicitly. In this case, we design numerical experiments to verify the approximation ability of deep network SMEs in Appendix C.2.1 in the revised manuscript by numerically approximating the drift term. Meanwhile, we study the Hessian-variance alignment property under the deep network setting. Meanwhile, we numerically verify the Hessian-variance alignment properties in deep network settings (Fig. 3 in Section 5.3 and Appendix C.2.2), and conduct a detailed analysis of the feasibility of SME application in complex network structures (Appendix C.1).
>
> $\textbf{Point 2}$
>
> The findings and conclusions derived from the theoretical analysis using GD may not fully reflect the behavior and performance of dropout regularization when applied in practice with SGD.
>
> $\textbf{Reply}$
>
> Due to the space limitation, please refer to the response for Point 2 of Reviewer h4F1.
>
> $\textbf{Point 3}$
>
> Assumption 1 seems to be quite strong? Any concrete example for this?
>
> $\textbf{Reply}$
>
> Assumption 1 plays a crucial role in ensuring the local existence of the solution to the SME and estimates for its second, fourth, and sixth moments.   As shown in the proof of Theorem 5.2.1 in [1], a Picard iteration sequence is constructed to demonstrate the existence of the solution to the SDEs. We remark that in  Theorem 5.2.1  in [1], they enforce a linear growth condition, i.e.,
> $$
> ||b \left(\theta_t\right)||\leq C(1+||\theta_t||),~~~~||\sigma \left(\theta_t\right)||\leq C(1+||\theta_t||),
> $$
> to ensure the global existence of the solution for any time $t>0$. In contrast, our requirement is more modest, necessitating the existence of a solution only within a finite time interval $[0, T^*]$. Consequently, the validity of the Picard iteration sequence persists for sufficiently small  $T^*$, thus indicating that  Assumption 1 automatically holds for small enough $T^*$.
>
> We also would like to point out that in the work by Li et al. [2], which is the major motivation for our paper, their imposition of a uniform Lipschitz condition on  $b \left(\theta_t\right)$ and $\sigma \left(\theta_t\right)$ is noteworthy. In Remark 11, Li et al. acknowledge that "In the above results, the most restrictive condition is probably the Lipschitz condition." In comparison, we circumvent the uniform Lipschitz condition by shrinking the time interval from $[0,\infty)$ to  $[0, T^*]$. Finally, we add a line of remark right below Assumption 1 in our revised manuscript to avoid potential confusion.
>
>
> [1] Oksendal, B. Stochastic differential equations: an introduction with applications.
>
> [2] Li Q, Tai C, Weinan E. Stochastic modified equations and dynamics of stochastic gradient algorithms i: Mathematical foundations[J].

---

> ### Author Response · Authors · 2023-11-18
> **Official Comment by Authors (Part II)**
>
> $\textbf{Point 4}$
>
> Do the results hold for any activation functions? Considering the impact of different activation functions on the behavior of dropout regularization would contribute to a more comprehensive understanding
>
> $\textbf{Reply}$
>
> The validity of the presented results is not contingent on the specific choice of activation function,   as long as the activation function  $\sigma$  is continuously differentiable up to order $6$. This condition ensures the applicability of Taylor's theorem with the Lagrange form of the remainder. However, in numerical experiments, the requirement of smoothness of the activation does not present an obstacle. Experiments using ReLU as the activation function also achieved good approximation results shown in Figs. 1, 2. The effect of different activation functions on the noise structure may not be obvious - they all work towards a flat solution. The drift term may have different performances under different activation functions, and we leave it to subsequent work.
>
> To preempt any potential confusion, we have incorporated a clarifying remark at the bottom of Page 2 in the revised manuscript, explicitly stating that  "we impose hereafter that the activation function..."
>
> $\textbf{Point 5}$
>
> The main theoretical result presented in this paper is an informal theorem, which may look a little bit weird.
>
> $\textbf{Reply}$
>
>  We thank the reviewer for pointing this out and we have replaced the informal statement of Theorem 1* with the formal Theorem 1 in our revised manuscript.

---

> ### Comment · Area_Chair_B2bX · 2023-11-20
> **Respond to authors' rebuttal**
>
> Please, confirm that you have read the author's response and the other reviewers' comments and indicate if you are willing to revise your rating.

---

### Author Response · Authors · 2023-11-18

We would like to extend our sincere appreciation to the reviewers for their valuable insights and comments, which we have incorporated into our revised manuscript.    In response to the reviewers' suggestions,    we have invested considerable effort and time to implement several key modifications to our manuscript, including the incorporation of some supplementary experimental results, a  more comprehensive discussion of our approximate theorem, and several improvements to the overall presentation of the article. We use blue to highlight our major changes in the main text.

Outlined below are the specific enhancements made, and for reference to all these figures mentioned below, one may turn to the main text and the appendix of the revised manuscript.

$\textbf{Add numerical verification of the approximation order of SDE simulations.}$

We generated point plots based on varying learning rates and dropout rates, which provide numerical evidence for the convergence-order verification of the first-order and second-order SME approximations. The results are now included in the revised manuscript as  {Figure 1(c, d)}. Notably, an empirical observation can be distilled from the figure: an increase in the learning rate or a decrease in $p$ leads to an escalation in error. This phenomenon is reasonably attributed to the heightened noise introduced by larger learning rates and smaller $p$.


$\textbf{Add detailed analysis and numerical verification of the SME framework for more complex network structures and other algorithms.}$

We add a detailed feasibility analysis of the SME framework for more complex network frameworks and other random algorithms in Appendix C. To further support our feasibility analysis, we numerically simulate the loss path of dropout implemented by SGD of complex network structure. For SGD, we follow the conjectured first-order approximation SME of dropout with SGD:
$$
d \theta_t=-\nabla_{\theta}L_{S}(\theta_t)d t+\widetilde{\Sigma} \left(\theta_t\right) d W_t,
$$
wherein the drift term remains invariant regardless of GD or SGD, while the diffusion term $\widetilde{\Sigma} \left(\theta_t\right)$ combines noise from both dropout and SGD. For complex network structures, we numerically approximate the drift term through multiple random samplings. Meanwhile, we add experiments to verify the Hessian-variance alignment combined with SGD noise with complex network structures. For detailed experimental settings and results, please refer to  Appendix C in the revised manuscript.

$\textbf{Add numerical analysis of noise structures.}$

 We conduct numerical experiments in Appendix B.3 to investigate the alignment of eigenvectors with a significant energy proportion between the Hessian matrix and the covariance matrix. The high similarity of two matrices in the eigenspace can be seen as an extension phenomenon of the Inverse Variance-Flatness relation discussed in detail in the first two subsections of Appendix B. The experimental findings underscore a pronounced alignment property in the eigenspace of two matrices.

$\textbf{Add numerical comparison with other randomized algorithms.}$

We experimentally compared the differences in output, flatness, and generalization capabilities of the models obtained by the three randomized algorithms, i.e., SGD, dropout, and parameter noise-injection method (Orvieto et al). Under the settings we are concerned about, i.e., the one-dimensional fitting task and the MNIST classification task, dropout achieves better performance in multiple parallel experiments. For specific settings and results, please refer to Appendix D in the revised manuscript.

Orvieto et al. Explicit regularization in overparametrized models via noise injection.

---

### Meta-Review · Area_Chair_B2bX · 2023-12-07

**Metareview:**

Summary:

This paper aims to deepen the understanding of dropout regularization in neural network training. The authors firstly derive stochastic modified equations (SMEs) that approximate the iterative process of the dropout algorithm. This provides valuable insights into dropout's dynamics. The study then conducts empirical investigations to explore how dropout assists in identifying flatter minima. By employing intuitive approximations and drawing analogies between the Hessian and covariance of dropout, the authors probe the mechanisms behind dropout's effectiveness. The empirical findings consistently demonstrate the presence of the Hessian-variance alignment relation throughout dropout's training process. This alignment relation, known for aiding in locating flatter minima, highlights dropout's implicit regularization effect, enhancing the model's generalization power.

Strengths:

- The authors present a rigorous theoretical derivation of the stochastic modified equations that approximate the iterative process of the dropout algorithm.
- The provided theoretical framework enhances the understanding of the underlying mechanisms behind dropout regularization.
- The empirical findings support the idea that dropout serves as an implicit regularizer by facilitating the identification of flatter minima.
- The findings contribute to a more profound comprehension of dropout's intrinsic characteristics and its ability to improve the model's generalization capabilities.
- This paper offers a rigorous theoretical analysis of the Stochastic Modified Equations associated with dropout.
- The authors conduct comprehensive experiments to explore the relationship between the Hessian matrix and the covariance of dropout's noise, which can unveil the genralization benifit of dropout.
- The article makes a substantial contribution to the understanding of dropout.
- The setup and analysis of dropout is well presented.
- The work uses relevant techniques to paint a picture of the inductive biases and some of the dynamical effects of the dropout procedure.
- The paper is moderately well written.
- Analyzing the effect of droput for generalization is an important problem in ML.

Weaknesses:

- The results presented in this paper are specifically applicable to shallow neural networks. The analysis and findings may not directly extend to deeper or more complex neural network architectures.
- The findings and conclusions derived from the theoretical analysis using GD may not fully reflect the behavior and performance of dropout regularization when applied in practice with SGD.
- Theoretical analysis focused on two-layer neural networks, but it is indeed a meaningful step towards understanding dropout.
- The results about the alignment of the Hessian and dropout noise seem somewhat incomplete.
- A question is whether or not the results hold for the larger learning rates which are common in practical ML settings, and in the SGD+dropout setting.
- Not clear if the conclusions will generalize to the case of deeper networks.

Recommendation:

A majority of reviewers lean towards acceptance. I, therefore, recommend accepting the paper and encourage the authors to use the feedback provided to improve the paper for the camera ready version.

**Justification For Why Not Higher Score:**

Reviewers only lean slightly towards acceptance. Paper has still several weaknesses:

- The results presented in this paper are specifically applicable to shallow neural networks. The analysis and findings may not directly extend to deeper or more complex neural network architectures.
- The findings and conclusions derived from the theoretical analysis using GD may not fully reflect the behavior and performance of dropout regularization when applied in practice with SGD.
- Theoretical analysis focused on two-layer neural networks, but it is indeed a meaningful step towards understanding dropout.
- The results about the alignment of the Hessian and dropout noise seem somewhat incomplete.
- A question is whether or not the results hold for the larger learning rates which are common in practical ML settings, and in the SGD+dropout setting.
- Not clear if the conclusions will generalize to the case of deeper networks.

**Justification For Why Not Lower Score:**

A majority of reviewers lean towards acceptance.

---

### Decision · Program_Chairs · 2024-01-16

Accept (poster)